

# Silicon cycle in a temperate forest ecosystem: role of fine roots and litterfall recycling and influence of soil types

Marie-Pierre Turpault[1], Christophe Calvaruso[2], Gil Kirchen[1], Paul-Olivier Redon[3], Carine Cochet[1]

[1]UR 1138, INRA "Biogéochimie des Ecosystèmes Forestiers", Centre INRA de Nancy, Champenoux, 54280, France

[2]EcoSustain, Environmental Engineering Office, Research and Development, Kanfen, 57330, France

[3]Andra, Direction de la Recherche et Développement, Centre de Meuse/Haute-Marne, Route départementale 960, Bure, 55290, France

*Correspondence to*: Marie-Pierre Turpault (marie-pierre.turpault@.inra.fr)





**Abstract**

The role of forest vegetation in the silicon (Si) cycle has been widely examined. However, to date, no study has investigated the specific role of fine roots. The main objectives of our study were to assess the influence of fine roots as well as the impact of soil properties on the Si cycle in a temperate forest in northeastern France. Silicon pools and fluxes in solid and solution phases were quantified within each ecosystem compartment, i.e., the

atmosphere, aboveground and belowground tree tissues, forest floor, and different soil horizons, on three plots, each with different soil types, i.e., Dystric Cambisol (plot S1), Eutric Cambisol (plot S2), and Rendzic Leptosol (plot S3). In this study, we took advantage of a natural soil gradient, from shallow calcic soil to deep moderately acidic soil, with similar climates, atmospheric depositions, species composition and management. Soil solutions were measured monthly for four years to study the seasonal dynamics of Si fluxes. A budget of dissolved Si was

also determined for the forest floor and soil layers. Our study highlighted the major role of fine roots in the Si cycle in forest ecosystems for all soil types. Because of the abundance of fine roots mainly in the superficial soil horizons, their high Si concentration (equivalent to that of leaves and two orders higher than that of coarse roots) and their rapid turnover rate (approximately one year), the mean annual Si fluxes in fine roots in the three plots ranged from 68 to 110 kg.ha$^{-1}$.y$^{-1}$ for the Rendzic Leptosol and the Dystric Cambisol, respectively. The turnover

of fine roots and leaves was approximately 71% and 28% of the total Si taken up by trees each year, respectively, demonstrating the importance of biological recycling in the Si cycle in forests. Less than 1% of the Si taken up by trees each year accumulated in the perennial tissues. This study also demonstrated the influence of soil type on the concentration of Si in the annual tissues and therefore on the Si fluxes in forests. The concentrations of Si in leaves and fine roots were approximately 1.5-2.0 times higher in the "Si-rich" Dystric Cambisol compared to the "Si-

poor" Rendzic Leptosol. In terms of the dissolved Si budget, there were large amounts of dissolved Si in the three plots on the forest floor (9.9 to 12.7 kg.ha$^{-1}$.y$^{-1}$) and in the superficial soil horizon (5.3 to 14.5 kg.ha$^{-1}$.y$^{-1}$), and Si decreased with depth in plot S1 (1.7 kg.ha$^{-1}$.y$^{-1}$). The amount of Si leached from the soil profile was relatively low compared to the annual uptake by trees (13% in plot S1 to 29% in plot S3). The monthly measurements demonstrated that the seasonal dynamics of the dissolved Si budget were mainly linked to biological activity.

Notably, the peak of dissolved Si production in the superficial soil horizon was during the winter and probably resulted from fine root decomposition. Our study reveals that biological processes, particularly those of fine roots, play a predominant role in the Si cycle in temperate forest ecosystems, while the geochemical processes appear to be limited.



## 1 Introduction

Silicon (Si) is ubiquitous in the earth's crust (Iler, 1979) and in the soil (McKeague and Cline, 1963), where it plays an important role in soil processes through the dissolution of silicate minerals and the precipitation of secondary minerals such as clay minerals or poorly crystalline and amorphous siliceous compounds (Dixon and Weed, 1989). In the 1980s and 1990s, several studies showed that Si in soils also had a biogenic origin (Bartoli and Wilding, 1980; Lucas et al., 1993). Monosilicic acid in pore soil solution is taken up by vascular plants and is

deposited as opal into the cell walls, cell luminas and intercellular spaces (Jones and Handreck, 1965).

Plants contain between 0.1 and 15% Si (Mitani and Ma, 2005), and solid hydrated amorphous Si is mainly incorporated in leaves, stems and roots (Piperno, 1988; Drees et al 1989). Si-biomineralization in plant organs enhances plant resistance against insects and pathogenic microorganisms by acting as a physical barrier to prevent penetration and by inducing plant defence responses (Cai et al., 2008; Ma and Yamaji, 2006; Massey and Hartley,

2009). Silicon also plays a role in plant detoxification by co-precipitating with aluminium and heavy metals (He et al, 2013). Depending on the mechanism of Si uptake, i.e., exclusive, passive and active, plants belong to three main categories referred to as Si-excluders, Si-intermediate types and Si-accumulators (Takahashi et al., 1990). The Si concentration of pore water solution in soils and plant transpiration have also influenced opal formation in plants (Cornelis et al., 2010b). Thus, Si recycling by plants is influenced by soil type when the soil develops from

contrasting bedrocks (rich in silicate and poor in silicate, such as limestone).

The relevance of biological Si cycling, namely, through the biogenic pool of Si in soils induced by organic matter decomposition, also has been demonstrated for other terrestrial ecosystems, i.e., grasslands (Blecker et al., 2006; White et al., 2012), tropical forests (Lucas et al., 1993, Alexandre et al., 1997) and temperate forests (Bartoli, 1983; Watteau and Villemin, 2001; Gerard et al, 2008; Cornelis et al., 2010a; Cornelis et al., 2011a; Sommer et

al., 2006; Sommer et al., 2013). Biogenic Si pools in soil can be subdivided into protozoic Si, microbial Si and phytogenic Si pools (Sommer et al., 2006), but the phytogenic Si pool is very difficult to quantify for two main reasons. First, amorphous Si extraction methods (Biermans and Baert, 1977; Kodoma and Ross, 1991) are not specific and extract not only biogenic Si but also pedogenic Si. Second, the phytolith separation method was only developed for soil fractions larger than clay (> 2 μm) (Kelly et al. 1990; Alexandre et al., 1997), and phytogenic

Si cannot be determined in the clay fraction. However, Krieger et al (2017) recently showed that Si in deciduous trees (European beech, *Fagus sylvatica* and sycamore maple, *Acer pseudoplatanus*) generally precipitates as a thin layer (< 0.5 μm) around the cells, especially in roots and bark. This recent observation may indicate that phytogenic Si in soils of a beech forest is dominant in the clay fraction and that rapid turnover of fine roots could significantly contribute to the biogenic Si in forest ecosystems, as already demonstrated for other elements (Gordon and

Jackson, 2000). An abundance of Si in the root system, however, is contradictory to the theory that the storage of Si is limited in roots because this element is quickly transferred through soap flow to the leaves (epidermis, apex, and stomata) where it accumulates (Bartoli and Souchier, 1986). Based on this assumption and because fine root sampling and cleaning before analyses are long and tedious processes, studies in temperate forest ecosystems mainly focus on the importance of litterfall recycling on the Si biogeochemical cycle without quantifying Si in the

roots (Gérard et al., 2008; Cornelis et al., 2010a; Sommer et al., 2013). In addition, the studies carried out in acidic soils, with mor or moder humus forms, present two main constraints. First, the slow decomposition rate of the organic layer increases the chance that litterfall fluxes are non-stationary (succession of vegetation, Sommer et al, 2013), which makes it difficult to identify the factors controlling the Si cycle. Second, the large organic layer at





the basis of the mor or moder humus forms is mixed with soil particles from the superficial layer, making the

assessment of the contribution of the organic horizons to Si recycling difficult (Cornelis et al., 2010a).

Based on the recent observation that a large amount of Si precipitates in roots (Krieger et al., 2017) and on the rapid turnover of fine roots in forest ecosystems (approximately one year in beech forests in Europe; Brunner et al., 2013), we hypothesized that fine roots could significantly contribute to the input of phytogenic Si into the soil. In addition, because the Si concentration of the soil solution was demonstrated to affect Si uptake by roots, we

hypothesized that soil type could influence the Si cycle.

The aim of this work was to establish and compare the Si cycle in a temperate forest ecosystem in three different soils, and to determine the respective contribution of the different ecosystem compartments, including the roots and litterfall, on the Si cycle. The study area was chosen to ensure that only soil conditions differ between modalities, climate conditions, atmospheric deposition and stand characteristics (age, species, stem density,

distribution, and management) being similar on the study zone. The Si pools were assessed in the tree biomass by compartment for the aboveground and belowground parts, in the forest floor and in the soil at different layers. During a four-year observation period (January 2012 to December 2015), Si inputs through dust deposition and outputs through drainage and immobilization in trees, as well as Si recycling through root and litter turnover, were assessed in a mature beech forest on three different soil types, ranging from a shallow calcic soil to a deep acidic

soil. The rate of Si recycling was then compared to the Si inputs and outputs and to the Si pools in the ecosystem. An input-output budget, layer-by-layer in the soil including the soil solution Si fluxes, was done to accurately determine the Si cycle. As most Si precipitates in very thin layers in tree compartments (Krieger et al., 2017), we decided to quantify the total Si in the different ecosystem compartments without specifically assessing the Si in phytoliths, where the clay fraction is not collected by the current separation methods (Saccone et al., 2007).

**2 Materials and Methods**

**2.1 Experimental site**

The experimental site, hereafter referred as the Montiers site (http://www.nancy.inra.fr/en/Outils-et-Ressources/montiers-ecosystem-research), is in the Montiers-sur-Saulx beech forest in northeastern France (Meuse, France, latitude 48° 31' 54'' N, longitude 5° 16' 08'' E). The site is 73 ha and has been managed jointly

by the INRA-BEF (French National Institute for Agricultural Research – Biogeochemical cycles in Forest Ecosystems research unit) and by the ANDRA (French National Radioactive Waste Management Agency) since 2012. The different steps of site establishment are described in detail in Calvaruso et al. (2017). The Montiers site is part of different national and international research networks (SOERE-OPE, SOERE F-ORE-T and AnaEE). The mean annual rainfall and temperature over the last twenty years are 1069 mm and 9.8°C, respectively (from

Météo-France data). The site is covered by a homogeneous, same-aged stand (approximately 50 years old in 2010) with the same management approaches. The stand was mainly composed of beech (89%) and 11% of other deciduous species, i.e., sycamore maple, ash (*Fraxinus excelsior*), pedunculate oak (*Quercus robur* L.), European hornbeam (*Carpinus betulus* L.), and wild cherry (*Prunus avium*). The site was also composed of three different soil types, i.e., Dystric Cambisol, Eutric Cambisol, and Rendzic Leptosol (FAO, 2016). A schematic representation

of the soil profiles and their location are presented in Kirchen et al. (2017). Table 1 presents the main characteristics of these different soil types, ranging from acidic and deep soils to calcic and superficial soils, developed on acidic



Valanginian and detritic sediments and Portlandian limestone, respectively. Humus type is a eutrophic mull for the Rendzic Leptosol sediments and an acidic mull for the Dystric Cambisol sediments.

Three experimental plots (S1, S2 and S3) were built on the three different soils to monitor water and element fluxes as well as tree growth, over four years. Each plot was composed of three subplots (replicates), equipped with the same monitoring devices designed for the sampling of aboveground and belowground solutions at different depths, soil at different depths, organic horizons, litterfall, and four subplots equipped for standing aboveground and belowground biomasses as well as tree growth. In addition, a 45-m high flux tower was placed within the site (close to plot S1) to collect rainfall and atmospheric deposits.

## 2.2 Sampling

### 2.2.1 Solutions and dust deposits

Solutions and dust deposits were sampled every four weeks between January 2012 and December 2015, representing four years of monitoring.

Rainfall was collected on top of the flux tower by three polyethylene collectors (0.24 m² opening) to obtain dust deposition. The procedure of dust deposit sampling is described in Lequy et al. (2014). Briefly, rainfall was centrifuged for 40 minutes at 3500 tr.min⁻¹ to separate the solid phase from the solution (the solid phase consists of the dust deposits). Rainfall volumes were obtained from a Météo-France weather station located in Biencourt-sur-Orge (Meuse, France), which is 4.3 km from the Montiers site.

The throughfall was collected in each replicate by 4 polyethylene gutters (0.39 m² opening), placed 1.2 m above the forest ground.

The stemflow was collected in each replicate on 6 trees of different sizes, using polyethylene collars attached horizontally to the stem at 1.50 m. Trees were chosen to cover most of the range of stem circumferences (C130) in each plot. To prevent the solution from freezing, the stemflow was collected in underground storage containers during the winter.

The gravitational soil solutions (zero-tension lysimeters, ZTL) were collected beneath the forest floor and at different soil depths, -10 and -30 cm (in S1, S2 and S3), -60 cm (in S1 and S2) and -90 cm (in S1), with large plate lysimeters (40 cm * 30 cm, 0.12 m²; 3 repetitions per soil depth and per replicate) or thin rod-like lysimeters (0.07 m²; in clusters of 8; 3 repetitions per soil depth and per replicate).

The bound soil solutions (tension lysimeters, TL) were collected by ceramic cups inserted in the soil at different depths, -10 and -30 cm (in S1, S2 and S3), -60 cm (in S1 and S2) and -90 cm (in S1), with 4 repetitions per depth and per replicate. These ceramic cups were connected to an electric vacuum pump that maintained a constant depression between -0.5 and -0.6 bar.

### 2.2.2 Tree compartments

Three beech trees were harvested in each plot in 2009 to collect stem wood and bark and branches. The branches latter were separated into different classes, i.e., < 4, 4-7 and > 7 cm diameter, according to Henry et al. (2011). The detailed procedure for collecting stem wood and bark and branches is described in Calvaruso et al. (2017).

The fine roots (< 2 mm diameter) were collected during March-April 2011 in three soil pits (approximately 0.4 m wide) for each replicate, where the soil material was cut and extracted by layer ( 0-5, 5-15, 15-30, 30-45, 45-60





cm, and 60-90 cm, when possible). A two-step procedure was applied to accurately assess the fine root biomass
(Bakker et al., 2008), without having to transport soil to the laboratory (at least fifty kg of soil sample). The first
step involved collecting, *in situ*, the fine roots from the block of soil extracted from each soil layer. Then, a part
of the soil block (approximately 2 kg) was collected. The second step, at the laboratory, consisted of using a
tweezer to collect all the remaining fine roots in this soil aliquot. This second step allowed for the assessment of
the fraction of fine roots uncollected during the first step. The fine roots collected during the two steps were washed
at the laboratory, dried in a stream air-drier for three days and then weighed. For each layer, the total biomass of
fine roots was obtained by summing the fine root biomass collected during the first step and the fine root biomass
collected during the second step, multiplied by the ratio total soil block mass / soil aliquot mass. Roots with a
diameter > 2 cm (small and coarse roots) were collected in February 2017 in three soil pits (approximately 0.4 m
wide) for each plot where soil material was cut and extracted at approximately 20 cm. This method does not allow
quantification of small and coarse root biomass, which were determined through allometric equations (Le Goff
and Ottorini, 2001). An aliquot of each root sample (fine, small and coarse) was then collected to determine mineral
content. Each aliquot was carefully washed under a binocular microscope with distilled water, using tweezers and
an ultrasound gun. The absence of soil particles was carefully checked under a binocular microscope. The
operation was repeated until all soil particles were removed to prevent soil pollution in the root analyses.

The litterfall was collected in 6 litter traps (0.34 $m^2$ each) per replicate. The litter was harvested seven times per
year, avoiding litter degradation in the litter traps. During the harvest, the litter was separated into three
compartments, i.e., (i) leaves and (ii) buds, beechnuts, fruit capsules (annual compartments), and (iii) small
branches falling from the trees (perennial compartment). The leaves, buds, beechnuts, and fruit capsules belong to
annual tree compartments (recycling each year) while small branches belong to perennial compartments.

### 2.2.3 Forest floor

We defined the forest floor by the set of organic horizons (Oln, Olv, Of and Oh) above the organo-mineral horizon
(Ah), and the small dead wood at the soil surface.

Organic horizons were collected in June 2010 in a calibrated metal frame (surface area of 0.1 $m^2$). Nine samples
were collected in each replicate. Because the lower organic horizons were in direct contact with the superficial soil
horizon, it was very difficult to sample them without soil contamination. The presence of soil particles, very rich
in Si, mixed with the organic horizons, can induce a drastic overestimation of the Si pool in this compartment. As
a result, we decided to carefully sample, on site, six organic horizon samples without the fraction contacting the
soil, called "pure organic horizons". These "pure organic horizons" were used to determine the soil fraction in the
organic horizon collected on the three plots (see the method in part 2.4.2).

Small dead wood from the previous thinning (winter 2009-2010) was harvested in June 2010 at the three stations
in a calibrated metal frame (surface area of 0.6084 $m^2$). Nine samples were collected in each replicate, according
to a grid.

### 2.2.4 Soil

Nine soil samples were collected in June 2010 in each replicate, along a 15 x 15 m grid. At each point, samples
were extracted through an auger, by layer, 0-5, 5-15, 15-30, 30-45, and 45-60 cm, and 60-90 cm when possible.



### 2.3 Analytical methods

#### 2.3.1 Si content in solutions

Solutions of rainwater, stemflow, throughfall, forest floor and soil were filtered at 0.45 µm, stored at 4°C and analysed during the week following the sampling. The Si content in the solutions was measured by inductively coupled plasma-atomic emission spectrometry (ICP-AES Agilent Technologies 700 type ICP-OES, Santa Clara, USA).

#### 2.3.2 Si content in biomass

Samples from the aboveground and belowground compartments of the trees, litterfall and forest floor were dried in a stream air-drier (at 65°C), then ground and encapsulated for analysis. The total Si content in the biomass was assessed by X fluorescence, using an X Fluorescence sequential spectrometer S8 TIGER 1kW (Bruker, Marne la vallée, France).

#### 2.3.3 Si content in soil and dust deposits

The total Si content in soil organo-mineral and mineral layers (preliminarily sieved at 2 mm) and in dust deposits were determined by inductively coupled plasma-atomic emission spectrometry (700 Series ICP-OES, AGILENT TECHNOLOGIES) after alkaline fusion in $LiBO_2$ and in $HNO_3$.

#### 2.3.4 Microscopic analysis

Samples of fine roots, stem and branch bark, fruit capsules, bud scales and fresh and altered leaves (from organic horizons) of beech tree samples were mounted on glass plates, using double-coated carbon conductive tabs and covered with carbon. The samples were examined at the GeoRessources laboratory (University of Lorraine) for biomineral occurrence and composition, using a Hitachi S-4800 scanning electron microscope (SEM) equipped with an energy-dispersive X-ray spectroscopy (EDX), containing a lithium-drifted Si detector. The SEM analyses were carried out using an acceleration voltage of 10 or 15 kV.

### 2.4 Calculation of Si pools and fluxes in solutions and solids

In each plot, Si fluxes and pools were obtained by multiplying the amount of solution or solid by the concentration of Si in the given compartment. All monthly Si fluxes were calculated on a one-hectare basis and were summed over calendar years to compute the annual fluxes. The dissolved Si budget was also calculated for forest floor and soil layers by the difference between input and output fluxes.

In the following sections (2.4.1 to 2.4.10), we will only present the Si fluxes or pools for which the method of calculation differs from the calculation of multiplying the amount of solution or solid by the concentration of Si in the compartment.

#### 2.4.1 Dust deposits



To take into account the loss of particles during the collection of dust deposits from rainfall, a test using standard minerals was done to assess the efficiency of the procedure (Lequy et al., 2014). The efficiency was estimated at 72%. Thus, the total weight of dust deposits per year was determined as the weight of dust deposits collected on
site, divided by a correction factor of 0.72.

### 2.4.2 Organic horizons

The percentage of soil mixed with the organic horizons was determined through the use of Ti. This element is a good tracer of soil pollution in the collected organic horizons because Ti is in very low abundance in pure organic horizons (< 0.3 mg.kg$^{-1}$), while it is more abundant in soils (> 4 mg.kg$^{-1}$). We measured Ti content in the soil
surface layer (0-5 cm), in the pure organic horizons and in the organic horizons collected on the three plots. The percentage of soil in the organic horizons was assessed following Eq. (1):

$$\text{Soil \%} = [(Ti_{Hb} - Ti_{Hp}) / (Ti_S - Ti_{Hp})] \tag{1}$$

where $Ti_{Hb}$ is the concentration of Ti in the organic horizons, $Ti_{Hp}$ is the concentration of Ti in the pure organic horizons, and $Ti_S$ is the mean concentration of Ti in the 0-5 cm horizon of soil for each plot. The mean soil fraction
represented less than five percent of the total organic horizon mass in our study. The fraction of Si brought by soil contamination was deducted to obtain the Si content in the organic horizons.

### 2.4.3 Stemflow and stand deposition

To transform the stemflow volumes to a water flux, stem circumference at 1.30 m height (C130) was assumed to explain the inter-individual stemflow volume variability within a species. Thus, all the trees in each plot were
separated into several C130 classes, and the correlation between the stemflow volume and the C130 was verified for the entire sampling period. Using a trend line equation, a mean monthly stemflow volume was then assigned to each C130 class. The stemflow at the plot scale for a given C130 class (SF$_z$; in mm) is given by following Eq. (2):

$$SF_z = V_z \cdot (\frac{N_z}{A}) \tag{2}$$

where z is the C130 class, $V_z$ is the mean stemflow volume per tree in the given C130 class (in l), $N_z$ is the number of trees in the given C130 class and A is the plot area (in m$^2$). Total stemflow at the plot scale was obtained by summing the stemflow fluxes of all C130 classes.

The Si stand deposition, i.e., the amount of Si (kg.ha$^{-1}$.y$^{-1}$) reaching the soil after crossing over the canopy, was determined as the sum of the Si fluxes in throughfall and stemflow.

### 255 2.4.4 Drainage flux

The BILJOU© model (Granier et al., 1999) was applied in the three plots at the Montiers site to assess the water drainage flux for the different soil layers. The detailed procedure and the data are presented in Kirchen et al. (2017). The gravitational water flux was determined for each soil layer and date from the collected gravitational volume. The bound water flux was obtained by subtracting the water gravitational flux from the modelled water drainage
flux. In this study, we determined that the water gravitational flux/water bound flux ratio was approximately 80/20, which is similar to the measurement from a Cl tracer in a beech temperate forest in Fougères in Legout et al. (2009).





Thus, the monthly elements drainage fluxes were calculated at each depth following Eq. (3):

$$D_{Si} = D_G \times C_{SiG} + D_B \times C_{SiB} \tag{3}$$

where $D(X)$ is the drainage flux of Si, $D_G$ is the water drainage via rapid gravitational transfer, $C_{SiG}$ is the concentration of Si in the gravitational soil solution collected by zero-tension lysimeters, $D_B$ is the water drainage via slow bound transfer, and $C_{SiB}$ is the concentration of Si in the bound soil solution collected by ceramic cups. The element mass balances were calculated for the following soil layers, according to the installation depths of the lysimeters in the three plots: forest floor (FF), from the forest floor to -10 cm (soil layer L1), between -10 and -30

cm (L2), between -30 and -60 cm (L3) and between -60 and -90 cm (L4). For each soil layer, the mass balance of the elements was calculated as the difference between the drainage at the bottom of the layer and the drainage entering the layer (Eq. 4):

$$MB_{Si} = D_{Si2} - D_{Si1} \tag{4}$$

where $MB_{Si}$ is the mass balance of Si in a given soil layer, $D_{Si1}$ is the incoming drainage flux of Si and $D_{Si2}$ is the

drainage flux at the bottom of the soil layer.

### 2.4.5 Aboveground tree biomass

The evaluation of aboveground tree biomass was calculated according to procedures described in Saint-André et al. (2005). It included four steps, (i) the circumference of all trees was measured at 1.30 m height, $C_{1.30}$, in 2011 and 2015; (ii) eight trees in each plot, representing the range of $C_{1.30}$, stem bark and wood and 0-4, 4-7 and > 7 cm

diameter branches were sampled; (iii) the weighed allometric equations fitted for each ecosystem compartment were calculated according to Calvaruso et al. (2017); and (iv) tree biomass (stem bark and wood and 0-4, 4-7 and > 7 cm diameter branches) was quantified per hectare by applying fitted equations to the stand inventories. Annual aboveground biomass production and Si immobilization in aboveground biomass were calculated as the difference between the biomass or Si amount in the biomass calculated for 2015 and 2011, divided by four.

### 2.4.6 Fine root flux

The fine root turnover rate is dependent on the fine root biomass and the annual production but also on the various methods and calculations used to determine the rate (Jourdan et al., 2008; Gaul et al., 2009; Finer et al., 2011; Yuan and Chen, 2010). In this study, the annual fine root production was calculated by using the mean fine root turnover rate of 1.11±0.21 $y^{-1}$, issued from the last available European data compilation for beech forests (Brunner

et al., 2013). The turnover rate corresponds to the ratio between the production of fine roots during the growing season and the mean biomass of living fine roots during the year. The Si flux from fine roots was calculated by multiplying the annual fine root production by the Si concentration in the fine roots.

### 2.4.7 Small and coarse roots

The small and coarse root biomass as well as the annual root increment were determined using allometric

equations, linking the stem diameter at breast level and root biomass of beech trees (Le Goff and Ottorini, 2001). The pools and fluxes of Si in small and coarse roots were calculated by multiplying the total biomass or the annual root increment by the Si concentration in small and coarse roots.

### 2.4.8 Exploitation residuals and harvest





To take into account the influence of forestry practices after 2010 on the Si cycle, we simulated a stand thinning
based on the forestry practices applied in the Montiers massif by the French National Forestry Office. At this stage
of stand development, the National Forestry Office carries out a thinning every seven years, with an aboveground
biomass cut of approximately 40 t ha$^{-1}$. Because the amount of biomass cut is dependent on the stand aboveground
biomass, we integrated this parameter into our calculation of exploitation residuals and harvest.

We determined that the aboveground biomass that will be cut during the next thinning (winter 2017-2018) will be
approximately 40.0, 44.3, and 35.0 t ha$^{-1}$ in plots S1, S2, and S3, respectively. The root biomass remaining from
this thinning will represent approximately 7.9, 9.6, and 6.9 t ha$^{-1}$ in plots S1, S2, and S3, respectively.

From the data regarding the proportion of the different tree compartments in the total aboveground biomass at the
Montiers site (stem wood and bark, < 4 cm, 4-7 cm and > 7 cm diameter branches; Calvaruso et al. 2017), we
determined the biomass of residuals (< 4 cm, and 4-7 cm diameter branches) and exports (> 7 cm diameter
branches, stem wood and bark) issued from this thinning for each station. The roots are not exported.

Because thinning in this region is generally done every seven years, we obtained the annual Si amounts restituted
to the soil and exported by dividing the total exploitation residuals by seven.

### 2.4.9 Foliar leaching

The amount of Si released in foliar leachates throughout the year ($Si_{FL}$, in kg of Si by ha$^{-1}$.y$^{-1}$) was assessed
following Eq. 5:

$$Si_{FL} = Si_{SD} - Si_R \qquad (5)$$

where $Si_{SD}$ is the amount of Si in the stand deposition throughout the year, and $Si_R$ is the amount of Si in annual
rainfall. All these parameters are assessed in kg of Si by ha$^{-1}$.y$^{-1}$.

### 2.4.10 Tree uptake

The amount of Si taken up by trees throughout the year ($Si_{Up}$, in kg of Si by ha$^{-1}$.y$^{-1}$) was assessed following Eq.
6:

$$Si_{Up} = Si_{I_{AG}} + Si_{I_{BG}} + Si_{R_{FL}} \qquad (6)$$

where $Si_{I_{AG}}$ is the amount of Si immobilized in the total aboveground biomass of trees (stem bark and wood,
branches, leaves and buds, beechnuts and fruit capsules) throughout the year, $Si_{I_{BG}}$ is the amount of Si immobilized
in the total belowground biomass of trees (coarse, small and fine roots) throughout the year, and $Si_{R_{FL}}$ is the amount
of Si released in foliar leachates throughout the year. All these parameters were assessed in kg of Si by ha$^{-1}$.y$^{-1}$.

### 2.5 Statistical analysis

The descriptive statistical parameters (e.g., mean, standard deviation, variation coefficient) were performed using
XLSTAT 2017 software. The normality of the distribution was checked, using the Shapiro-Wilk test. As our data
did not follow a normal distribution, the non-parametrical Kruskal-Wallis test was performed to compare the
different soil types, biomass pools, biomass increments, Si content, Si pools, and Si fluxes for each tree
compartment, and the total and amorphous soil Si at the threshold level of 0.05. The post hoc Bonferroni correction
was used for the pairwise comparison. We used the R version 3.3.1 statistical software (R Development Core



Team, 2016) and specifically, the R package nlme to test the effect of soil type on annual Si fluxes, by means of a mixed linear analysis of variance (ANOVA) with soil type, year and their interaction as fixed effects. The significance of differences in element content between the gravitational and bound solutions and between plots was tested by the Student's t-test. Confidence intervals were established at the 0.05 probability level for all statistical tests.

### 3 Results

#### 3.1 Si in solids

##### 3.1.1 Microscopic observations of Si deposits in vegetation and the forest floor

In fresh leaves, Si precipitates in cell walls but also in intercellular spaces, generally forming Si deposits called phytoliths, which are several micrometres (Figure 1a). In all tree compartments, except wood, these Si deposits mostly occurred as fine coating layers thinner than 0.3 μm in the inner cell walls of fruit capsules (Figure 1b), stem bark (Figures 1d and 1e), bud scales (Figure 1f) and roots (Figures 1g, 1h and 1i). The cells covered with Si deposits were in the external parts of the roots and the branch and stem bark (Figures 1d and 1g). Occasionally, Si was present on cell lumina (Figure 1e).

Altered leaves in the organic horizon were colonized by hyphae and amoebae (Figure 1c) and presented large voids. The Si deposits disappeared from the plant cells but were present in the observed amoebae.

##### 3.1.2 Si pools and fluxes in aboveground tree biomass

The calculated standing aboveground biomass in 2011 was significantly higher on the Eutric Cambisol than on the Rendzic Leptosol (164.2 and 115.2 t.ha$^{-1}$, respectively), and the aboveground biomass on the Dystric Cambisol is between the values for the other two soils at 125.8 t.ha$^{-1}$ (Table 2). The stem bark had the highest Si concentration in the three plots, and the Si pool in this compartment represented approximately 40% of the total Si pool in the aboveground tree biomass. The younger the structures were, the higher Si concentration. Small branches were approximately three times more concentrated than coarse branches in the three soils (Table 2). The amount of Si immobilized in the standing aboveground biomass ranged from 20.1 kg.ha$^{-1}$ on the Rendzic Leptosol to 26.2 kg.ha$^{-1}$ on the Eutric Cambisol. The annual biomass production between 2011 and 2015 was significantly higher on the Dystric Cambisol than on the Rendzic Leptosol (10.0 and 5.8 t.ha$^{-1}$.year$^{-1}$, respectively), while biomass production was between these two values for the Eutric Cambisol (8.0 t.ha$^{-1}$.year$^{-1}$). As a result, the amount of Si immobilized in the aboveground biomass each year between 2011 and 2015 ranged from 0.98 kg.ha$^{-1}$ on the Rendzic Leptosol to 1.82 kg.ha$^{-1}$ on the Dystric Cambisol.

##### 3.1.3 Si pools and fluxes in belowground tree biomass

The fine root biomass measured for the entire soil profile was calculated as 7.3 t.ha$^{-1}$ for the Dystric Cambisol (90 cm thickness), 8.7 t.ha$^{-1}$ for the Rendzic Leptosol (30 cm thickness), and 10.6 t.ha$^{-1}$ for the Eutric Cambisol (90 cm thickness) (Table 2). Regardless of the soil type, fine root biomass decreased with depth. No significant difference in fine root biomass was observed for any soil layer between the three soils. The concentrations of Si in fine roots were high in the three soils but were significantly higher in the Dystric Cambisol (between 12.3 and 15.0 g.kg$^{-1}$ of dry matter) compared to in the Rendzic Leptosol (between 4.9 and 7.8 g.kg$^{-1}$ of dry matter). The Si





pools in the fine roots were important and ranged from 67.9 kg.ha⁻¹ in the Rendzic Leptosol to 109.5 kg.ha⁻¹ in the Dystric Cambisol. Based on the turnover rate of fine roots, as determined by Brunner et al. (2013) for beech trees, i.e., $1.11 \pm 0.21$ y⁻¹, we calculated that the annual Si fluxes resulting from fine root decomposition ranged from $67.9 \pm 14.3$ kg.ha⁻¹ in the Rendzic Leptosol to $109.5 \pm 23.0$ kg.ha⁻¹ in the Dystric Cambisol.

The calculated small and coarse root biomass was 24.4 t.ha⁻¹ for the Dystric Cambisol, 26.0 t.ha⁻¹ for the Rendzic

Leptosol, and 32.3 t.ha⁻¹ for the Eutric Cambisol (90 cm), representing approximately 75% of the total root biomass in the three plots. The coarse root biomass in the Dystric Cambisol was higher than in the Eutric Cambisol and Rendzic Leptosol. The concentrations of Si in coarse roots (from 0.05 g.kg⁻¹ DM in S3 to 0.11 g.kg⁻¹ DM in S1) were two orders of magnitude lower than the concentration in fine roots. As observed for fine roots, the Si concentrations in coarse roots were higher in the Dystric Cambisol compared to the Rendzic Leptosol. The annual

immobilization of Si in coarse roots was very low for the three soils, 0.19 to 0.31 kg.ha⁻¹, and was negligible in comparison to the flux induced by fine root functioning.

### 3.1.4. Si fluxes in exploitation residues and harvests

The biomass of belowground and aboveground exploitation residues, expressed on an annual basis, ranged from 2.1 t.ha⁻¹.y⁻¹ on the Rendzic Leptosol to 2.8 t.ha⁻¹.y⁻¹ on the Eutric Cambisol (Table 2). The mean concentration of

the aboveground exploitation residues ranged from 0.24 g.kg⁻¹ DM for S3 to 0.33 g.kg⁻¹ DM for S1. The mean concentration of the belowground exploitation residues ranged from 0.06 g.kg⁻¹ DM for S3 to 0.11 g.kg⁻¹ DM for S1. The total concentration of exploitation residues represented between 0.33 and 0.50 kg.ha⁻¹.y⁻¹ of Si returning to the soil, respectively.

The biomass of the harvests ranged from 3.9 t.ha⁻¹.y⁻¹ on the Rendzic Leptosol to 4.9 t.ha⁻¹.y⁻¹ on the Eutric

Cambisol. These values represented between 0.57 and 0.72 kg.ha⁻¹ of Si exported from the ecosystem each year.

### 3.1.5 Si pool in forest floor

In 2010, the forest floor biomass ranged from 10.9 t.ha⁻¹ on the Rendzic Leptosol (eutrophic mull) to 19.0 t.ha⁻¹ on the Dystric Cambisol (acid mull). The part of small wood (residuals from the previous thinning) was higher in the Dystric Cambisol compared to the other two soil types, making up approximately 40% and 20% of the total

forest floor, respectively (Table 2). The Si pools in the forest floor ranged from 154.3 kg.ha⁻¹ on the Rendzic Leptosol to 252.9 kg.ha⁻¹ on the Dystric Cambisol. Because organic horizons have higher concentrations of Si than small woods (13.2 vs 0.8 g.kg⁻¹ DM, 8.5 vs 1.8 g.kg⁻¹ DM, 9.8 vs 1.2 g.kg⁻¹ DM for the Dystric Cambisol, Eutric Cambisol, and Rendzic Leptosol, respectively), organic horizons represented more than 95% of the Si pools in the forest floor.

### 3.1.6 Si fluxes in litterfall

The annual litterfall between 2012 and 2015 ranged from 5.2 and 6.0 t.ha⁻¹ (Table 2). No significant difference was observed between the three plots, regardless of the tree compartment. Dead leaves represented approximately 70% of the total annual litterfall, while branches and twigs represented 10%, and buds, beechnuts and fruit capsules represented 20%. Regardless of the soil type, the Si content of leaves was higher than the other litterfall

compartments, measuring 9-10 times higher than branches/twigs and 2-5 times higher than buds, beechnuts, fruit



capsules. Because of their high biomass and Si concentration compared to the other litterfall compartments, leaves were the main fraction of the Si pool (> 90%) in the litterfall in the three plots. Litter leaves collected in Dystric Cambisol were twice as concentrated in Si than litter leaves collected in Rendzic Leptosol (11.3 against 5.6 g.kg$^{-1}$), meaning that the annual Si flux from litterfall was significantly higher on the Dystric Cambisol (44.8 kg.ha$^{-1}$)

compared to the Rendzic Leptosol (25.2 kg.ha$^{-1}$).

**3.1.7 Si pool in soils and flux of dust deposits**

The total Si content and pools in the fine earth fraction were significantly lower in the Rendzic Leptosol compared to the Dystric Cambisol and to the Eutric Cambisol (Table 3). The total Si pools in the first 90 cm of soil overpassed 2 400 000 kg.ha$^{-1}$ in the Dystric Cambisol and Eutric Cambisol as opposed to approximately 720 000 kg.ha$^{-1}$ in

the Rendzic Leptosol.

The dust deposit annual flux between 2012 and 2015, collected on the flux tower of the S1 plot above the canopy, was approximately 40.5 kg.ha$^{-1}$.y$^{-1}$ with a Si content of 140 mg.kg$^{-1}$ of dry matter, representing an annual Si flux of approximately 6.0 kg.ha$^{-1}$ (Table 4).

**3.2 Si in solution: Dissolved Si**

**3.2.1 Si flux in aboveground solutions**

The mean annual Si concentration in the rainfall was very low (0.04 ± 0.08 mg.l$^{-1}$; Table 4) compared to stand deposition, representing an annual Si flux of approximately 0.2 kg.ha$^{-1}$. Consequently, the stand deposition and foliar leaching did not significantly differ between the three plots, i.e., 1.2 to 1.4 kg.ha$^{-1}$.y$^{-1}$ and 0.9 to 1.1 kg.ha$^{-1}$.y$^{-1}$ (Table 4). In the three plots, the throughfall solution was enriched in Si

(annual mean 0.14± 0.16 mg.l$^{-1}$; Table 4), and its maximum concentration (0.3 to 1 mg.l$^{-1}$) occurred in during the leafed period, especially during the senescence period (Figure 2). Although the stemflow solution was more concentrated in dissolved Si (annual mean 0.44±0.37 mg.l$^{-1}$; Table 4) than the throughfall, throughfall contributed a large amount (up to 85%) to the Si stand deposition.

**3.2.2 Si fluxes in the forest floor**

Over the study period (2012-2015), the solution collected under the forest floor was mainly enriched in Si compared to the aboveground solution (approximately one order of magnitude; Table 4). The mean annual concentration varied from 1.4 ± 0.8 mg.l$^{-1}$ in plot S3 to 1.7 ± 0.8 mg.l$^{-1}$ in plot S1. The net Si production in the forest floor was highest between September and January and was at a minimum in April, particularly in plot S3 (Figure 3). The mean annual dissolved Si production in the forest floor

ranged between 12.4 to 9.5 kg.ha$^{-1}$.y$^{-1}$ in plots S1 and S3, respectively (Table 4).

**3.2.3 Si fluxes in the soil profile**

Regardless of the soil type, the mean annual dissolved Si concentration generally increased with soil depth for both kinds of solutions, except in the deeper soil layers where the Si concentration remained constant (Figure 4a). The dissolved Si concentrations in the gravitational solution (ZTL) in the 0 to 30 cm soil layers and in the bound-

solutions (TL) in the 0-60 cm soil layers increased less than in the forest floor. Regardless of the soil type and



depth, the TL solutions were more concentrated in dissolved Si than the ZTL solutions (approximately 1.1 to 1.8 times more; Figure 4a). No matter the depth and the soil type, dissolved Si concentrations in TL solutions showed seasonal variations, with high concentrations between August and December and low concentrations between February and June, which was not the case for ZTL concentrations

(Figure 4b). The maximum concentration of dissolved Si did not depend on the drainage fluxes (data not shown).

The Si budget revealed a net annual production of dissolved Si in the 0-10 cm and 10-30 cm layers, ranging from 5.3 kg.ha$^{-1}$.y$^{1}$ in plot S1 to 14.5 kg.ha$^{-1}$.y$^{-1}$ in plot S3 and from 2.3 kg.ha$^{-1}$.y$^{-1}$ in plot S1 to 5.4 kg.ha$^{-1}$.y$^{-1}$ in plot S2, respectively (Figure 5). The production of dissolved Si drastically decreased

with the depth. In the 60-90 cm layer of plot S1, we observed the consumption of dissolved Si (Figure 5), resulting from consumption during the autumn (Figure 3). In addition, we observed high seasonal variations of the dissolved Si budget, which were more marked in the top soil layers (Figure 3). The lowest net production in these horizons was between June and August, while the maximum production rates were observed between September and February.

**3.3 Si flux taken up by trees**

By adding amounts of the Si immobilized each year in the different tree compartments, i.e., perennial aboveground biomass, leaves, bud scales, beechnuts and fruit capsules, small and coarse roots, and fine roots and the foliar leachate, we determined that the annual uptake of Si by the stand was approximately 157, 141, and 95 kg.ha$^{-1}$ in plots S1, S2, and S3, respectively.

**4 Discussion**

In the following sections, the pools and fluxes of Si in the different compartments of the ecosystem (tree stand, forest floor, and soil) are discussed, a global cycle of Si at the stand scale is proposed, and the influence of soil type on the Si cycle is assessed. Figure 6 was created to integrate and summarize the different pools and fluxes of Si in the different compartments, allowing a comprehensive view of the Si cycle at the stand scale.

**4.1 Si immobilization and internal fluxes in trees**

Perennial tissues, such as stem, branches and coarse roots, whose biomass represented more than 90% of the total tree biomass, contained between 15% (plot S1) and 20% (plot S3) of the Si accumulated in the stand. Annual tissues, such as fine roots and litterfall, contained more than approximately half (from 56% in plot S3 to 58% in plot S1 for fine roots) and a quarter (from 23% in plot S3 to 26% in plot S1 for litterfall) of the Si contained in the

stand. High Si deposition in plant tissues enhances their strength and rigidity but also improves their resistance to plant diseases by stimulating defence reaction mechanisms (Epstein, 1999; Richmond and Sussman, 2003). The high amount of Si accumulated in fine roots resulted not only from a higher Si concentration in this compartment but also from an important type of biomass. The Si content in fine roots ranged between 4.9 to 15.0 g.kg$^{-1}$, while approximately 0.1 g.kg$^{-1}$ Si content was in coarse roots. The use of a rigorous protocol of root cleaning and control

prevented soil pollution of the roots. The beech fine root biomass ranged from 7.3 to 10.6 t.ha$^{-1}$ on the Montiers site. These values correspond to the upper part of the range of 2.4 to 9.6 t.ha$^{-1}$ reported in the literature for beech




stands in Europe (Hendrik and Bianchi, 1995; Le Goff and Ottorini 2001; Schmid, 2002, Claus and George 2005; Bolte and Villanueva, 2006) and are in agreement with the fine root biomass determined for another beech forest located in the northeastern France (7.4 to 9.8 t.ha$^{-1}$; Bakker et al., 2008).

Because most of the Si accumulated in leaves and fine roots with rapid turnover (annual for leaves and estimated at 1.11±0.21 y$^{-1}$ for beech fine roots; Brunner et al., 2013), the main part of the Si taken up by trees returned to the soil each year via litterfall degradation (28%, from 25.2 kg.ha$^{-1}$ in plot S3 to 44.9 kg.ha$^{-1}$ in plot S1) and via the decomposition of fine root necromass (approximately 71%, from 67.9 kg.ha$^{-1}$ in plot S3 to 109.5 kg.ha$^{-1}$ in plot S1) (Figure 6, Table 2). As demonstrated by Sommer et al. 2003, only a small fraction (approximately 1%; from

1.0 kg.ha$^{-1}$ in plot S3 to 1.8 kg.ha$^{-1}$ in plot S1) of the Si taken up by the tree stand accumulated each year in the perennial tree compartments, i.e., the stem, branch and coarse roots (Figure 6, Table 2). As a consequence,, approximately 99% of the Si taken up by the stand each year returned to the soil via recycling of fine roots and leaves. The Si amount accumulated in the tree stand and returning to the soil (without considering the exploitation residuals) in the Montiers site ranged from 93 kg.ha$^{-1}$.y$^{-1}$ to 154 kg.ha$^{-1}$.y$^{-1}$. The Si accumulated is higher than in

other beech ecosystems previously studied, i.e., 20 kg.ha$^{-1}$.y$^{-1}$ (Cornelis et al, 2010a) and 34 kg.ha$^{-1}$.y$^{-1}$ (Sommer et al. 2013), mainly because the role of fine roots in the Si cycle was underestimated in previous studies. For example, Gérard et al. (2008), who modelled the cycle of Si in the soil of a temperate forest, estimated that the Si amount accumulated in Douglas fir roots was less than 1% of the total uptake.

## 4.2 Si residence time and budget in the forest floor

Because the amount of Si in the small wood was negligible in the three plots in comparison to the organic horizons (< 3% of the Si contained in the forest floor), only the organic horizons will be discussed below.

### 4.2.1 Soil pollution in organic horizons

Cornelis et al. (2010a) estimated that the proportion of soil with a moder humus type was approximately 40% for a deciduous temperate forest. In our study, we determined that the fraction of soil mixed in the organic horizons,

i.e., mull form, did not surpass 5%. The higher rate of soil pollution in the study of Cornelis et al. (2010a) can be explained by the presence of a thick Oh layer in the moder that was in direct contact with the superficial soil layer and was characterized by an intense mixing of degraded organic matter with soil particles, induced by biological activities. The Si pollution by dust deposits in the organic horizons was negligible, with a maximum value of 6.0 kg.ha$^{-1}$.y$^{-1}$ (no stand interception) against 151 to 246 kg.ha$^{-1}$. Lequy et al. (2014), who studied the mineralogy

of the dust deposits of the Montiers site, observed that the Si in the litterfall was mainly quartz.

### 4.2.2 Si residence time in organic horizons

The main phytogenic Si input into the organic horizons was opal phytoliths (Krieger et al.2017), which dissolve slowly (Fraysse et al., 2009) in comparison to the rate of organic matter mineralization. The residence time of Si in the organic horizons is higher than carbon (5.3 ± 0.8 vs 1.9 ± 0.4 y). In addition, the presence of testate amoebae,

organisms with a skeleton that is rich in Si (Figure 1; Sommer et al., 2013), in the organic horizons suggests that a part of the Si from the phytoliths belonged to the zoogenic Si pool. Sommer et al. (2013) estimated that the




zoogenic pool could represent half of the Si input by litterfall (17 kg.ha[-1] vs 34 kg.ha[-1]) in beech organic horizons in Europe.

### 4.2.3 Si budget in organic horizons

During the study period (2012-2015), the Si input in the organic horizons via litterfall were primarily higher than the Si output via soluble transport (assessed in ZTL solutions under the forest floor) for the three soils. This observation suggests an accumulation of Si in the organic horizons during the study (which may double the amount of Si in the organic horizons in eight years) and/or the existence of another output flux not quantified in our study. If the accumulation of Si occurs in the organic horizons, it is likely limited on our site compared to the loss of Si

in the organic horizons, induced by solid particulate migration toward the topsoil layer. In our study, the solid particulate migration from the organic horizons to the topsoil may consist of the transport of amoebae or the sedimentation of phytoliths or testate amoebae, which are denser than organic matter.

### 4.3 Si budget and origin in soil

The Si production (source) in the soil mainly results from the dissolution of soil minerals and amorphous Si as
well as from plant tissues and testate amoebae (Cornelis et al., 2011; Sommer et al., 2013). The immobilization (sink) of dissolved Si in the soil is due to plant and organism immobilization and precipitation of secondary minerals, such as phyllosilicates or Si-amorphous (Dahlgren and Ugolini, 1989; Ma and Yamaji, 2006; Sommer et al., 2013; Tubana et al., 2016; Kabata-Pendias and Mukherjee, 2007).

A net production of dissolved Si in the soils was observed on the three studied plots until a depth of 60 cm, showing
a positive production/immobilization budget. The net production of Si in the soil, ranging from 7.0 to 16.7 kg.ha[-1].y[-1], was mainly located in the 0-10 cm layer, which probably accumulated amorphous Si from organic horizons that contained a large portion of fine roots from the soil. This is corroborated by the strong relationship between annual Si production in the 10-60 cm soil layers and fine root content. The contribution of fine roots to the production of dissolved Si was higher in the superficial layer and decreased in the deep soil layers, where testate
amoebae and phytoliths accumulated after being transferred from the organic horizons. A peak of net Si production was observed during fall (Figure 3), which was probably due to an increase in Si production through the decomposition of dead roots. This finding is consistent with the studies of Meier and Leuschne (2008) and Konopka (2009), who demonstrated that fine root necromass is highest at the end of the summer, when the soil is the driest, favouring root mortality. At our site, this period was also characterized by a maximum concentration of
Si in the bound waters and a negative budget in the 10-cm and 60- cm soil layers, resulting from the precipitation of secondary minerals, likely of biogenic origin. As a result, a drastic decrease of Si production was observed in the surface layer during the vegetation period, where Si uptake by plant occurred (Figure 3). In the deeper layer, the dissolved Si budget was significantly negative and likely corresponded to mineral precipitation, induced by a decrease of Si drainage with the depth, as observed by Sommer et al. (2013).

The Si produced in the soils was mainly leached out of the soil profile by drainage during winter. This drainage flux represented only a small fraction of the Si taken up by trees, i.e., 4%, 10% and 18% in plots S1, S2 and S3, respectively. In addition, our data suggest that the Si produced in the soil mainly originated from the biological cycle. As already demonstrated in other studies in temperate forests (Bartoli, 1983; Watteau and Villemin, 2001;





Gerard et al, 2008; Cornelis et al., 2010a; Cornelis et al., 2011a; Sommer et al., 2006; Sommer et al., 2013), very few of the Si leached into the soil profile directly results from the dissolution of soil minerals.

**4.4 Si cycle at stand scale**

Silicon inputs and outputs have minor contributions to the global Si budget in forest ecosystems, and the Si cycle is mainly driven by internal fluxes, especially recycling of biogenic Si.

As explained above, the main part of the Si taken up by trees was allocated to annual compartments, i.e., 28% to
leaves, buds, beechnuts and fruit capsules and 71% to fine roots (Figure 6). Only 1% of the Si taken up by trees was allocated to perennial tissues, i.e., stem and branches, coarse roots (Figure 6). In addition, about half of the Si accumulated in the perennial tree compartments returned each year to the soil via branch falls and exploitation residues (< 7 cm diameter branches left on the floor and small/coarse roots left in the soil) and approximately 40% was exported out of the site (stem and > 7 cm diameter branches). As a result, the amount of Si immobilized in
trees remained almost constant over time at the stand scale (mean Si immobilization for the three plots, 0.1 kg.ha$^{-1}$.y$^{-1}$).

In the organic horizons and in the soil, mainly in the 0-10 cm layer, we observed a high net Si production, likely resulting from the decomposition of litter leaves and amoebae in the organic horizons and of fine roots in the soil (Figure 6). The seasonal dynamics of net Si production during the year suggest a relationship between biological
activities and Si production, i.e., high net Si production at the end of the summer is linked to fine root decomposition and lower net Si production during spring/summer is induced by tree uptake. Net Si production decreased with depth, and a consumption of Si was observed in the deeper soil horizon in plot S1 (Figure 6). This likely resulted from both a decrease in Si production (less root and clay) and the precipitation of Si through the formation of secondary minerals, resulting from reduced drainage flux.
The assessment of Si fluxes and pools in the different compartments of our forested site coupled with a seasonal dynamic follow-up reveal a rapid and almost total recycling of Si in our site and show the strong influence of biological partners, mainly fine roots, and processes in the Si cycle.

**4.5 Soil influence in Si cycle**

In this study, we took advantage of a natural soil gradient, from shallow calcic soils to deep acidic soils all with
similar climates, atmospheric depositions, species composition, and management, to assess the influence of soil type on the Si cycle in a beech temperate forest ecosystem. We hypothesized that soil characteristics, including soil depth (> 200 cm for plot S1 to < 30 cm for plot S3) and a large range of Si content (> 3.10$^3$ t.ha$^{-1}$ for plot S1 to 0.7.10$^3$ t.ha$^{-1}$ for plot S3), and their influence on stand growth and functioning could influence the Si pools and fluxes between the different compartments of the ecosystem.
We showed that the Si content of plant compartments (leaves, organic horizons, aboveground and belowground biomasses) were higher in the Si rich soils (plots S1 and S2) compared to plot S3. This is in agreement with the observations of Heineman et al. (2016) in tropical forests, which demonstrated that nutrient concentrations in wood and leaves correlated positively with soil Ca, K, Mg and P concentrations. Moreover, the annual tree compartments (leaves and fine roots) were more concentrated than the perennial compartments (branches, stem and coarse roots).
The variations of Si content in the annual tree compartments induced by the soil type significantly affected the Si fluxes in the ecosystem. The annual uptake and Si recycling (leaves + buds, beechnuts, fruit capsules + fine roots)



were 127.2 and 154.0 kg.ha$^{-1}$, respectively, in plot S1, as opposed to 94.8 and 92.7 kg.ha$^{-1}$, respectively, in plot S3.

In return, the bound solutions were more concentrated in plot S3 compared to plot S1. This is partly due to the higher clay content in plot S3 compared to plot S1 (clay was two times higher in plot S3). This considerably increases the specific surface area of minerals and improves their weatherability and water retention capacity (Carroll and Starkey, 1971; De Jonge et al., 1996).

**5 Conclusion**

By coupling different approaches (annual budget in solid and solution phases and monthly dynamics of solutions) and methods (direct *in situ* measurements and standard and site specific modelling) to quantify Si pools and fluxes in the different ecosystem compartments, our study allowed us to assess the complete Si cycle at the forest stand scale. Interestingly, our study highlights the main contribution of fine roots and, to a lesser extent, of leaves in the Si cycle. Almost all the dissolved Si was taken up by trees at any given time (very weak leaching out of the soil profile) and was recycled each year (approximately 99%, only 1% immobilized in perennial tissues). This suggests that Si cycle is almost closed during the vegetation period; dissolved Si is taken up by vegetation then Si returned to the soil mainly through root and leave decomposition to give dissolved Si, which is again taken up by vegetation. This observation is consistent with the observation of Sommer et al. (2013), who demonstrated a low contribution of geochemical weathering processes to the Si cycle in a forest biogeosystem. The seasonal dynamics of dissolved Si confirmed the key role of biological processes in the Si cycle, notably through the production of dissolved Si during the decomposition of fine roots. Our study also revealed that soil type influences the Si cycle at different levels. The plant compartments were Si-enriched in the soil with higher Si, i.e., Dystric Cambisol (plot S1) compared to plant compartments in the Rendzic Leptosol (plot S3), resulting in 1.6-times higher recycling in plot S1 compared to plot S3. While Si production was relatively similar in the organic horizons for the three plots, its production in the soil, mainly in the 0-10 cm layer, was twice higher in plot S3 and richer in clays than plot S1. Further research is needed in the mid-term to improve our understanding of processes governing the fluxes of Si in forest ecosystems. Notably, it will be interesting to quantify the flux of Si resulting from the migration of solid particulates from the forest floor to the topsoil as well as the contribution of clay dissolution in Si production in the soil profile.

**Acknowledgement**

We acknowledge S. Didier for site implementation and management, L. Franoux and A. Genêt for the development of allometric equations, C. Pantigny, L. Gelhaye, B. Simon, C. Nys, J. Mangin, C. Goldstein, F. César, M. D'Arbaumont and M. Simon for technical help, Salsi, L. for preparing the samples and performing the SEM and EDX analyses, and the ONF officers for the management of the Montiers forest. We would like to thank the National Forest Office (ONF) for welcoming us into the domanial forest of Montiers and for the management of the forest. The authors acknowledge the facilities of the French National Institute for Agricultural Research and the Service d'Analyse des Roches et des Minéraux of the French National Center for Scientific Research. This work was supported by the Andra and INRA (accord spécifique N°9) and GIP-Ecofor (contract N°1138451B).



The Montiers site belongs to the SOERE F-ORE-T (http://www.gip-ecofor.org/f-ore-t/index.php), AnaEE France
(http://www.anaee-s.fr/) and ICOS (www.icosinfrastructure.eu) networks.

The authors declare that they have no conflict of interest.

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





**Figure caption**

**Fig. 1:** Si in biological tissues of beech trees observed through Scanning Electron Microscopy. (a) Si precipitates in the intercellular space of fresh leaves, forming phytoliths (white arrow). Deposits of Si (white arrows) in the

inner cell walls of fruit capsules (b), stem bark (d and e), bud scales (f), and roots (g, h, and i). (c) Hyphae, amoebae and large voids in altered litter leaves. Si deposits only present in the testate amoebae (white arrows).

**Fig. 2:** Seasonal dynamics on four years (January 2012 to December 2015) of dissolved Si concentration in throughfall solution for the three plots S1, S2, and S3.

**Fig. 3:** Seasonal dynamics over four years (January 2012 to December 2015) of the dissolved Si budget in the

different layers (forest floor: FF; soil 0-10 cm: L0-10; soil 10-30 cm: L10-30; soil 30-60 cm: L30-60; and soil 60-90 cm: L60-90) for the three plots S1, S2, and S3.

**Fig. 4:** a. Mean dissolved Si concentration over four years (January 2012 to December 2015) in a zero-tension lysimeter (ZTL) and tension lysimeter (TL) with soil solutions at different depths (0-10 cm, 10-30, 30-60, and 60-90 cm) in plots S1 and S3. B. Seasonal dynamics over four years (January 2012 to December 2015) of Si

concentrations in ZTL and TL soil (TL) in the layers 0-10 cm (L0-10) and 10-30 cm (L10-30) of plot S3.

**Fig. 5:** Mean annual dissolved Si budget in the different layers of the forest floor, FF; soil 0-10 cm: L0-10; soil 10-30 cm: L10-30; soil 30-60 cm: L30-60; and soil 60-90 cm: L60-90) for the three plots S1, S2, and S3. Bars represent the standard deviations. Positive and negative values represent the production or consumption of dissolved Si in the given layer. Histograms with an asterisk are significantly different from 0, according to a

Kruskal-Wallis test at the threshold P value level of 0.05.

**Fig. 6:** Summary scheme of Si cycling on the plots S1, S2 and S3 of our study forest site, including (i) pools of Si in the biomass, (ii) internal fluxes, i.e., in the soil-plant system, (iii) external fluxes entering or leaving the soil-plant system, and (iv) the dissolved Si budget in the different layers of the ecosystem. Pools are presented by rectangular boxes (tree annual and perennial parts, organic horizons and small dead wood, and soil). Internal fluxes

(solid form from the tree to the soil, i.e., fine roots, litterfall including leaves, buds and branches, and exploitation residues; and in solution from the soil to the plant, i.e., the tree uptake) are presented in boxes with rounded edges. Grey/black arrows indicate the direction and the intensity of the internal fluxes. The external fluxes (inputs: rainfall and dust deposits, and outputs: drainage and biomass harvest) are presented in flag boxes. For each pool and flux, values presented are those of the plots S1, S2, and S3, respectively. The dissolved Si budget in the different layers

(forest floor and different soil horizons) are represented with white arrows, which indicate the direction and the intensity of the fluxes. Arrows leaving the layer indicate the production of dissolved Si in this layer. In contrast, arrows entering the layer indicate the consumption of dissolved Si in this layer. Values presented in each box and arrow are annual mean values for plots S1, S2, and S3, respectively (except for atmosphere values which are similar for the three plots). The AG and BG correspond to aboveground and belowground tree compartments.


**Table 1:** Physicochemical properties of the three studied soils in the Montiers site (plot S1 – Dystric Cambisol; plot S2 – Eutric Cambisol; plot S3 – Rendzic Leptosol). Are presented the mean values for bulk density (g.cm3), textural distribution (g kg$^{-1}$), total rock volume (RV), soil water holding capacity (SWHC), soil water pH, organic matter content (OM), cation exchange capacity (CEC; cmol+ kg$^{-1}$) and base-cation saturation ratio (S/CEC, with S = sum of base cations). Standard deviation values are given in italic. Table adapted from Kirchen et al. (2017).

| | Depth (cm) | B. density (g cm$^{-3}$) | Clay (g kg$^{-1}$) | F. silt | C. silt | F. sand | C. sand | RV (%) | SWHC (mm) | pH$_{water}$ | OM (g kg$^{-1}$) | CEC (cmol+ kg$^{-1}$) | S/CEC (%) |
|---|---|---|---|---|---|---|---|---|---|---|---|---|---|
| S1 Dystric Cambisol | 0-5 | 0.98 *0.12* | 255 *25* | 281 *24* | 160 *17* | 185 *36* | 121 *19* | 1.4 | 8.2 | 4.9 | 68 *22* | 6.7 *3.0* | 64 *23* |
| | 5-15 | 0.94 *0.17* | 245 *26* | 276 *29* | 162 *17* | 184 *40* | 131 *24* | 1.4 | 16.5 | 4.8 | 43 *16* | 4.2 *2.2* | 35 *21* |
| | 15-30 | 1.23 *0.22* | 268 *28* | 280 *31* | 161 *21* | 170 *44* | 115 *31* | 1.8 | 22.7 | 4.8 | 26 *9* | 3.5 *0.9* | 26 *14* |
| | 30-45 | 1.36 *0.18* | 306 *65* | 262 *45* | 150 *27* | 161 *47* | 119 *32* | 2.3 | 22.6 | 4.9 | 15 *5* | 4.3 *1.6* | 36 *16* |
| | 45-60 | 1.45 *0.15* | 355 *100* | 229 *45* | 126 *31* | 166 *49* | 141 *39* | 3.6 | 18.1 | 5.1 | 10 *2* | 5.7 *2.6* | 55 *22* |
| S2 Eutric Cambisol | 0-5 | 1.03 *0.11* | 242 *52* | 242 *16* | 143 *13* | 290 *36* | 83 *24* | 2.3 | 9.2 | 5.4 | 73 *26* | 10.1 *5.4* | 83 *14* |
| | 5-15 | 0.93 *0.13* | 241 *65* | 246 *17* | 145 *13* | 287 *45* | 82 *24* | 3.1 | 18.2 | 5.2 | 45 *29* | 7.8 *7.3* | 59 *24* |
| | 15-30 | 1.23 *0.19* | 294 *83* | 234 *23* | 136 *17* | 273 *55* | 64 *11* | 7.6 | 19.1 | 5.3 | 27 *13* | 7.7 *3.9* | 61 *23* |
| | 30-45 | 1.35 *0.18* | 420 *141* | 188 *43* | 107 *31* | 214 *63* | 71 *20* | 29.0 | 14.7 | 5.3 | 17 *8* | 13.2 *6.9* | 68 *27* |
| | 45-60 | 1.32 *0.23* | 523 *136* | 154 *42* | 85 *32* | 176 *57* | 63 *31* | 40.3 | 10.3 | 5.4 | 11 *4* | 17.8 *8.8* | 76 *17* |
| S3 Rendzic Leptosol | 0-5 | 0.88 *0.14* | 449 *80* | 227 *54* | 123 *26* | 119 *39* | 41 *15* | 2.3 | 9.8 | 5.7 | 109 *27* | 24.9 *8.3* | 98 *5* |
| | 5-15 | 0.98 *0.12* | 430 *82* | 224 *56* | 114 *36* | 123 *37* | 59 *21* | 4.9 | 19.2 | 5.7 | 71 *23* | 20.0 *7.9* | 94 *7* |
| | 15-30 | 1.06 *0.22* | 516 *81* | 169 *50* | 77 *38* | 102 *42* | 63 *24* | 36.4 | 12.5 | 6.0 | 42 *10* | 23.2 *6.4* | 99 *5* |





**Table 2:** Mean Si contents, pools and fluxes in the biomass of the three soils of the Montiers site. Standard deviation values are given in brackets. Values with different letters are significantly different according to a Kruskal-Wallis test at the threshold P value level of 0.05 (soil effect).


| Plot | Compartment | Biomass pools (t DM ha⁻¹) | Biomass increment (t DM ha⁻¹ yr⁻¹) | Si content (g kg⁻¹) | Si pools (kg ha⁻¹) | Si fluxes (kg ha⁻¹ yr⁻¹) |
|---|---|---|---|---|---|---|
| | Leaves | 3.8 (0.4) ᵃ | 3.8 (0.4) ᵃ | 11.3 (1.8) ᵇ | 42.7 (4.3) ᵇ | 42.7 (4.3) ᵇ |
| | Branches/twigs with bark | 0.3 (0.2) ᵃ | 0.3 (0.2) ᵃ | 1.1 (0.3) ᵃ | 0.3 (0.2) ᵃ | 0.3 (0.2) ᵃ |
| | Buds, beechnuts, fruit capsules | 1.1 (1.1) ᵃ | 1.1 (1.1) ᵃ | 2.4 (1.0) ᵃ | 1.8 (0.9) ᵃ | 1.8 (0.9) ᵃ |
| | **Total litterfall** | 5.2 (1.1) ᵃ | 5.2 (1.1) ᵃ | | 44.8 (5.1) ᵇ | 44.8 (5.1) ᵇ |
| | Organic horizons | 11.5 (2.0) ᵃ | | 21.4 (1.6) ᵃ | 246.4 (53.1) ᵃ | |
| | Small wood | 7.5 (1.9) ᵃ | | 0.8 (0.3) ᵃ | 6.5 (3.5) ᵃ | |
| | **Forest floor** | 19.0 (2.7) ᵃ | | | 252.9 (53.1) ᵃ | |
| | Stem bark | 5.5 (0.7) ᵃ | 0.5 (0.0) ᵇ | 1.70 (0.33) ᵃ | 9.4 (1.2) ᵃ | 0.65 (0.03) ᵇ |
| | Stem wood | 84.8 (11.7) ᵃᵇ | 6.4 (0.3) ᵇ | 0.05 (0.00) ᵃ | 4.0 (0.5) ᵃ | 0.30 (0.02) ᵃ |
| S1 : | Small branches (B+W) | 18.7 (2.5) ᵃᵇ | 1.2 (0.1) ᵇ | 0.40 (0.05) ᵃ | 7.4 (1.0) ᵃ | 0.49 (0.03) ᵇ |
| Dystric | Medium branches (B+W) | 10.2 (1.8) ᵃᵇ | 1.1 (0.1) ᵇ | 0.26 (0.04) ᵃ | 2.6 (0.5) ᵃᵇ | 0.29 (0.02) ᵇ |
| Cambisol | Coarse branches (B+W) | 5.1 (1.1) ᵃᵇ | 0.8 (0.1) ᵇ | 0.13 (0.04) ᵃ | 0.7 (0.1) ᵃᵇ | 0.10 (0.01) ᵇ |
| | **Aboveground biomass** | 125.8 (17.9) ᵃᵇ | 10.0 (0.3) ᵇ | | 24.1 (3.3) ᵃᵇ | 1.82 (0.10) ᵇ |
| | Fine roots (0-10 cm) | 3.2 (0.8) ᵃ | 3.5 (0.9) ᵃ | 12.8 (2.3) ᵃ | 39.5 (7.5) ᵃ | 43.9 (8.3) ᵃ |
| | Fine roots (10-30 cm) | 2.9 (1.1) ᵃ | 3.2 (1.2) ᵃ | 15.0 (2.3) ᶜ | 43.9 (6.6) ᵇ | 48.8 (7.3) ᵇ |
| | Fine roots (30-60 cm) | 0.9 (0.6) ᵃ | 1.0 (0.7) ᵃ | 12.3 | 10.5 | 11.7 |
| | Fine roots (60-90 cm) | 0.4 (0.1) ᵃ | 0.4 (0.1) ᵃ | 12.7 | 4.7 | 5.2 |
| | **Total fine roots (0-90 cm)** | 7.3 (1.8) ᵃ | 8.0 (2.0) | | 98.7 (13.5) ᵇ | 109.5 (15.0) ᵇ |
| | **Total coarse roots** | 24.4 (3.5) ᵃ | 2.83 (0.47) ᵃ | 0.11 (0.15) ᵃ | 2.66 (0.39) ᵇ | 0.31 (0.05) ᵇ |
| | Exploitation residues AG | | 1.3 | 0.33 | | 0.42 |
| | Exploitation residues BG | | 1.1 | 0.11 (0.15) ᵃ | | 0.12 |
| | **Total exploitation residues** | | 2.4 | | | 0.54 |
| | **Harvests** | | 4.4 | 0.16 | | 0.71 |
| | Leaves | 4.1 (0.5) ᵃ | 4.1 (0.5) ᵃ | 8.9 (1.6) ᵃᵇ | 35.4 (2.8) ᵃᵇ | 35.4 (2.8) ᵃᵇ |
| | Branches/twigs with bark | 0.6 (0.4) ᵃ | 0.6 (0.4) ᵃ | 0.9 (0.2) ᵃ | 0.4 (0.2) ᵃ | 0.4 (0.2) ᵃ |
| | Buds, beechnuts, fruit capsules | 1.3 (1.1) ᵃ | 1.3 (1.1) ᵃ | 3.4 (1.9) ᵃ | 3.0 (0.5) ᵇ | 3.0 (0.5) ᵇ |
| | **Total litterfall** | 6.0 (1.1) ᵃ | 6.0 (1.1) ᵃ | | 38.7 (3.1) ᵃᵇ | 38.7 (3.1) ᵃᵇ |
| | Organic horizons | 9.6 (1.4) ᵃ | | 17.6 (0.8) ᵃ | 174.2 (32.8) ᵃᵇ | |
| | Small wood | 2.6 (1.2) ᵃ | | 1.8 (1.1) ᵃ | 3.9 (1.3) ᵃ | |
| | **Forest floor** | 12.5 (0.6) ᵃ | | | 178.1 (32.6) ᵃᵇ | |
| | Stem bark | 6.1 (0.2) ᵃ | 0.4 (0.0) ᵃᵇ | 1.53 (0.28) ᵃ | 9.3 (0.3) ᵃ | 0.39 (0.04) ᵃ |
| | Stem wood | 109.9 (3.8) ᵇ | 5.0 (0.6) ᵃᵇ | 0.05 (0.00) ᵃ | 5.1 (0.2) ᵃ | 0.23 (0.02) ᵃ |
| S2 : | Small branches (B+W) | 20.8 (0.7) ᵇ | 0.8 (0.1) ᵃᵇ | 0.38 (0.08) ᵃ | 7.9 (0.3) ᵃ | 0.31 (0.04) ᵃᵇ |
| Eutric | Medium branches (B+W) | 15.2 (0.6) ᵇ | 1.0 (0.1) ᵃᵇ | 0.23 (0.05) ᵃ | 3.5 (0.1) ᵇ | 0.23 (0.02) ᵃᵇ |
| Cambisol | Coarse branches (B+W) | 9.8 (0.6) ᵇ | 0.9 (0.1) ᵇ | 0.10 (0.03) ᵃ | 1.0 (0.1) ᵃ | 0.09 (0.01) ᵃᵇ |
| | **Aboveground biomass** | 164.2 (5.7) ᵇ | 8.0 (0.9) ᵃᵇ | | 26.9 (0.9) ᵇ | 1.25 (0.13) ᵃᵇ |
| | Fine roots (0-10 cm) | 4.6 (2.1) ᵃ | 5.1 (2.4) ᵃ | 9.6 (2.9) ᵃᵇ | 44.5 (13.9) ᵃ | 49.4 (15.4) ᵃ |
| | Fine roots (10-30 cm) | 4.5 (1.8) ᵃ | 5.0 (1.9) ᵃ | 8.2 (1.6) ᵇ | 37.0 (7.1) ᵇ | 41.1 (7.8) ᵇ |
| | Fine roots (30-60 cm) | 1.2 (0.7) ᵃ | 1.3 (0.8) ᵃ | 7.5 | 8.7 | 9.7 |
| | Fine roots (60-90 cm) | 0.4 (0.1) ᵃ | 0.5 (0.1) ᵃ | - | - | - |
| | **Total fine roots (0-90 cm)** | 10.6 (4.1) ᵃ | 11.7 (4.5) | | 90.2 (20.8) ᵇ | 100.1 (23.1) ᵇ |
| | **Total coarse roots** | 32.3 (1.2) ᵇ | 4.08 (0.16) ᵇ | 0.05 (0.08) ᵃ | 1.51 (0.05) ᵃ | 0.19 (0.01) ᵃ |
| | Exploitation residues AG | | 1.4 | 0.31 | | 0.43 |
| | Exploitation residues BG | | 1.4 | 0.05 (0.08) ᵃ | | 0.06 |
| | **Total exploitation residues** | | 2.8 | | | 0.50 |
| | **Harvests** | | 4.9 | 0.15 | | 0.72 |
| | Leaves | 4.0 (0.4) ᵃ | 4.0 (0.4) ᵃ | 5.6 (1.3) ᵃ | 22.2 (3.1) ᵃ | 22.2 (3.1) ᵃ |
| | Branches/twigs with bark | 0.5 (0.3) ᵃ | 0.5 (0.3) ᵃ | 0.7 (0.1) ᵃ | 0.3 (0.2) ᵃ | 0.3 (0.2) ᵃ |
| | Buds, beechnuts, fruit capsules | 1.2 (0.9) ᵃ | 1.2 (0.9) ᵃ | 3.2 (1.6) ᵃ | 2.6 (0.5) ᵃᵇ | 2.6 (0.5) ᵃᵇ |
| | **Total litterfall** | 5.7 (1.0) ᵃ | 5.7 (1.0) ᵃ | | 25.2 (3.4) ᵃ | 25.2 (3.4) ᵃ |
| S3 : | Organic horizons | 8.8 (1.5) ᵃ | | 16.9 (1.4) ᵃ | 151.3 (22.6) ᵇ | |
| Rendzic | Small wood | 1.9 (2.4) ᵃ | | 1.3 (0.7) ᵃ | 4.4 (5.7) ᵃ | |
| Leptosol | **Forest floor** | 10.9 (2.8) ᵃ | | | 154.3 (25.3) ᵃ | |
| | Stem bark | 6.8 (0.6) ᵃ | 0.3 (0.0) ᵃ | 1.34 (0.27) ᵃ | 9.1 (0.8) ᵃ | 0.41 (0.05) ᵃᵇ |
| | Stem wood | 80.1 (8.3) ᵃ | 3.9 (0.5) ᵃ | 0.06 (0.00) ᵃ | 5.0 (0.5) ᵃ | 0.24 (0.03) ᵃ |
| | Small branches (B+W) | 15.0 (1.4) ᵃ | 0.6 (0.1) ᵃ | 0.29 (0.04) ᵃ | 4.3 (0.4) ᵃ | 0.18 (0.02) ᵃ |
| | Medium branches (B+W) | 8.6 (1.4) ᵃ | 0.6 (0.1) ᵃ | 0.19 (0.04) ᵃ | 1.6 (0.3) ᵃ | 0.11 (0.02) ᵃ |
| | Coarse branches (B+W) | 4.6 (1.0) ᵃ | 0.4 (0.1) ᵃ | 0.10 (0.03) ᵃ | 0.5 (0.1) ᵃ | 0.04 (0.01) ᵃ |



| | | | | | |
|---|---|---|---|---|---|
| **Aboveground biomass** | 115.2 (12.8)[a] | 5.8 (0.8)[a] | | 20.5 (2.1)[a] | 0.98 (0.13)[a] |
| Fine roots (0-10 cm) | 5.1 (1.4)[a] | 5.6 (1.6)[a] | 7.8 (2.2)[a] | 43.5 (14.1)[a] | 48.3 (15.6)[a] |
| Fine roots (10-30 cm) | 3.6 (1.6)[a] | 4.0 (1.8)[a] | 4.9 (0.8)[a] | 17.6 (3.0)[a] | 19.6 (3.3)[a] |
| Fine roots (30-60 cm) | NS | NS | - | - | - |
| Fine roots (60-90 cm) | NS | NS | - | - | - |
| **Total fine roots** (0-30 cm) | 8.7 (3.0)[a] | 9.6 (3.3) | | 61.2 (16.0)[a] | 67.9 (17.7)[a] |
| **Total coarse roots** | 26.0 (3.0)[a] | 3.09 (0.44)[a] | 0.06 (0.05)[a] | 1.62 (0.19)[a] | 0.19 (0.03)[a] |
| Exploitation residues AG | | 1.1 | 0.24 | | 0.27 |
| Exploitation residues BG | | 1.0 | 0.06 (0.05)[a] | | 0.06 |
| **Total exploitation residues** | | 2.1 | | | 0.33 |
| **Harvests** | | 3.9 | 0.15 | | 0.57 |



**Table 3:** Mean total Si content and pool in the fine earth fraction of the three soils of the Montiers site at different depths. Standard deviation values are given in brackets. Values with different letters are significantly different according to a Kruskal-Wallis test at the threshold P value level of 0.05 (soil effect).


| Soil type | Compartment | Total Si content (g kg$^{-1}$) | Total Si pool (t ha$^{-1}$) |
|---|---|---|---|
| S1 : Dystric Cambisol | 0-10 cm | 305 (13) | 297 (33) |
| | 10-30 cm | 313 (9) | 708 (50) |
| | 30-60 cm | 296 (18) | 1 301 (422) |
| | 60-90 cm | 230 (28) | 858 (80) |
| | **Total** 0-90 cm | | 3 164 (487) [b] |
| S2 : Eutric Cambisol | 0-10 cm | 361 (11) | 411 (30) |
| | 10-30 cm | 360 (13) | 791 (127) |
| | 30-60 cm | 295 (62) | 871 (290) |
| | 60-90 cm | 224 (28) | 348 (117) |
| | **Total** 0-90 cm | | 2 421 (410) [b] |
| S3 : Rendzic Leptosol | 0-10 cm | 287 (27) | 233 (18) |
| | 10-30 cm | 276 (23) | 427 (27) |
| | 30-60 cm | 175 (37) | 42 (27) |
| | 60-90 cm | 144 (39) | 27 (8) |
| | **Total** 0-90 cm | | 720 (38) [a] |





**Table 4:** Si content and fluxes in the ZTL (Zero Tension Lysimeters) and TL (Tension Lysimeters) solutions of the three soils of the Montiers site. Standard deviation values are given in brackets. Values with different letters are significantly different according to a Kruskal-Wallis test at the threshold P value level of 0.05 (soil effect).


| Plot | Level | $Si_{ZTL}$ content (mg l$^{-1}$) | $Si_{TL}$ content (mg l$^{-1}$) | Si fluxes (kg ha$^{-1}$ y$^{-1}$) |
|---|---|---|---|---|
| S1 : Dystric Cambisol | Rainfall | 0.04 (0.08) | | 0.2 (0.1) |
| | Throughfall | 0.15 (0.18) | | 1.2 (0.6) |
| | Stemflow | 0.38 (0.32) | | 0.1 (0.5) |
| | Stand deposition | | | 1.3 (0.3) |
| | Forest floor | 1.7 (0.8) | | 13.7 (2.7) |
| | L-10 cm | 2.0 (0.7) | 2.9 (1.0) | 19.0 (5.6) |
| | L-30 cm | 2.6 (0.4) | 3.5 (1.1) | 21.4 (8.3) |
| | L-60 cm | 2.6 (0.5) | 4.1 (1.4) | 22.4 (9.8) |
| | L-90 cm | 2.5 (0.3) | 3.7 (0.6) | 20.7 (7.4) |
| S2 : Eutric Cambisol | Rainfall | 0.04 (0.08) | | 0.2 (0.1) |
| | Throughfall | 0.16 (0.16) | | 1.2 (0.6) |
| | Stemflow | 0.53 (0.38) | | 0.2 (0.6) |
| | Stand deposition | | | 1.4 (0.6) |
| | Forest floor | 1.5 (0.6) | | 12.6 (4.2) |
| | L-10 cm | 2.1 (0.7) | 3.2 (1.1) | 21.6 (4.8) |
| | L-30 cm | 3.5 (1.6) | 4.0 (1.1) | 25.5 (5.9) |
| | L-60 cm | 2.8 (0.6) | 4.5 (1.1) | 26.2 (6.6) |
| S3 : Rendzic Leptosol | Rainfall | 0.04 (0.08) | | 0.2 (0.1) |
| | Throughfall | 0.13 (0.14) | | 1.0 (0.5) |
| | Stemflow | 0.42 (0.41) | | 0.1 (0.4) |
| | Stand deposition | | | 1.2 (0.5) |
| | Forest floor | 1.4 (0.8) | | 10.7 (1.4) |
| | L-10 cm | 2.1 (1.1) | 3.8 (1.2) | 25.2 (9.9) |
| | L-30 cm | 2.3 (1.0) | 4.2 (1.2) | 27.4 (9.0) |



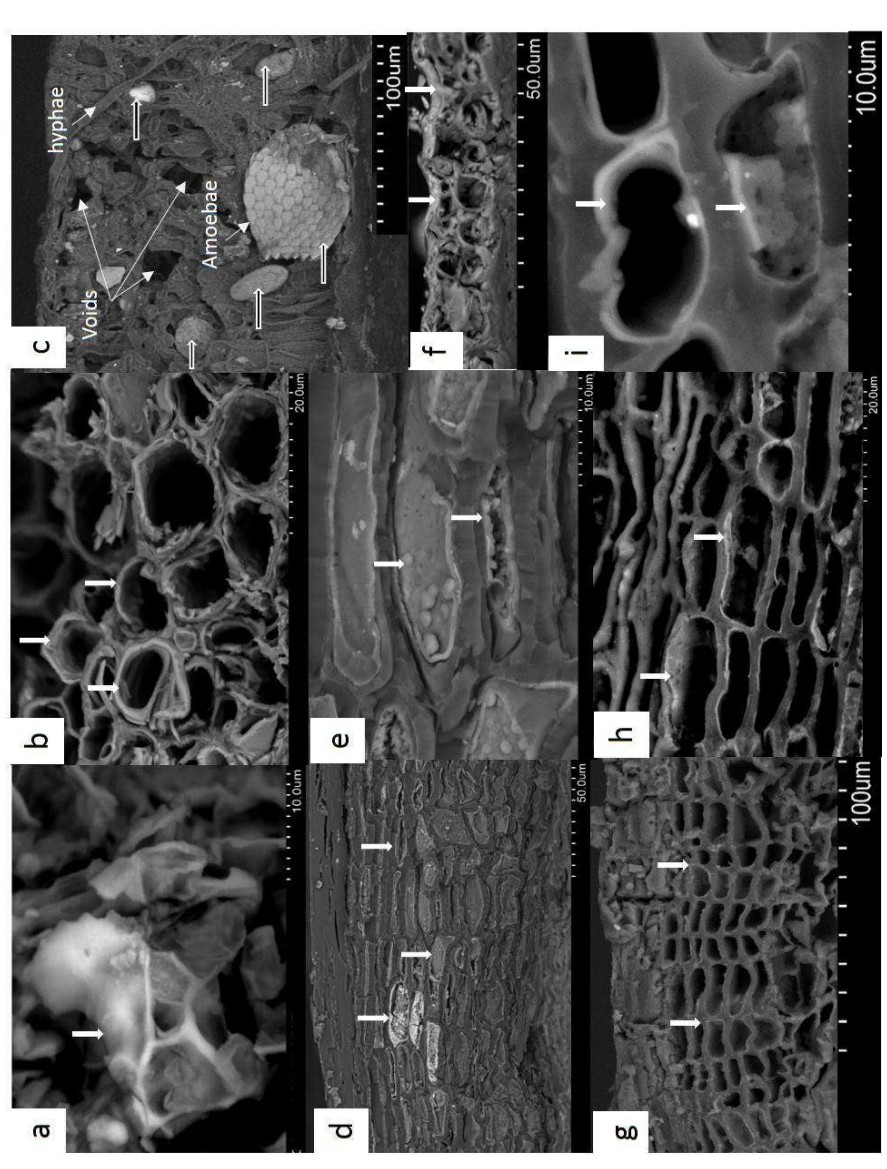

Figure 1



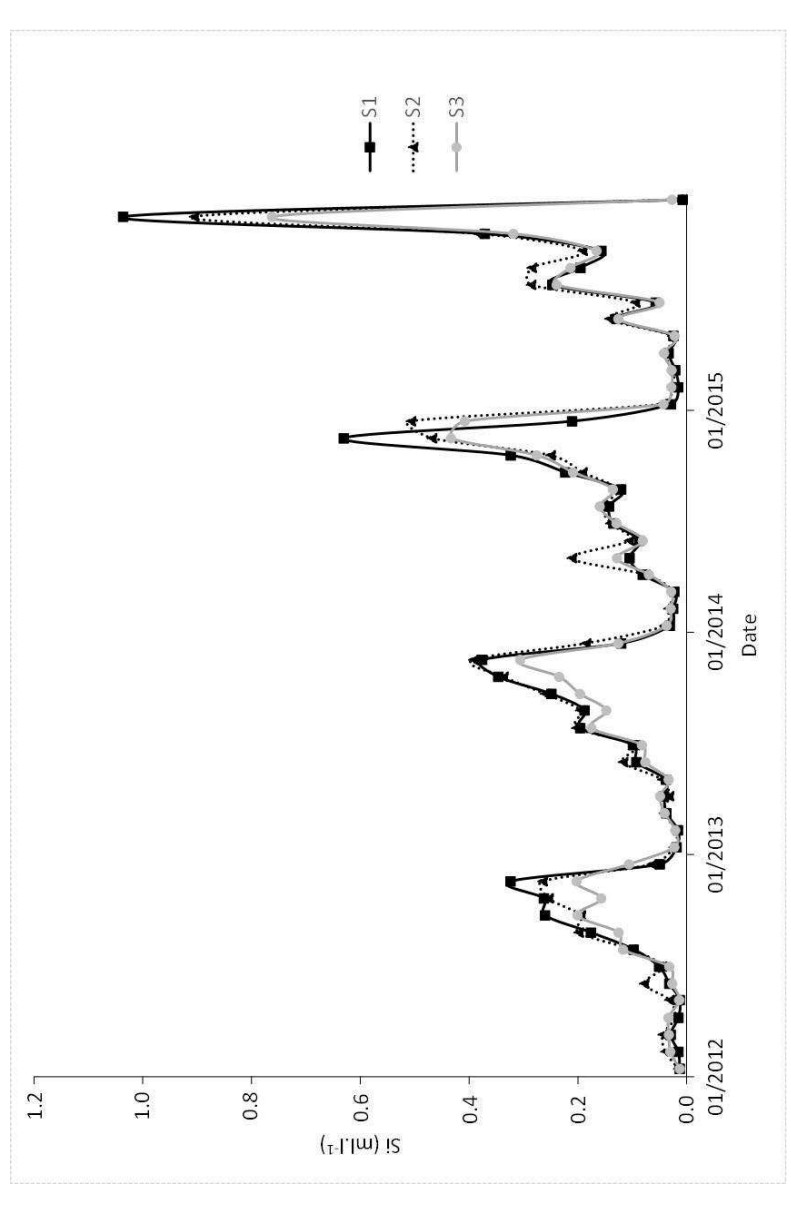

Figure 2





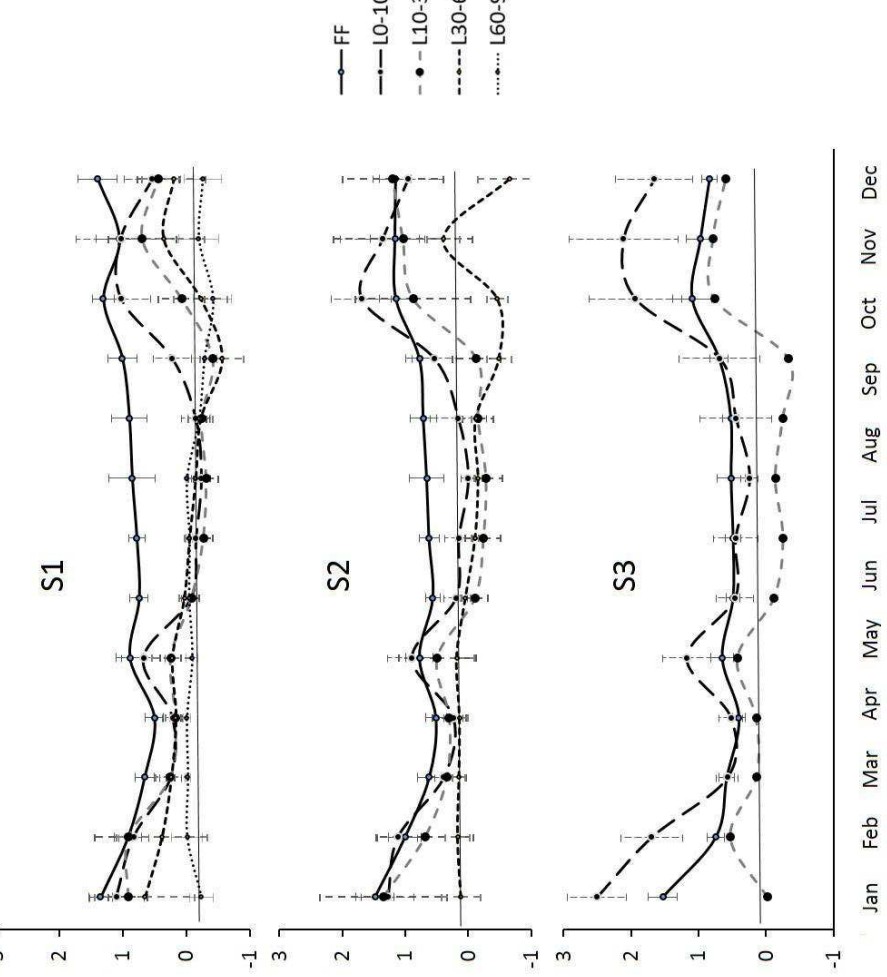

Figure 3



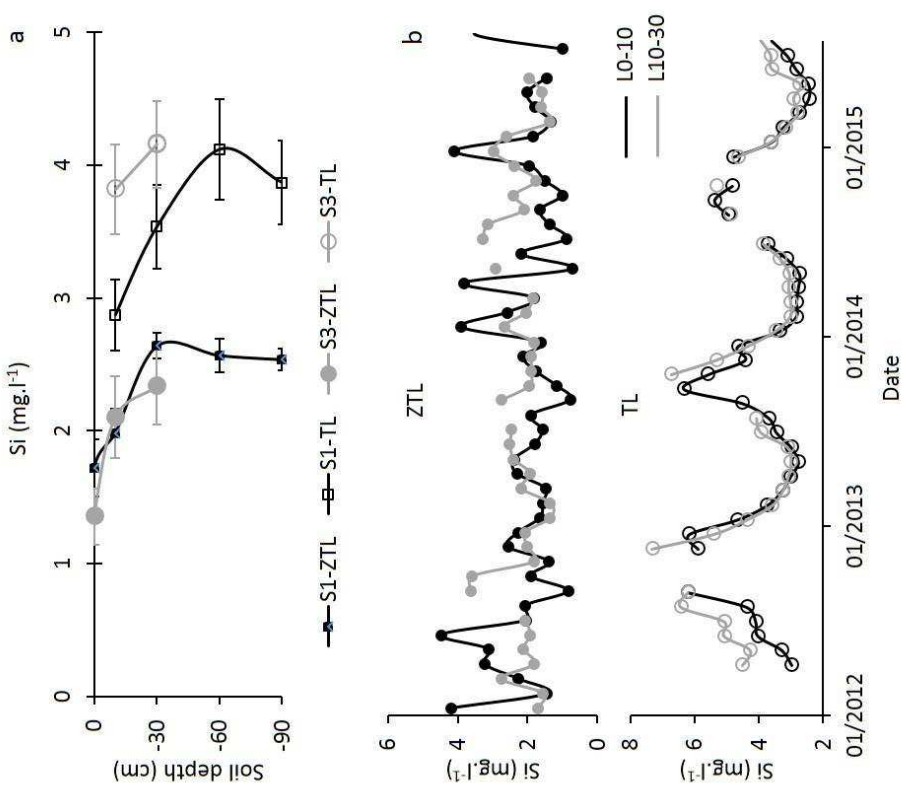

Figure 4





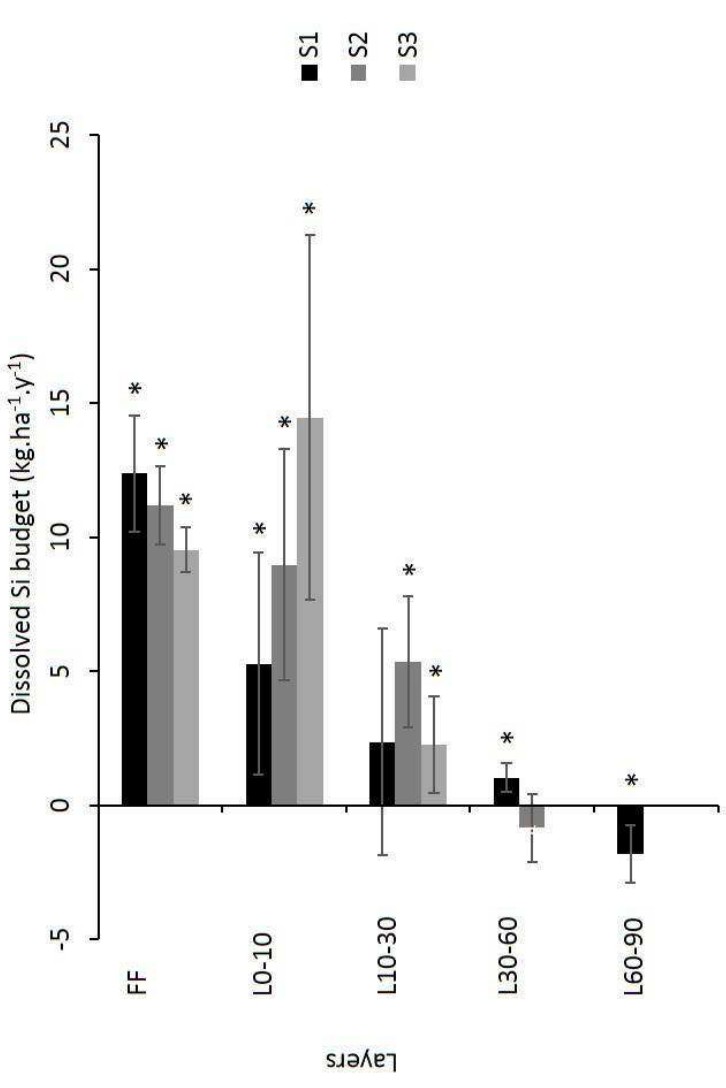

Figure 5





Figure 6

