# Peer review of "Silicon cycle in a temperate forest ecosystem: role of fine roots and litterfall recycling and influence of soil types"

_Biogeosciences, 2017_

## Referee Comment (RC1) · Anonymous Referee #1 · 3 Jan 2018

Review
Manuscript number: bg-2017-469

General comments

First of all I would like to thank the Associate Editor in charge for the opportunity to review the present manuscript entitled 'Silicon cycle in a temperate forest ecosystem: role of fine roots and litterfall recycling and influence of soil types'. The authors (Turpault et al.) analyzed silicon (Si) cycling in three temperate forest ecosystems with

different soil types (Dystric Cambisol, Eutric Cambisol, Rendzic Leptosol). In this context, Turpault et al. aimed to unravel the specific role of fine roots and soil properties on Si cycling. The authors found that fine roots potentially play an important role in Si cycling as their Si concentration seems to be comparable to the Si concentration of leaves. Furthermore, Turpault et al. found the Si concentrations in fine roots and leaves to be dependent on the concentration of dissolved Si in the soils. Turpault et al. concluded from their results that biological processes play a predominant role in Si cycling of the studied sites. In my opinion the article of Turpault et al. generally is of interest for the readers of BIOGEOSCIENCES. However, I identified several shortcomings of the manuscript which should be addressed before potential publication.

In general, the authors should:

- Use units following the rules of the 'International System of Units' (e.g., g kg-1 and not g.kg-1; Please check the whole manuscript on that because in almost all units these dots were used)

- add some literature that is most relevant to their article from my point of view and will help to present a more appropriate discussion of their results (please see my specific comments to the single sections below)

- reconsider the presentation of their results (in the current form I found reading of some subsections of the results section quite exhausting as the authors only repeat the data one-on-one as given in the Tables; I also miss a 'joining' of data, e.g., by some simple correlation analyses)

- rework some subsections (in the current manuscript there are some redundant passages in different subsections; Additionally, specific paragraphs should be displaced to corresponding subsections, e.g., methods should be given only in the Materials and Methods section)

Please find corresponding details on the different subsections listed below. I am really

looking forward to reading the revised manuscript.

Abstract

l.16: Actually, there is at least one publication where Si pools of fine roots were determined (Maguire, T. J., Templer, P. H., Battles, J. J., & Fulweiler, R. W. (2017). Winter climate change and fine root biogenic silica in sugar maple trees (Acer saccharum): Implications for silica in the Anthropocene. Journal of Geophysical Research: Biogeosciences, 122(3), 708-715). So, please relativize your statement ('rare is known...', for example).

l.21: I would recommend using DC, EC and RL for Dystric Cambisol, Eutric Cambisol and Rendzic Leptosol, respectively, instead of S1, S2, S3. If you follow my recommendation, please change this within the whole manuscript (and figures and tables).

Introduction

l.48: Please change '...Si in soils also had a biogenic origin...' to '...Si in soils can also be of biogenic origin...'.

l.58: Please change '...transpiration have also influenced...' to '...transpiration also influence...'.

l.65: I would recommend using the classification of BSi pools as given in Puppe et al. (2015) (Puppe, D., Ehrmann, O., Kaczorek, D., Wanner, M., & Sommer, M. (2015). The protozoic Si pool in temperate forest ecosystems — Quantification, abiotic controls and interactions with earthworms. Geoderma, 243, 196-204), i.e., zoogenic, phytogenic, microbial and protistic Si pools.

l.70: From my point of view you should add Meunier et al. (2017) (Meunier, J. D., Barboni, D., Anwar-ul-Haq, M., Levard, C., Chaurand, P., Vidal, V., Grauby, O., Huc, R., Laffont-Schwob, I., Rabier, J., and Keller, C.: Effect of phytoliths for mitigating water stress in durum wheat, New Phytol., 215, 229–239, https://doi.org/10.1111/nph.14554, 2017) and Puppe et al. (2017) (Puppe, D., Höhn, A., Kaczorek, D., Wanner, M.,

Wehrhan, M., & Sommer, M. (2017). How big is the influence of biogenic silicon pools on short-term changes in water-soluble silicon in soils? Implications from a study of a 10-year-old soil–plant system. Biogeosciences, 14(22), 5239-5252) here as these articles also show the importance especially of small-scale phytogenic Si.

l.76: I guess you mean 'sap' instead of 'soap' here, right?

l.93: Please change '...soil conditions differ between...' to ...soil conditions differ, whereas climate conditions, ...'.

ll.95-102: From my point of view this paragraph belongs to the Material and Methods section.

l.104: Please replace 'where' by 'because'.

Materials and Methods

l.107: Please change 'referred' to 'referred to'.

l.108: Please add 'located' after 'is'.

l.113: I would recommend giving the meanings of these abbreviations here.

l.114: Please change 'are' to 'were' and add 'calculated' before 'from'.

l.117: Please add the scientific name of sycamore maple (Acer pseudoplatanus?).

l.142: Please add 'at 130 cm height' after 'circumferences'.

ll.154/155: Do you mean: 'Subsequently, the branches were separated...'?

l.155: Please add 'in' before 'diameter'.

l.160: I would recommend deleting '(at least fifty kg of soil sample)'.

l.169: I guess you mean '20 cm depth', right?

ll.171/172: Do you mean 'element concentration' instead of 'mineral content'?

l.172: Please add the magnification used for microscopical analyses.

l.174: Did you check these samples, e.g., by SEM-EDX, to ensure that you removed all soil particles (especially the ones on a $\mu$m-scale)?

l.216: Please change 'spectroscopy' to 'spectrometer'.

l.232: Please add 'Titanium' before 'Ti' and set 'Ti' in brackets.

l.243: You already introduced 'C130' as abbreviation before (l. 142).

l.265: Please replace 'D(X)' by 'DSi'.

l.278 & l.279: Please change 'C1.30' to 'C130'.

l.310: Please replace 'are' by 'were'.

l.318 & l.320 & l.326: Please replace 'kg of Si by ha-1.y-1' by 'kg Si ha-1 y-1'.

l.332: How did you analyze the amorphous Si fraction (alkaline extraction?)? I cannot find it in the M&M section.

l.335: Why did you use 'year' as a factor here? You generally assume Si pools, in- and outputs to be more or less equal each year (otherwise you would not calculate means for the analyzed period 2012-2015), so you should not expect any time-related effects, right?

l.337: If your data are not normally distributed (as you said before) you should use nonparametric tests only (i.e., the Mann-Whitney U test instead of the Student's t-test). Statistical analyses: From my point of view you should also use some simple correlations (Spearman's rank correlation) to detect relationships within your data. If you find (and you will find, I am quite sure) significant correlations you can have a closer look at these relationships using some more elaborated statistical tests.

Results

l.348: Do you mean 'Aged' instead of 'Altered'?

l.348 & l. 349: I would recommend using 'testate amoebae' instead of 'amoebae'.

l.373: The numbers did not change after calculation? Please check again your calculations.

Results: I know the results section of a paper often is not like a thriller. However, you should try to make it at least easy to read. So please do not only repeat the data as they are already given in the figures and tables because this is quite exhausting to read. Please rework your results section (from my point of view, subsection 3.1.6 is a good example how to present you results in a more appropriate way).

l.414: Please use powers of 10 for such big numbers.

ll.451-454: Please avoid to give redundant information (see 3.2.2) and to 'jump' between your figures (try to refer to every figure only one time).

ll.456-459: Where can I find this information (Fig., Table?)?

Discussion

ll.461-464: I would recommend deleting this paragraph.

l.473: You should discuss your results in the context of the results of Maguire et al. (2017) here.

l.484: Do you mean 'Sommer et al. (2013)' instead of 'Sommer et al. 2003'?

l.497: I would recommend using 'Mineral soil content' instead of 'Soil pollution'.

l.502: Please give an example for biological activities (e.g., bioturbation by earthworms).

l.503: Please replace 'Si pollution' by 'Si input'.

l.505: What did they study, dust deposits or Si in the litterfall? Please be more precise.

l.507: A space is missing between 'et al.' and '2017'.
l.509: Please add 'that of' after 'than' and give references for these data.

l.510: Diverse taxa of testate amoebae synthesize SiO2-platelets for shell construction, but they do not possess a skeleton.

l.511: Actually, testate amoeba shells represent the protozoic Si pool in soils and not the zoogenic one (which is represented by sponge spicules) (see Puppe et al. 2015).

l.512: I would recommend changing 'zoogenic pool could represent half. . .' to 'testate amoebae may use half . . . for shell synthesis'.

l.513: I would recommend deleting 'in Europe' as Sommer et al. (2013) only analyzed one site (in Germany).

l.518: Another output flux is only likely if you assume balanced in- and outputs in general. From my point of view your data clearly indicate an accumulation of BSi in the organic layers. Please give some more references here to support your findings.

ll.520-522: Please give some references to support your assumption.

l.524: In general, the amorphous Si fraction includes pedogenic and biogenic Si.

l.525: I would recommend adding Puppe et al. (2015) here as they analyzed testate amoebae and corresponding Si pools in detail.

l.527: What do you mean with 'Si-amorphous'?

l.529: Please replace 'until' by 'down to'.

l.532: Do you show these relationships in the results section? If not, you should do so.

ll.534/535: Do you have data on this? If yes, you should present them in your paper. If not, please give some references to support your assumption.

l.541: What do you mean with 'biogenic origin'? Please clarify.

l.542: Please change 'plant' to 'plants'.

l.550: I would recommend stating here that BSi in general is more soluble than soil minerals (Fraysse and co-workers did some nice experiments on this).

l.552: What about deforestation as an important Si output (anthropogenic desilicification)? Please also discuss this important factor and give corresponding literature.

l.563: Please replace 'amoebae' by 'testate amoebae'. Do you have an idea about the population size of testate amoebae at your site (individual numbers)?

ll.564-566: Please avoid to give redundant information.

ll.571/572: Please change '...strong influence of biological partners, mainly fine roots, and processes in the Si cycle' to '...strong biological influence mainly of fine roots.'

ll.576-579: Please avoid redundant information.

l.577 & l.578: Do you mean '3 x $10^3$' and '0.7 x $10^3$' here?

ll.580-583: It is known that the concentration of dissolved Si is a key factor for Si concentrations of plant components (as you also write in your introduction). So please give corresponding literature here and do not highlight this result as a new one. Furthermore, there is also a phylogenetic factor, i.e., phytolith production is probably more influenced by the phylogenetic position of a plant than by environmental factors like temperature or Si availability. In this context, you should also discuss and cite, for example, Hodson et al. (2005) (Hodson, M. J., P. J. White, A. Mead & M. R. Broadley (2005). Phylogenetic variation in the silicon composition of plants. Annals of Botany 96, 1027-1046).

l.583: Please add 'in soils' after 'concentrations'.

l.584. Why Si concentrations are higher especially in these plant components (leaves: transpiration termini; Roots: special protection of relatively fast growing fine roots)? Please give a more detailed discussion here and add corresponding literature.

Conclusions

l.596: Please be careful with statements like 'the complete Si cycle'. I would recommend using 'the Si cycle' instead.

l.601: Please replace 'to give dissolved Si' by 'in the form of dissolved Si'.

l.603: Please add 'on a decadal time scale' after 'biogeosystem'.

l.606: Please add 'concentrations' after 'Si'.

ll.608/609: I would recommend using 'release' or 'instead of 'production'.

Figure captions

Fig. 1: Did you use EDX for elemental analyses?

l.760: Please replace 'amoebae' by 'testate amoebae'.

l.761: Please change 'altered' to 'aged' and replace 'testate amoebae' by 'testate amoeba shells'.

l.774: Please replace 'Histograms' by 'Bars'.

Tables

l.792: Please replace 'Are presented the mean values. . .' by 'Presented are the mean values. . .'

l.798: Please specify which differences were evaluated (DC vs. EC vs. RL?).

Table 3: Why did you test only the total soil depth on statistical significance? Interestingly, the upper compartments are quite comparable at the three sites, only in the deeper soils (30-90 cm) there seem to be significant differences.

Table 4: I cannot find any letters marking statistical significances.

Better use 'Si concentration' for Si in g kg-1 or mg l-1 instead of 'Si content'.

What about the mineral composition of the different soils? It would be nice to have also

data on this.

Figures

Fig. 1: What are the black arrows pointing at (micrograph c)? Please specify or give uniform arrows.

Fig. 2 & 3: Why do you use single data of four years in one diagram (Fig. 2) and means with standard deviations in another one (Fig. 3)? I would recommend unifying the presentation of your results.

Fig. 5: Please correct the unit of dissolved Si.

Fig. 6: As you can go full color in this journal I would recommend using different colors for data of the different sites.

Fig. 6: Please correct the values of soil Si pools ($x10^3$).

Fig. 6: What about soil pH effects? I especially wonder at Si drainage values of S3 (0-10 cm). You should also give a more detailed discussion on this aspect.

---

## Referee Comment (RC2) · Anonymous Referee #2 · 22 Jan 2018

In this manuscript, Marie-Pierre Turpault et al. address the role of fine roots, litterfall and soil type on Si cycling in a temperate forest system. The main and surprising novelty of this manuscript lies in the observation that fine roots actually are a large Si reservoir in forest soils. To my knowledge, no other authors have ever performed a similarly detailed exercise to quantify the amount of Si in the forest root system. Quantifying root biomass is difficult, and these authors have done a tremendous effort to take on this challenge.

While this is a finding worth publishing in itself, I have strong reservations regarding the mass balance the authors have made for the whole forest ecosystem. These reservations are mainly related to the applied methodology to analyse for Si in the soil system, which is inadequate to assess the complicated Si cycle in the soil, as it does not distinguish any pedogenic nor biogenic Si fractions from the abundant mineral fractions. This prevents to make any major conclusions on the role of soil type in the Si mass balance, and also makes it difficult to assess the cycling of litterfall Si in soils, once dissolved. Multiple secondary pedogenic fractions are accumulated deeper in the soil.

In conclusion, I am impressed with the root Si quantification the authors have performed, and I think that a focused manuscript emphasizing the importance of roots in the forest Si cycle is worthy of publication. I also think that a more focused manuscript would have a larger impact on the interested scientific community. The authors should either improve methodology if they want to address the full Si cycle in the forest, or far better emphasize the methodological shortfalls in their discussion, that prevent to make any statement on the full forest Si cycle, and focus on the interesting story of the roots. I will make more detailed comments below.

Line 45: I am becoming a bit annoyed by all Si manuscripts starting with the same statement. Can we just accept that it is now common knowledge that there is a lot of Si in the Earth's crust, and that minerals dissolve. This manuscript is about forest Si cycling, and the role of biological processes in the Si cycle. This has been well described in several review papers over the last years (e.g. Conley, GBC, 2002, Volume 16; Cornelis et al., Biogeosciences, 2011, Volume 8; Struyf Conley, 2012, Biogeochemistry, Volume 107).

Line 57 and beyond: I really don't see why this is important to this manuscript. The division between accumulators, excluders and neutrals is anyway arbitrary, if based on concentration. The Si uptake of plants is also governed by external Si factors, such as its availability.

Line 62: Why also? You have not referred to forests before, so 'also' seems out of place here. What about wetlands, one of the most studied system in the biological Si

cycle? If you provide a list, wetlands should be there.

Line 67: Here comes the first reference to later methodological issues. This statement is untrue. In recent years, methodologies have been developed that allow to distinguish pedogenic, reactive mineral and biogenic Si phases in soils (e.g. Barao et al. 2014, European Journal of Soil Science, 65, Barao et al., LO Methods, 13, 2015; Georgiadis et al. , 2015, Soil Research 52).

Line 76: soap? Probably sap is meant.

Line 90: the second hypothesis is not really novel, Cornelis (et al.) (see also reference list of paper) has already published multiple papers on this issue. In these papers, he shows that methododology is quintessential in addressing the complicated soil type-Si cycling coupling, and the applied method that does not distinguish any secondary soil Si fractions from minerals is inadequate to address the hypothesis.

Line 93-95: awkward wording, consider revising

Line 104: Why? If you want to address the whole forest Si cycle, the soil is of the essence. If you do not apply best available methods (see above) here, then you start with a strong handicap.

Line 113: without any reference to these networks, their relevance is not clear.

General: ceramic cups? Why not plastic? Can ceramic cups potentially add Si to solution? Has this been tested?

Line 209: total fusion is unable to provide sufficiently detailed results for assessing soil Si cycling, where multiple secondary Si fractions form that are actually essential in the whole ecosystem Si balance.

General: I miss any comparison with recent studies that have also made forest Si efflux quantifications. How do your fluxes compare to e.g. Struyf et al. (2010, Nature Communications, 1 and Clymans et al. 2013, Biogeochemistry, 11). I think a section putting

the observed effluxes in the context of other literature, would be far more interesting than the attempt to discuss the role of soil Si processes in the forest Si cycle, given the flawed methodology here. The suction cups do provide an idea of the leakage, and focus should be on how this compares to root turnover and forest Si uptake. In general, I have the impression that Si efflux in this paper is rather low compared to other studies. Is this maybe because these are young forests? Or due to management?

Line 451: Consumption during autumn? Rather contradictory to forest growth in spring and summer? Pedogenic processes at play? Also in apparent contrast to later references to a net Si efflux in fall (Line 536)?

Line 462: I would not use "global" in this local ecosystem context

Line 515-522: I don't understand. First a significant accumulation is discussed, but a few lines below limited accumulation is mentioned?

Line 547: I don't understand how you can state the biological origin, if you apply total fusion.

---

## Author Comment (AC1) · 13 Feb 2018

**Dear colleague,**

Firstly, we would like to thank you (reviewer 1) for this careful review of our paper and the relevance of your comments and suggestions, notably regarding the literature proposed. This has considerably improved the manuscript. Below, presented are your remarks (in black), and our responses and the location of the modifications brought to the text (in blue).

**General comments**

First of all I would like to thank the Associate Editor in charge for the opportunity to review the present manuscript entitled 'Silicon cycle in a temperate forest ecosystem: role of fine roots and litterfall recycling and influence of soil types'. The authors (Turpault et al.) analyzed silicon (Si) cycling in three temperate forest ecosystems with different soil types (Dystric Cambisol, Eutric Cambisol, Rendzic Leptosol). In this context, Turpault et al. aimed to unravel the specific role of fine roots and soil properties on Si cycling. The authors found that fine roots potentially play an important role in Si cycling as their Si concentration seems to be comparable to the Si concentration of leaves. Furthermore, Turpault et al. found the Si concentrations in fine roots and leaves to be dependent on the concentration of dissolved Si in the soils. Turpault et al. concluded from their results that biological processes play a predominant role in Si cycling of the studied sites. In my opinion the article of Turpault et al. generally is of interest for the readers of BIOGEOSCIENCES. However, I identified several shortcomings of the manuscript which should be addressed before potential publication.

**In general, the authors should:**

- Use units following the rules of the 'International System of Units' (e.g., g kg-1 and not g.kg-1; Please check the whole manuscript on that because in almost all units these dots were used)

The units were modified in the whole revised version of the manuscript to follow the rules of the 'International System of Units'.

- add some literature that is most relevant to their article from my point of view and will help to present a more appropriate discussion of their results (please see my specific comments to the single sections below)

Some relevant literature, including the references proposed by both reviewers, was added in the manuscript. Besides the introduction was considerably modified.

See below the detail.

- reconsider the presentation of their results (in the current form I found reading of some subsections of the results section quite exhausting as the authors only repeat the data one-on-one as given in the Tables; I also miss a 'joining' of data, e.g., by some simple correlation analyses).

The whole result section was rewritten to improve its reading.

See an example in the result section.

- rework some subsections (in the current manuscript there are some redundant passages in different subsections; Additionnally, specific paragraphs should be displaced to corresponding subsections, e.g., methods should be given only in the Materials and Methods section)

Some subsections or paragraphs were deleted or displaced in the appropriate sections. See below the detail.

Please find corresponding details on the different subsections listed below. I am really looking forward to reading the revised manuscript.

**Abstract**

I.16: Actually, there is at least one publication where Si pools of fine roots were determined (Maguire, T. J., Templer, P. H., Battles, J. J., & Fulweiler, R. W. (2017). Winter climate change and fine root biogenic silica in sugar maple trees (Acer saccharum): Implications for silica in the Anthropocene. Journal of Geophysical Research: Biogeosciences, 122(3), 708-715). So, please relativize your statement ('rare is known. . .', for example).

Thanks for this remark. We thus modified the sentence in the abstract. The main results and conclusions of the work of Maguire et al. (2017) was introduced in the discussion section.

Please refer to line 21: However, to date, rare is known about the specific role of fine roots.

Please refer to line 540: The Si content in beech fine roots was very higher (2 to 6 times) than that measured by Maguire et al. (2017) for another deciduous species, i.e. sugar maple (*Acer saccharum*) but in a cooler environment. Besides Maguire et al. (2017) demonstrated in this study that increased soil freezing significantly lowers the Si content of sugar maple fine roots.

I.21: I would recommend using DC, EC and RL for Dystric Cambisol, Eutric Cambisol and Rendzic Leptosol, respectively, instead of S1, S2, S3. If you follow my recommendation, please change this within the whole manuscript (and figures and tables).

DC, EC, and RL are now used in the manuscript (text, tables and figures) instead of Dystric Cambisol, Eutric Cambisol and Rendzic Leptosol and instead of S1, S2, and S3.

**Introduction**

As suggested by reviewer 2, the introduction was rewritten to focus on forest ecosystems and on the main objective of the study.

Please refer to line 45: 1 Introduction:

It has recently been shown that intense biogeochemical cycling of Si occurs in the different terrestrial ecosystems, i.e., wetlands (Struyf et al., 2007; Emsens et al., 2016), grasslands (Blecker et al., 2006; White et al., 2012), tropical forests (Lucas et al., 1993, Alexandre et al., 1997) and temperate forests (Bartoli, 1983; Watteau and Villemin, 2001; Gerard et al, 2008; Cornelis et al., 2010a; Cornelis et al., 2011a; Sommer et al., 2006; Sommer et al., 2013). Several review papers well described that soil DSi is taken up by vascular plants and translocated into biogenic Si (BSi) under opal form which is deposited into the cell walls, cell luminas and intercellular spaces (Jones and Handreck, 1965; Conley et al., 2002; Cornelis et al, 2011b; Struyf and Conley, 2012). These structures are called phytoliths. Other important

producers of biogenic Si are animals especially diatoms, sponges and testate amoebae. (Struyf and Conley, 2012; Sommers et al., 2006; Puppe et al., 2014; Puppe et al., 2015).

According to Conley (2002), the annual fixation of DSi into terrestrial ecosystems has been estimated to range from 60 to 200 Tmoles. That represents 10 to 40 times more than yearly export DSi and suspended biogenic Si from the terrestrial geobiosphere to the coastal zone (Conley, 2002). Vegetation can thus be considered as a factory of BSi which returns to the soil as organic matter through biological recycling. Because BSi in general is more soluble than silicate minerals, BSi strongly contributes to the DSi pool (Fraysse et al., 2009; Cornelis and Delvaux, 2016).

Based on the assumption that the storage of Si is limited in roots (Bartoli and Souchier, 1986) and because fine root sampling and cleaning before analyses are long and tedious processes, studies in forest ecosystems mainly focus on the importance of litterfall recycling on the Si biogeochemical cycle without quantifying Si in the roots (Gérard et al., 2008; Cornelis et al., 2010a; Sommer et al., 2013). However, Krieger et al (2017) recently showed that Si in deciduous trees (European beech, *Fagus sylvatica* and sycamore maple, *Acer pseudoplatanus*) generally precipitates as a thin layer (< 0.5  $\mu$ m) around the cells, especially in roots and bark. These small-scale phytogenic Si was demonstrated to influence various soil and plant processes (Meunier et al., 2017; Puppe et al., 2017).

Considering the large amount of Si precipitates in roots (Krieger et al., 2017) and the rapid turnover of fine roots in forest ecosystems (approximately one year in beech forests in Europe; Brunner et al., 2013), we hypothesized that fine roots could significantly contribute to the input of BSi into the soil. To test this hypothesis, we quantified during a four-year observation period (i) the total and annual accumulations of Si in stand belowground and abovegound biomasses while distinguishing annual and perennial compartments, ii) the Si input fluxes in the forest floor (litterfall and small woods, aboveground exploitation residues) and in the soil (fine roots and belowground exploitation residues). The study was led in a lowland (low lateral transfer of material) deciduous temperate forest developed on three soils, ranging from a shallow calcic soil to a deep acidic soil, with mull to acid mull humus. These humus forms quickly degrades, contain few soil particles and no root thus allowing to determine the DSi issued from the degradation of organic layers contrary to mor or moder humus forms (Sommer et al., 2006; Cornelis et al., 2010a). In addition, we monthly quantified in these ecosystems the Dsi inputs and outputs, i.e., rainfall, foliar leaching and drainage, in order to assess the seasonal dynamics of these fluxes induced by biological activities.

I.48: Please change '. . . Si in soils also had a biogenic origin. . . ' to '. . . Si in soils can also be of biogenic origin. . . '.

This sentence was deleted.

1.58: Please change '...transpiration have also influenced...' to '...transpiration also influence...'.

**This sentence was deleted.**

l.65: I would recommend using the classification of BSi pools as given in Puppe et al. (2015) (Puppe, D., Ehrmann, O., Kaczorek, D., Wanner, M., & Sommer, M. (2015). The protozoic Si pool in temperate forest ecosystems âA T Quantification, abiotic controls \* and interactions with earthworms. Geoderma, 243, 196-204), i.e., zoogenic, phytogenic, microbial and protistic Si pools.

**This sentence was deleted.**

I.70: From my point of view you should add Meunier et al. (2017) (Meunier, J. D., Barboni, D., Anwarul-Haq, M., Levard, C., Chaurand, P., Vidal, V., Grauby, O., Huc, R., Laffont-Schwob, I., Rabier, J., and Keller, C.: Effect of phytoliths for mitigating water stress in durum wheat, New Phytol., 215, 229–239, <a href="https://doi.org/10.1111/nph.14554">https://doi.org/10.1111/nph.14554</a>, 2017) and Puppe et al. (2017) (Puppe, D., Höhn, A., Kaczorek, D., Wanner, M., Wehrhan, M., & Sommer, M. (2017). How big is the influence of biogenic silicon pools on short-term changes in water-soluble silicon in soils? Implications from a study of a 10-year-old soil—

plant system. Biogeosciences, 14(22), 5239-5252) here as these articles also show the importance especially of small-scale phytogenic Si.

We agree that this literature is of importance so we added it in the revised version of the manuscript.

Please refer to line 65: However, Krieger et al (2017) recently showed that Si in deciduous trees (European beech, Fagus sylvatica and sycamore maple, Acer pseudoplatanus) generally precipitates as a thin layer (< 0.5  $\mu$ m) around the cells, especially in roots and bark. These small-scale phytogenic Si was demonstrated to influence various soil and plant processes (Meunier et al., 2017; Puppe et al., 2017).

1.76: I guess you mean 'sap' instead of 'soap' here, right?

This sentence was deleted.

I.93: Please change '. . .soil conditions differ between. . .' to . . .soil conditions differ, whereas climate conditions, . . .'.

This sentence was deleted.

1.95-102: From my point of view this paragraph belongs to the Material and Methods section.

This sentence was deleted.

I.104: Please replace 'where' by 'because'.

This sentence was deleted.

Materials and Methods

l.107: Please change 'referred' to 'referred to'. l.108: Please add 'located' after 'is'.

These were modified.

Please refer to line 149: The experimental site, hereafter referred to as the Montiers site (http://www.nancy.inra.fr/en/Outils-et-Ressources/montiers-ecosystem-research), is located in the Montiers-sur-Saulx beech forest in northeastern France (Meuse, France, latitude 48° 31′ 54″ N, longitude 5° 16′ 08″ E).

I.113: I would recommend giving the meanings of these abbreviations here.

The meanings of these abbreviations were added.

Please refer to line 154: The Montiers site is part of different national and international research networks, i.e., SOERE (Long-lasting observation and experimentation for the research on environment)-OPE (Perennial Environment Observatory) and F-ORE-T (Functioning of Forest Ecosystems), and AnaEE (Analysis and Experimentations on Ecosystems).

I.114: Please change 'are' to 'were' and add 'calculated' before 'from'.

This was modified.

Please refer to line 159: The mean annual rainfall and temperature over the last twenty years were 1069 mm and 9.8°C, respectively (calculated from Météo-France data).

I.117: Please add the scientific name of sycamore maple (Acer pseudoplatanus?).

This was added.

Please refer to line 166: The stand was mainly composed of beech (89%) and 11% of other deciduous species, i.e., sycamore maple (*Acer pseudoplatanus*), ash (*Fraxinus excelsior*), pedunculate oak (*Quercus robur* L.), European hornbeam (*Carpinus betulus* L.), and wild cherry (*Prunus avium*).

I.142: Please add 'at 130 cm height' after 'circumferences'.

This was added.

Please refer to line 193: Trees were chosen to cover most of the range of stem circumferences at 130 cm height (C130) in each plot.

I.154/155: Do you mean: 'Subsequently, the branches were separated. . . '?

Yes, the sentence was modified.

Please refer to line 205: Subsequently, the branches latter were separated into different classes, i.e., < 4, 4-7 and > 7 cm diameter, according to Henry et al. (2011).

I.155: Please add 'in' before 'diameter'.

This was added.

Please refer to line 206: the branches latter were separated into different classes, i.e., < 4, 4-7 and > 7 cm in diameter, according to Henry et al. (2011).

1.160: I would recommend deleting '(at least fifty kg of soil sample)'.

This was deleted.

Please refer to line 211: A two-step procedure was applied to accurately assess the fine root biomass (Bakker et al., 2008), without having to transport soil to the laboratory.

I.169: I guess you mean '20 cm depth', right?

Right so we modified the text.

Please refer to line 219: Roots with a diameter > 2 cm (small and coarse roots) were collected in February 2017 in three soil pits (approximately 0.4 m wide) for each plot where soil material was cut and extracted at approximately 20 cm depth.

I.171/172: Do you mean 'element concentration' instead of 'mineral content'?

Yes, this was modified.

Please refer to line 224: An aliquot of each root sample (fine, small and coarse) was then collected to determine element concentration.

I.172: Please add the magnification used for microscopical analyses.

This was added.

Please refer to line 226: The absence of soil particles was carefully checked under a binocular microscope with a magnification of 10x.

I.174: Did you check these samples, e.g., by SEM-EDX, to ensure that you removed all soil particles (especially the ones on a  $\mu$ m-scale)?

All samples were observed with binocular microscope but only some samples of fine roots were observed by SEM-EDX (see part 2.3.4. Microscopic analysis).

I.216: Please change 'spectroscopy' to 'spectrometer'.

This was changed.

Please refer to line 267: The samples were examined at the GeoRessources laboratory (University of Lorraine) for biomineral occurrence and composition, using a Hitachi S-4800 scanning electron microscope (SEM) equipped with an energy-dispersive X-ray spectrometer (EDX), containing a lithium-drifted Si detector.

1.232: Please add 'Titanium' before 'Ti' and set 'Ti' in brackets.

This was added.

Please refer to line 285: The percentage of soil mixed with the organic horizons was determined through the use of titanium (Ti).

I.243: You already introduced 'C130' as abbreviation before (I. 142).

This was corrected.

Please refer to line 296: To transform the stemflow volumes to a water flux, C130 was assumed to explain the inter-individual stemflow volume variability within a species.

I.265: Please replace 'D(X)' by 'DSi'.

This was replaced.

Please refer to line 318: where  $D_{Si}$  is the drainage flux of Si,  $D_G$  is the water drainage via rapid gravitational transfer,

I.278 & I.279: Please change 'C1.30' to 'C130'.

This was changed.

Please refer to lines 332: It included four steps, (i) the circumference of all trees was measured at 1.30 m height,  $C_{130}$ , in 2011 and 2015; (ii) eight trees in each plot, representing the range of  $C_{130}$ , stem bark and wood and 0-4, 4-7 and > 7 cm diameter branches were sampled; (iii) the weighed allometric equations fitted for each ecosystem compartment were calculated according to Calvaruso et al. (2017); and (iv) tree biomass (stem bark and wood and 0-4, 4-7 and > 7 cm diameter branches) was quantified per hectare by applying fitted equations to the stand inventories.

I.310: Please replace 'are' by 'were'.

This was replaced.

Please refer to line 364: The roots were not exported

I.318 & I.320 & I.326: Please replace 'kg of Si by ha-1.y-1' by 'kg Si ha-1 y-1'.

This was replaced in the whole manuscript.

I.332: How did you analyze the amorphous Si fraction (alkaline extraction?)? I cannot find it in the M&M section.

The data of the amorphous Si fraction are not presented in this manuscript. This was deleted.

Please refer to line 383: The normality of the distribution was checked, using the Shapiro-Wilk test. As our data did not follow a normal distribution, the non-parametrical Kruskal-Wallis test was performed to compare the different soil types, biomass pools, biomass increments, Si content, Si pools, and Si fluxes for each tree compartment, and the total soil Si at the threshold level of 0.05.

I.335: Why did you use 'year' as a factor here? You generally assume Si pools, in- and outputs to be more or less equal each year (otherwise you would not calculate means for the analyzed period 2012-2015), so you should not expect any time-related effects, right?

This was a mistake. The term "year" was deleted.

Please refer to line 389: We used the R version 3.3.1 statistical software (R Development Core Team, 2016) and specifically, the R package nlme to test the effect of soil type on annual Si fluxes, by means of a mixed linear analysis of variance (ANOVA) with soil type and their interaction as fixed effects.

I.337: If your data are not normally distributed (as you said before) you should use nonparametric tests only (i.e., the Mann-Whitney U test instead of the Student's test).

We do not agree, the Kruskal-Wallis test is also a non-parametric test used to test at least three samples. I join the procedure of selection of the statistical test for this study at the end of this document.

**Results**

I.348: Do you mean 'Aged' instead of 'Altered'? I.348 & I. 349: I would recommend using 'testate amoebae' instead of 'amoebae'.

Yes, this was corrected.

Please refer to line 402: Aged leaves in the organic horizon were colonized by hyphae and amoebae (Figure 1c) and presented large voids. The Si deposits disappeared from the plant cells but were present in the observed testate amoebae.

1.373: The numbers did not change after calculation? Please check again your calculations.

Sorry for the mistake. This was modified.

Please refer to line 427: The Si pools in the fine roots were important and ranged from 61.2 kg ha-1 in the RL to 98.7 kg ha-1 in the DC. Based on the turnover rate of fine roots, as determined by Brunner et al. (2013) for beech trees, i.e.,  $1.11 \pm 0.21 \text{ y}^{-1}$ , we calculated that the annual Si fluxes resulting from fine root decomposition ranged from 67.9  $\pm$  14.3 kg ha-1 in the RL to 109.5  $\pm$  23.0 kg ha-1 in the DC.

Results: I know the results section of a paper often is not like a thriller. However, you should try to make it at least easy to read. So please do not only repeat the data as they are already given in the figures and tables because this is quite exhausting to read. Please rework your results section (from my point of view, subsection 3.1.6 is a good example how to present you results in a more appropriate way).

The results section was rewritten to be less exhausting to read.

Please refer to lines 393 to 526:

**Example: 3.1.2 Si pools and fluxes in aboveground tree biomass**

The calculated standing aboveground biomass in 2011 increased as follows: RL < DC < EC with significant differences between EC and RL (factor 1.4). (Table 2). The stem bark had the highest Si concentration in the three plots, and the Si pool in this compartment represented approximately 40% of the total Si pool in the aboveground tree biomass. The younger the structures were, the higher Si concentration. Small branches were approximately three times more concentrated than coarse branches in the three soils (Table 2). The amount of Si immobilized in the standing aboveground biomass ranged from 20.1 kg ha-1 on the RL to 26.2 kg ha-1 on the EC. The annual biomass production between 2011 and 2015 increased as follows: RL < EC < DC with significant differences between DC

and RL (factor 1.7). As a result, the amount of Si immobilized in the aboveground biomass each year between 2011 and 2015 ranged from 0.98 kg ha-1 on the RL to 1.82 kg ha-1 on the DC.

I.414: Please use powers of 10 for such big numbers.

This was modified.

Please refer to line 477: The total Si pools in the first 90 cm of soil overpassed  $2.4.10^6$  kg ha-1 in the DC and EC as opposed to approximately  $7.2.10^5$  kg ha-1 in the RL.

II.451-454: Please avoid to give redundant information (see 3.2.2) and to 'jump' between your figures (try to refer to every figure only one time).

We agree that it is not the optimal mean, but in this specific case, we have to make reference a second time to the figure 3 to make the link with the observations resulting from figure 5.

II.456-459: Where can I find this information (Fig., Table?)?

This information is presented in the synthesis figure 6.

Discussion

II.461-464: I would recommend deleting this paragraph.

We agree, this paragraph was deleted.

1.473: You should discuss your results in the context of the results of Maguire et al. (2017) here.

This was added.

Please refer to line 540: This Si content in beech fine roots was very higher (2 to 6 times) than that measured by Maguire et al. (2017) for another deciduous species, i.e. sugar maple (*Acer saccharum*) in a cooler environment. Besides Maguire et al. (2017) demonstrated in this study that increased soil freezing significantly lowers the Si content of sugar maple fine roots.

I.484: Do you mean 'Sommer et al. (2013)' instead of 'Sommer et al. 2003'?

Yes, sorry for the mistake.

Please refer to lines 555: As demonstrated by Sommer et al. 2013, only a small fraction (approximately 1%; from 1.0 kg ha-1 in plot S3 to 1.8 kg ha-1 in plot S1) of the Si taken up by the tree stand accumulated each year in the perennial tree compartments, i.e., the stem, branch and coarse roots (Figure 6, Table 2).

1.497: I would recommend using 'Mineral soil content' instead of 'Soil pollution'.

This was changed.

Please refer to line 569: 4.2.1 Mineral soil content in organic horizons

1.502: Please give an example for biological activities (e.g., bioturbation by earthworms).

This was added.

Please refer to line 573: The higher rate of soil pollution in the study of Cornelis et al. (2010a) can be explained by the presence of a thick Oh layer in the moder that was in direct contact with the superficial soil layer and was characterized by an intense mixing of degraded organic matter with soil

particles, induced by biological activities, mainly bioturbation by earthworms in these soils (Lavelle, 1988).

I.503: Please replace 'Si pollution' by 'Si input'.

This was replaced.

Please refer to line 575: The Si input by dust deposits in the organic horizons was negligible, with a maximum value of 6.0 kg ha-1  $y^{-1}$  (no stand interception) against 151 to 246 kg ha-1.

1.505: What did they study, dust deposits or Si in the litterfall? Please be more precise.

This was clarified.

Please refer to line 577: Lequy et al. (2014), who studied the mineralogy of the dust deposits of the Montiers site, observed that the Si deposits in throughfall was mainly quartz.

I.507: A space is missing between 'et al.' and '2017'.

This was corrected.

Please refer to line 581: The main phytogenic Si input into the organic horizons was opal phytoliths (Krieger et al., 2017), which dissolve slowly (Fraysse et al., 2009) in comparison to the rate of organic matter mineralization.

1.509: Please add 'that of' after 'than' and give references for these data.

This was added.

Please refer to line 582: The residence time of Si in the organic horizons is higher than that of carbon  $(5.3 \pm 0.8 \text{ vs } 1.9 \pm 0.4 \text{ y})$ .

I.510: Diverse taxa of testate amoebae synthesize SiO2-platelets for shell construction, but they do not possess a skeleton. I.511: Actually, testate amoeba shells represent the protozoic Si pool in soils and not the zoogenic one (which is represented by sponge spicules) (see Puppe et al. 2015).

This was corrected.

Please refer to line 583: In addition, the presence of testate amoebae, organisms rich in Si (Figure 1; Sommer et al., 2013), in the organic horizons suggests that a part of the Si from the phytoliths belonged to the protozoic Si pool.

l.512: I would recommend changing 'zoogenic pool could represent half...' to 'testate amoebae may use half... for shell synthesis'. l.513: I would recommend deleting 'in Europe' as Sommer et al. (2013) only analyzed one site (in Germany).

This was changed.

Please refer to line 585: Sommer et al. (2013) estimated that testate amoebae may use half of the Si input by litterfall in beech organic horizons (17 kg ha-1 vs 34 kg ha-1) for shell synthesis.

l.518: Another output flux is only likely if you assume balanced in- and outputs in general. From my point of view your data clearly indicate an accumulation of BSi in the organic layers. Please give some more references here to support your findings. l.520-522: Please give some references to support your assumption.

This paragraph was modified to be clearer, and some relevant literature was added to support our assumptions.

Please refer to lines: During the study period (2012-2015), the Si input in the organic horizons via litterfall were primarily higher than the Si output via soluble transport (assessed in ZTL solutions under

the forest floor) for the three soils. This net flux of Si should have induced the accumulation of Si in the organic horizons, what we did not observe in the four years of the study. This suggests that another output flux existed but was not quantified in our study. This flux is likely the solid particulate migration toward the topsoil layer, as demonstrated by Ugolini et al. (1977). These authors observed that organic particles containing notably silicon were predominant in the migrant material in the upper soil horizons. In our study, the solid particulate migration from the organic horizons to the topsoil may consist of the colloid transport of amoebae (Harter et al., 2000) or the transport of phytoliths (Fishkis et al. 2010). These latter observed, though a field study using fluorescent labelling, that the downrard transport distance of phytoliths after one year was 3.99± 1.21 cm for a Cambisol with a preferential translocation of small-sized phytoliths.

I.524: In general, the amorphous Si fraction includes pedogenic and biogenic Si. I.525: I would recommend adding Puppe et al. (2015) here as they analyzed testate amoebae and corresponding Si pools in detail.

The sentence was modified to integrate these remarks.

Please refer to lines: The Si production (source) in the soil mainly results from pedogenic Si from soil mineral dissolution and from biogenic Si from plant tissues and testate amoebae (Cornelis et al., 2011; Sommer et al., 2013; Puppe et al., 2015).

I.527: What do you mean with 'Si-amorphous'?

The sentence was modified to be more specific.

Please refer to line 606: The immobilization (sink) of dissolved Si in the soil is due to plant and organism immobilization and precipitation of secondary minerals, such as phyllosilicates or Si-bearing short range organization minerals or allophane, immogolite (Dahlgren and Ugolini, 1989; Ma and Yamaji, 2006; Sommer et al., 2013; Tubana et al., 2016; Kabata-Pendias and Mukherjee, 2007).

1.529: Please replace 'until' by 'down to'.

"Until" was replaced by "down to"

Please refer to line 612: A net production of dissolved Si in the soils was observed on the three studied plots down to a depth of 60 cm, showing a positive production/immobilization budget.

1.532: Do you show these relationships in the results section? If not, you should do so.

The correlation coefficient was added in the text.

Please refer to line 615: This is corroborated by the strong relationship between annual Si production in the 10-60 cm soil layers and fine root content (data not shown,  $r^2 = 0.94$ ).

I.534/535: Do you have data on this? If yes, you should present them in your paper. If not, please give some references to support your assumption. Do you have data on this: where testate amoebae and phytoliths accumulated after being transferred from the organic horizons.

We do not have data on this so we deleted this speculative part.

I.541: What do you mean with 'biogenic origin'? Please clarify.

This part of the sentence was deleted to avoid misunderstandings.

Please refer to line 622: At our site, this period was also characterized by a maximum concentration of Si in the bound waters and a negative budget in the 10-cm and 60-cm soil layers, resulting from the precipitation of secondary minerals.

542: Please change 'plant' to 'plants'.

This was changed.

Please refer to line 625: As a result, a drastic decrease of Si production was observed in the surface layer during the vegetation period, where Si uptake by plants occurred (Figure 3).

I.550: I would recommend starting here that BSi in general is more soluble than soil minerals (Fraysse and co-workers did some nice experiments on this).

The sentence was modified as suggested.

Please refer to line 638: Because biogenic Si in general is more soluble than lithogenic or pedogenic Si (Fraysse et al., 2009; Cornelis and Delvaux, 2016), very few of the Si leached within the soil profile directly results from the dissolution of soil minerals, as demonstrated in other studies in temperate forests (Bartoli, 1983; Watteau and Villemin, 2001; Gerard et al., 2008; Cornelis et al., 2010a; Cornelis et al., 2011a; Sommer et al., 2006; Sommer et al., 2013).

I.552: What about deforestation as an important Si output (anthropogenic desilicification)? Please also discuss this important factor and give corresponding literature.

This information was introduced in the manuscript.

Please refer to line 647: Silicon inputs and outputs have minor contributions to the global Si budget in our forest ecosystems, and the Si cycle is mainly driven by internal fluxes, especially recycling of biogenic Si. However, Struyf et al. (2010) observed that land use is the most important controlling factor of Si mobilization in European watersheds. These authors showed that deforestation and conversion to agricultural land or other land uses leads to a twofold to threefold decrease in baseflow delivery of Si.

I.563: Please replace 'amoebae' by 'testate amoebae'. Do you have an idea about the population size of testate amoebae at your site (individual numbers)?

"Amoebae" was replaced by "testate amoebae". We do not assess the population size of testate amoebae in our study sit.

Please refer to line 661: In the organic horizons and in the soil, mainly in the 0-10 cm layer, we observed a high net Si production, likely resulting from the decomposition of litter leaves and testate amoebae in the organic horizons and of fine roots in the soil (Figure 6).

I.564-566: Please avoid to give redundant information.

We think that this information is partially redundant but important here.

II.571/572: Please change '. . .strong influence of biological partners, mainly fine roots, and processes in the Si cycle' to '. . .strong biological influence mainly of fine roots.'

This was changed.

Please refer to line 668: The assessment of Si fluxes and pools in the different compartments of our forested site coupled with a seasonal dynamic follow-up reveal a rapid and almost total recycling of Si in our site and show the strong biological influence, mainly fine roots, and processes in the Si cycle.

I.576-579: Please avoid redundant information.

This sentence was deleted.

I.577 & I.578: Do you mean '3 x 103' and '0.7 x 103' here?

This sentence was deleted.

II.580-583: It is known that the concentration of dissolved Si is a key factor for Si concentrations of plant components (as you also write in your introduction). So please give corresponding literature here and do not highlight this result as a new one. Furthermore, there is also a phylogenetic factor, i.e., phytolith production is probably more influenced by the phylogenetic position of a plant than by environmental factors like temperature or Si availability. In this context, you should also discuss and cite, for example, Hodson et al. (2005) (Hodson, M. J., P. J. White, A. Mead & M. R. Broadley (2005). Phylogenetic variation in the silicon composition of plants. Annals of Botany 96, 1027-1046).

Thank you for this interesting remark and paper. This information was added in the manuscript.

Please refer to line 681: The concentration of dissolved Si in the soil is known to influence opal formation in plants (Cornelis et al., 2010b) but phytolith production seems to be more affected by the phylogenetic position of a plant than by environmental factors (Hodson et al., 2005). For example, these authors demonstrated through meta-analysis of the data, that in general ferns, gymnosperms and angiosperms accumulated less Si in their shoots than non-vascular plant species and horsetails.

1.583: Please add 'in soils' after 'concentrations'.

This was added.

Please refer to line 679: This is in agreement with the observations of Heineman et al. (2016) in tropical forests, which demonstrated that nutrient concentrations in wood and leaves correlated positively with soil Ca, K, Mg and P concentrations in soils.

I.584. Why Si concentrations are higher especially in these plant components (leaves: transpiration termini; Roots: special protection of relatively fast growing fine roots)? Please give a more detailed discussion here and add corresponding literature.

A paragraph dealing with the importance of Si in leaves and roots was added.

Please refer to line 687: Silicon plays several physiological and ecological functions in leaves and roots, such as an involvement in the detoxification of aluminum, oxalic acid, and heavy metals, in the regulation of ion balance, in the reduction of hydric, salt, and temperature stresses (Currie and Perry, 2007; Meunier et al., 2017). They also contribute to the optimization of photosynthesis by gathering and scattering light in the leaves, confer mechanical support and tissue rigidity, and facilitate pollen release, germination, and tube growth (Bauer, Elbaum, & Weiss, 2011; Currie and Perry, 2007; Gal et al.,, 2012). In addition to these physiological functions, Si has also ecological significance by protecting plants against herbivores and phytopathogens (Currie and Perry, 2007; Lins et al., 2002).

**Conclusions**

A synthesis figure was added in the manuscript to summarize the main findings and compare with the data of other studies in similar stand conditions.

Please refer to line 951: **Fig. 7:** Summary scheme of the main findings of this study (TS) and comparison with other studies (L).

I.596: Please be careful with statements like 'the complete Si cycle'. I would recommend using 'the Si cycle' instead.

We agree and thus modified.

Please refer to line 703: By coupling different approaches (annual budget in solid vegetal and solution phases and monthly dynamics of solutions) and methods (direct *in situ* measurements and standard and site specific modelling) to quantify Si pools and fluxes in the different ecosystem compartments, our study allowed us to assess the Si cycle at the forest stand scale.

I.601: Please replace 'to give dissolved Si' by 'in the form of dissolved Si'.

This was replaced.

Please refer to line 709: This suggests that Si cycle is almost closed during the vegetation period; dissolved Si is taken up by vegetation then Si returned to the soil mainly through root and leave decomposition in the form of dissolved Si, which is again taken up by vegetation.

I.603: Please add 'on a decadal time scale' after 'biogeosystem'.

This was added.

Please refer to line 711: This observation is consistent with the observation of Sommer et al. (2013), who demonstrated a low contribution of geochemical weathering processes to the Si cycle in a forest biogeosystem on a decadal time scale.

I.606: Please add 'concentrations' after 'Si'.

This was added.

Please refer to line 715: The plant compartments were Si-enriched in the soil with higher Si concentration, i.e., DC (plot S1) compared to plant compartments in the RL (plot S3), resulting in 1.6-times higher recycling in plot S1 compared to plot S3.

1.608/609: I would recommend using 'release' or 'instead of 'production'.

This was changed.

Please refer to line 718: While Si release was relatively similar in the organic horizons for the three plots, its production in the soil, mainly in the 0-10 cm layer, was twice higher in plot S3 and richer in clays than plot S1

**Figure captions**

Fig. 1: Did you use EDX for elemental analyses?

Yes, the presence of Si was confirmed by EDX for each point with arrows. I join below one of the spectrum carried out on fine roots.

I.760: Please replace 'amoebae' by 'testate amoebae'. I.761: Please change 'altered' to 'aged' and replace 'testate amoebae' by 'testate amoeba shells'.

**This was replaced.**

Please refer to line 916: **Fig. 1:** Si in biological tissues of beech trees observed through Scanning Electron Microscopy. (a) Si precipitates in the intercellular space of fresh leaves, forming phytoliths (white arrow). Deposits of Si (vertical white arrows) in the inner cell walls of fruit capsules (b), stem bark (d and e), bud scales (f), and roots (g, h, and i). (c) Hyphae, testate amoebae and large voids in aged litter leaves. Si deposits only present in the testate amoeba shells (horizontal white empty arrows). The presence of Si was confirmed with EDX (analyzed zones indicated by white vertical arrows).

1.774: Please replace 'Histograms' by 'Bars'.

This was replaced.

Please refer to line 935: Bars with an asterisk are significantly different from 0, according to a Kruskal-Wallis test at the threshold P value level of 0.05.

**Tables**

1.792: Please replace 'Are presented the mean values. . .' by 'Presented are the mean values. . .'

This was modified.

Please refer to line 953: **Table 1:** Physicochemical properties of the three studied soils in the Montiers site (plot S1 – DC; plot S2 – EC; plot S3 – RL). Presented are the mean values for bulk density (g cm-3), textural distribution (g kg-1), total rock volume (RV), soil water holding capacity (SWHC), soil water pH, organic matter content (OM), cation exchange capacity (CEC; cmol+ kg-1) and base-cation saturation ratio (S/CEC, with S = sum of base cations). Standard deviation values are given in italic. Table adapted from Kirchen et al. (2017).

1.798: Please specify which differences were evaluated (DC vs. EC vs. RL?).

This was specified.

Please refer to line 960: **Table 2:** Mean Si contents, pools and fluxes in the biomass of the three soils of the Montiers site. Standard deviation values are given in brackets. Values with different letters are significantly different according to a Kruskal-Wallis test at the threshold P value level of 0.05 (soil effect, DC vs. EC vs. RL).

Table 3: Why did you test only the total soil depth on statistical significance? Interestingly, the upper compartments are quite comparable at the three sites, only in the deeper soils (30-90 cm) there seem to be significant differences.

Statistical significance was added for each depth.

Please refer to line 965.

Table 4: I cannot find any letters marking statistical significances. Better use 'Si concentration' for Si in g kg-1 or mg l-1 instead of 'Si content'. What about the mineral composition of the different soils? It would be nice to have also data on this.

Letters marking statistical significances were added and Si concentration was used.

Please refer to line 970.

Information regarding the geology and the mineralogy of the site was also added but details were already presented in Calvaruso et al. (2017)

Please refer to line 160: The geology of the Montiers site consists of two overlapping soil parent materials: an underlying Tithonian limestone surmounted by detrital acidic Valanginian sediments. The calcareous bedrock contains mainly calcium carbonate and ~3.4% clay minerals. The overlying detrital sediments are complex, as they result from various depositions and are composed of silt, clay, coarse sand and iron oxide nodules (for more details, see Calvaruso et al., 2017).

**Figures**

Fig. 1: What are the black arrows pointing at (micrograph c)? Please specify or give uniform arrows.

The empty horizontal white arrows indicate the location of testate amoebae. This was added in the legend of Figure 1.

Please refer to line 916: **Fig. 1:** Si in biological tissues of beech trees observed through Scanning Electron Microscopy. (a) Si precipitates in the intercellular space of fresh leaves, forming phytoliths

(vertical white arrow). Deposits of Si (white arrows) in the inner cell walls of fruit capsules (b), stem bark (d and e), bud scales (f), and roots (g, h, and i). (c) Hyphae, testate amoebae and large voids in aged litter leaves. Si deposits only present in the testate amoeba shells (horizontal empty white arrows). The presence of Si was confirmed with EDX (analyzed zones indicated by white vertical arrows).

Fig. 2 & 3: Why do you use single data of four years in one diagram (Fig. 2) and means with standard deviations in another one (Fig. 3)? I would recommend unifying the presentation of your results.

The objectives of the two figures are different. In the Figure 2, we want to show the seasonal and interannual variations (on four years) of the dissolved Si in the throughfall solution. In the figure 3, we want to show the seasonal variations over four years of the dissolved Si in the different soil compartments.

Fig. 5: Please correct the unit of dissolved Si.

**This was corrected.**

Fig. 6: As you can go full color in this journal I would recommend using different colors for data of the different sites.

Good idea so we did it.

Please refer to line 944: **Fig. 6:** ...For each pool and flux, values presented are those of the plots S1 (in green), S2 (in orange), and S3 (in blue), respectively...

Fig. 6: Please correct the values of soil Si pools (x103).

**This was corrected.**

Fig. 6: What about soil pH effects? I especially wonder at Si drainage values of S3 (0-10 cm). You should also give a more detailed discussion on this aspect.

Interesting suggestion, however too much soil parameters, mainly soil texture/structure, affect the drainage and it is complicated to discriminate the influence of each one. So we prefer to do not deal with the effect of pH on drainage in this paper. Maybe in another one. However we added a part in the discussion where we compare the drainage flux in our study site with other data in the literature, and we succinctly present the possible origin of the differences observed.

Please refer to line 916: The annual drainage flux ranged from 21 to 27 kg Si ha-1 y-1 in the three soils of the Montiers site which is higher than those measured in other beech forests by Bartoli (1983; 0 kg Si ha-1 y-1), Cornelis et al. (2010b, 6 kg Si ha-1 y-1), Sommer et al. (2013; 14 kg Si ha-1 y-1), and Clymans et al. (2011; 18 kg Si ha-1 y-1). The differences can result from multiple factors, i.e., topography, soil properties (texture, structure, pH), rainfall (level and intensity) and other climatic factors, and stand characteristics (tree species and age, stem density, ground vegetal cover...).

Contribution of tree fine roots to the Ssilicon cycle in a temperate forest ecosystem developed on three soil types: role of fine roots and litterfall recycling and influence of soil types

Marie-Pierre Turpault1, Christophe Calvaruso2, Gil Kirchen1, Paul-Olivier Redon3, Carine Cochet1

[revised manuscript text omitted]

| Plot                                   | Compartment                                       | Biomass pools                                 | Biomass                                   | Si content               | Si pools                                          | Si fluxes                                         |
|----------------------------------------|---------------------------------------------------|-----------------------------------------------|-------------------------------------------|--------------------------|---------------------------------------------------|---------------------------------------------------|
|                                        |                                                   | (t DM ha -1 )                      | increment                                 | (g kg -1 )    | (kg bo:1)                                         | (lea bar1 ver1)                                   |
|                                        | Lagues                                            | 2.0 (0.4) 3                                   | (t DM ha -1 yr -1 ) | 11 2 /1 0\h              | (kg ha -1 )                            | (kg ha -1 yr -1 )           |
|                                        | Leaves                                            | 3.8 (0.4) a                        | 3.8 (0.4) a                    | 11.3 (1.8) b  | 42.7 (4.3) b                           | 42.7 (4.3) b                           |
|                                        | Branches/twigs with bark                          | 0.3 (0.2) a                                   | 0.3 (0.2) a                               | 1.1 (0.3) a              | 0.3 (0.2) a                                       | 0.3 (0.2) a                                       |
|                                        | Buds, beechnuts, fruit capsules  Total litterfall | 1.1 (1.1) a 5.2 (1.1) a | 1.1 (1.1) a                               | 2.4 (1.0) a   | 1.8 (0.9) a
44.8 (5.1) b | 1.8 (0.9) a
44.8 (5.1) b |
|                                        | Organic horizons                                  | 11.5 (2.0) °                                  | 5.2 (1.1) a                    | 21.4 (1.6) a             | 246.4 (53.1) a                         | 44.6 (5.1)                                        |
|                                        | Small wood                                        | 7.5 (1.9) *                                   |                                           | 0.8 (0.3) a   | 6.5 (3.5) °                                       |                                                   |
|                                        | Forest floor                                      | 19.0 (2.7) °                                  |                                           | 0.8 (0.3)                | 252.9 (53.1) a                         |                                                   |
|                                        | Stem bark                                         | 5.5 (0.7) °                                   | 0.5 (0.0) b                               | 1.70 (0.33) a            | 9.4 (1.2) °                                       | 0.65 (0.03) b                                     |
|                                        | Stem wood                                         | 84.8 (11.7) ab                                | 6.4 (0.3) b                    | 0.05 (0.00) a            | 4.0 (0.5) a                            | 0.30 (0.02) 3                                     |
|                                        | Small branches (B+W)                              | 18.7 (2.5) ab                                 | 1.2 (0.1) b                    | 0.40 (0.05) °            | 7.4 (1.0) °                                       | 0.49 (0.03) b                                     |
| <del>S1 :</del>                        | Medium branches (B+W)                             | 10.2 (1.8) ab                                 | 1.1 (0.1) b                    | 0.26 (0.04) a | 2.6 (0.5) ab                                      | 0.29 (0.02) b                          |
| Dystric                                | Coarse branches (B+W)                             | 5.1 (1.1) ab                                  | 0.8 (0.1) ab                              | 0.13 (0.04)              | 0.7 (0.1) ab                                      | 0.23 (0.02)
0.10 (0.01) b                      |
| Cambisol                               | Aboveground biomass                               | 125.8 (17.9) ab                               | 10.0 (0.5) b                              | 0.13 (0.04)              | 24.1 (3.3) ab                                     | 1.82 (0.10) b                                     |
| Cumbison                               | Fine roots (0-10 cm)                              | 3.2 (0.8) a                                   | 3.5 (0.9) °                               | 12.8 (2.3) b             | 39.5 (7.5) °                                      | 43.9 (8.3) a                           |
|                                        | Fine roots (10-30 cm)                             | 2.9 (1.1) °                                   | 3.2 (1.2) °                               | 15.0 (2.3)°              | 43.9 (6.6) b                           | 48.8 (7.3) b                                      |
|                                        | Fine roots (30-60 cm)                             | 0.9 (0.6) a                        | 1.0 (0.7) a                    | 12.3                     | 10.5                                              | 11.7                                              |
|                                        | Fine roots (60-90 cm)                             | 0.4 (0.1) a                        | 0.4 (0.1) a                               | 12.7                     | 4.7                                               | 5.2                                               |
|                                        | Total fine roots (0-90 cm)                        | 7.3 (1.8) a                        | 8.0 (2.0)                                 | 12.7                     | 98.7 (13.5) b                          | 109.5 (15.0) b                                    |
|                                        | Total coarse roots                                | 24.4 (3.5) a                       | 2.83 (0.47) a                  | 0.11 (0.15) a            | 2.66 (0.39) b                                     | 0.31 (0.05) b                                     |
|                                        | Exploitation residues AG                          | 24.4 (3.3)                                    | 1.3                                       | 0.33                     | 2.00 (0.39)                                       | 0.42                                              |
|                                        | Exploitation residues BG                          |                                               | 1.1                                       | 0.11 (0.15) °            |                                                   | 0.12                                              |
|                                        | Total exploitation residues                       |                                               | 2.4                                       | 0.11 (0.13)              |                                                   | 0.54                                              |
|                                        | Harvests                                          |                                               | 4.4                                       | 0.16                     |                                                   | 0.71                                              |
|                                        | Leaves                                            | 4.1 (0.5) a                                   | 4.1 (0.5) a                    | 8.9 (1.6) ab             | 35.4 (2.8) ab                                     | 35.4 (2.8) ab                                     |
|                                        | Branches/twigs with bark                          | 0.6 (0.4) a                                   | 0.6 (0.4) a                               | 0.9 (0.2) *              | 0.4 (0.2) a                                       | 0.4 (0.2) a                            |
|                                        | Buds, beechnuts, fruit capsules                   | 1.3 (1.1) °                                   | 1.3 (1.1) °                               | 3.4 (1.9) a   | 3.0 (0.5) b                            | 3.0 (0.5) b                                       |
|                                        | Total litterfall                                  | 6.0 (1.1) a                        | 6.0 (1.1) a                    | 3.4 (1.3)                | 38.7 (3.1) ab                                     | 38.7 (3.1) ab                                     |
|                                        | Organic horizons                                  | 9.6 (1.4) a                        | 0.0 (1.1)                                 | 17.6 (0.8) a             | 174.2 (32.8) ab                                   | 36.7 (3.1)                                        |
|                                        | Small wood                                        | 2.6 (1.2) a                        |                                           | 1.8 (1.1) a              | 3.9 (1.3) a                            |                                                   |
|                                        | Forest floor                                      | 12.5 (0.6) a                                  |                                           | 1.0 (1.1)                | 178.1 (32.6) ab                                   |                                                   |
|                                        | Stem bark                                         | 6.1 (0.2) a                        | 0.4 (0.0) ab                              | 1.53 (0.28) a            | 9.3 (0.3) a                                       | 0.39 (0.04) a                                     |
|                                        | Stem wood                                         | 109.9 (3.8) b                                 | 5.0 (0.6) ab                              | 0.05 (0.00) a            | 5.1 (0.2) a                                       | 0.23 (0.02) a                                     |
| <del>\$2 :</del>
Eutric
Cambisol | Small branches (B+W)                              | 20.8 (0.7) b                                  | 0.8 (0.1) ab                              | 0.38 (0.08) a | 7.9 (0.3) a                            | 0.31 (0.04) ab                                    |
|                                        | Medium branches (B+W)                             | 15.2 (0.6) b                       | 1.0 (0.1) ab                              | 0.23 (0.05) a | 3.5 (0.1) b                                       | 0.23 (0.02) ab                                    |
|                                        | Coarse branches (B+W)                             | 9.8 (0.6) b                                   | 0.9 (0.1) b                    | 0.10 (0.03) a            | 1.0 (0.1) b                                       | 0.09 (0.01) ab                                    |
|                                        | Aboveground biomass                               | 164.2 (5.7) b                                 | 8.0 (0.9) ab                              | 0.10 (0.00)              | 26.9 (0.9) b                                      | 1.25 (0.13) ab                                    |
|                                        | Fine roots (0-10 cm)                              | 4.6 (2.1) a                                   | 5.1 (2.4) a                               | 9.6 (2.9) ab             | 44.5 (13.9) a                                     | 49.4 (15.4) a                                     |
|                                        | Fine roots (10-30 cm)                             | 4.5 (1.8) a                                   | 5.0 (1.9) a                    | 8.2 (1.6) b              | 37.0 (7.1) b                                      | 41.1 (7.8) b                                      |
|                                        | Fine roots (30-60 cm)                             | 1.2 (0.7) a                                   | 1.3 (0.8) a                    | 7.5                      | 8.7                                               | 9.7                                               |
|                                        | Fine roots (60-90 cm)                             | 0.4 (0.1) a                                   | 0.5 (0.1) a                               | -                        | -                                                 | -                                                 |
|                                        | Total fine roots (0-90 cm)                        | 10.6 (4.1) a                                  | 11.7 (4.5)                                |                          | 90.2 (20.8) b                                     | 100.1 (23.1) b                                    |
|                                        | Total coarse roots                                | 32.3 (1.2) b                                  | 4.08 (0.16) b                             | 0.05 (0.08) a            | 1.51 (0.05) a                                     | 0.19 (0.01) a                                     |
|                                        | Exploitation residues AG                          | , , , , , , , , , , , , , , , , , , ,         | 1.4                                       | 0.31                     | , , , , , , , , , , , , , , , , , , ,             | 0.43                                              |
|                                        | Exploitation residues BG                          |                                               | 1.4                                       | 0.05 (0.08) a            |                                                   | 0.06                                              |
|                                        | Total exploitation residues                       |                                               | 2.8                                       | ,                        |                                                   | 0.50                                              |
|                                        | Harvests                                          |                                               | 4.9                                       | 0.15                     |                                                   | 0.72                                              |
|                                        | Leaves                                            | 4.0 (0.4) a                                   | 4.0 (0.4) a                               | 5.6 (1.3) a              | 22.2 (3.1) a                                      | 22.2 (3.1) a                                      |
|                                        | Branches/twigs with bark                          | 0.5 (0.3) a                                   | 0.5 (0.3) a                               | 0.7 (0.1) a   | 0.3 (0.2) a                                       | 0.3 (0.2) a                                       |
|                                        | Buds, beechnuts, fruit capsules                   | 1.2 (0.9) a                                   | 1.2 (0.9) a                               | 3.2 (1.6) a   | 2.6 (0.5) ab                                      | 2.6 (0.5) ab                                      |
|                                        | Total litterfall                                  | 5.7 (1.0) a                                   | 5.7 (1.0) a                    | . ,                      | 25.2 (3.4) a                                      | 25.2 (3.4) a                           |
|                                        | Organic horizons                                  | 8.8 (1.5) a                                   | · ,                                       | 16.9 (1.4) a             | 151.3 (22.6) b                                    | , , ,                                             |
| <del>S3</del>                          | Small wood                                        | 1.9 (2.4) a                                   |                                           | 1.3 (0.7) a              | 4.4 (5.7) °                                       |                                                   |
| Rendzic                                | Forest floor                                      | 10.9 (2.8) a                       |                                           | ,                        | 154.3 (25.3) a                                    |                                                   |
| Leptosol                               | Stem bark                                         | 6.8 (0.6) a                                   | 0.3 (0.0) a                               | 1.34 (0.27) a            | 9.1 (0.8) a                            | 0.41 (0.05) ab                                    |
|                                        | Stem wood                                         | 80.1 (8.3) a                       | 3.9 (0.5) a                    | 0.06 (0.03) a            | 5.0 (0.5) a                            | 0.24 (0.03) a                                     |
|                                        | Small branches (B+W)                              | 15.0 (1.4) a                       | 0.6 (0.1) a                    | 0.29 (0.04) a | 4.3 (0.4) a                            | 0.18 (0.02) a                                     |
|                                        | Medium branches (B+W)                             | 8.6 (1.4) a                                   | 0.6 (0.1) a                               | 0.19 (0.04) a            | 1.6 (0.3) a                                       | 0.11 (0.02) a                                     |
|                                        | Coarse branches (B+W)                             | 4.6 (1.0) °                                   | 0.4 (0.1) a                               | 0.10 (0.03)              | 0.5 (0.1) °                                       | 0.04 (0.01) °                                     |
|                                        | Course Dialienes (DIVV)                           | 1.0 (1.0)                                     | J (J1)                                    | 0.10 (0.03)              | 3.3 (0.1)                                         | 3.07 (J.U1)                                       |

| Aboveground biomass         | 115.2 (12.8) a         | 5.8 (0.8) a              |               | 20.5 (2.1) a  | 0.98 (0.13) a           |
|-----------------------------|------------------------|--------------------------|---------------|--------------------------|-------------------------|
| Fine roots (0-10 cm)        | 5.1 (1.4) a            | 5.6 (1.6) a              | 7.8 (2.2) a   | 43.5 (14.1) a            | 48.3 (15.6) a           |
| Fine roots (10-30 cm)       | 3.6 (1.6) a | 4.0 (1.8) a              | 4.9 (0.8) a   | 17.6 (3.0) a             | 19.6 (3.3) a |
| Fine roots (30-60 cm)       | NS                     | NS                       | -             | -                        | -                       |
| Fine roots (60-90 cm)       | NS                     | NS                       | -             | -                        | -                       |
| Total fine roots (0-30 cm)  | 8.7 (3.0) a | 9.6 (3.3)                |               | 61.2 (16.0) a            | 67.9 (17.7)             |
| Total coarse roots          | 26.0 (3.0) a           | 3.09 (0.44) a | 0.06 (0.05) a | 1.62 (0.19) a | 0.19 (0.03)             |
| Exploitation residues AG    |                        | 1.1                      | 0.24          |                          | 0.27                    |
| Exploitation residues BG    |                        | 1.0                      | 0.06 (0.05) a |                          | 0.06                    |
| Total exploitation residues |                        | 2.1                      |               | •                        | 0.33                    |
| Harvests                    |                        | 3.9                      | 0.15          |                          | 0.57                    |

**Table 3:** Mean total Si content and pool in the fine earth fraction of the three soils of the Montiers site at different depths. Standard deviation values are given in brackets. Values with different letters are significantly different according to a Kruskal-Wallis test at the threshold P value level of 0.05 (soil effect).

| Soil type       | Compartment   | Total Si content      | Total Si pool            |
|-----------------|---------------|-----------------------|--------------------------|
|                 |               | (g kg -1 ) | (t ha -1 )    |
| <del>S1 :</del> | 0-10 cm       | 305 (13)-             | 297 (33)          |
| Dystric         | 10-30 cm      | 313 (9)ª              | 708 (50)          |
| Cambisol        | 30-60 cm      | 296 (18)       | 1 301 (422)       |
|                 | 60-90 cm      | 230 (28)       | 858 (80) ⊆    |
|                 | Total 0-90 cm |                       | 3 164 (487) b |
| <del>S2 :</del> | 0-10 cm       | 361 (11)       | 411 (30) -        |
| Eutric          | 10-30 cm      | 360 (13)       | 791 (127)         |
| Cambisol        | 30-60 cm      | 295 (62)       | 871 (290)         |
|                 | 60-90 cm      | 224 (28)       | 348 (117)         |
|                 | Total 0-90 cm |                       | 2 421 (410) b |
| <del>S3 :</del> | 0-10 cm       | 287 (27)ª             | 233 (18)ª                |
| Rendzic         | 10-30 cm      | 276 (23)ª             | 427 (27)ª                |
| Leptosol        | 30-60 cm      | 175 (37)-             | 42 (27) <del>-</del>     |
|                 | 60-90 cm      | 144 (39)ª             | 27 (8) a          |
|                 | Total 0-90 cm | •                     | 720 (38) a    |

**Table 4:** Si content and fluxes in the ZTL (Zero Tension Lysimeters) and TL (Tension Lysimeters) solutions of the three soils of the Montiers site. Standard deviation values are given in brackets. Values with different letters are significantly different according to a Kruskal-Wallis test at the threshold P value level of 0.05 (soil effect).

[revised manuscript text omitted]

---

## Author Comment (AC2) · 13 Feb 2018

Dear colleague,

First of all we would like to thank you (reviewer 2) for the relevance of your comments. In addition with those of the reviewer 1, this has considerably improved the quality of the manuscript. Notably, from your comments, we realized that some important parts of the manuscript were not enough clear (we shall be taking a closer look on this aspect below) and that some relevant literature was missing (as also specified by reviewer 1). In consequence we did important modifications in the manuscript to clarify our approach and objectives, as well as to introduce the relevant literature. Please find below, our responses (in blue) to each of your remark (in black), and the location of the modifications brought to the text (in blue).

In this manuscript, Marie-Pierre Turpault et al. address the role of fine roots, litterfall and soil type on Si cycling in a temperate forest system. The main and surprising novelty of this manuscript lies in the observation that fine roots actually are a large Si reservoir in forest soils. To my knowledge, no other authors have ever performed a similarly detailed exercise to quantify the amount of Si in the forest root system. Quantifying root biomass is difficult, and these authors have done a tremendous effort to take on this challenge. While this is a finding worth publishing in itself, I have strong reservations regarding the mass balance the authors have made for the whole forest ecosystem. These reservations are mainly related to the applied methodology to analyse for Si in the soil system, which is inadequate to assess the complicated Si cycle in the soil, as it does not distinguish any pedogenic nor biogenic Si fractions from the abundant mineral fractions. This prevents to make any major conclusions on the role of soil type in the Si mass balance, and also makes it difficult to assess the cycling of litterfall Si in soils, once dissolved. Multiple secondary pedogenic fractions are accumulated deeper in the soil. In conclusion, I am impressed with the root Si quantification the authors have performed, and I think that a focused manuscript emphasizing the importance of roots in the forest Si cycle is worthy of publication. I also think that a more focused manuscript would have a larger impact on the interested scientific community. The authors should either improve methodology if they want to address the full Si cycle in the forest, or far better emphasize the methodological shortfalls in their discussion, that prevent to make any statement on the full forest Si cycle, and focus on the interesting story of the roots. I will make more detailed comments below.

We agree with your remarks, the methodologies used in our study prevent to conclude on the whole Si budget in the ecosystem. But this was not the objective of our study and we understand by reading your review that our manuscript was not enough clear. In consequence, we drastically modified several sections to focus on the interesting findings of our study, as you suggested. The main changes made on the revised version of the manuscript are:

i) The title was modified as follows: "Contribution of tree fine roots to the silicon cycle in a temperate forest ecosystem developed on three soil types"

ii) The introduction was rewritten to focus on the Si cycle in forest ecosystems and on the possible contribution of fine roots to the Si cycle which introduce our study (see below in details).

iii) The discussion was partially rewritten to focus on the interesting results and discuss them in comparison with the literature (see below in details). Some speculative interpretations were deleted.

iv) A new paragraph was added at the end of the conclusion to introduce succinctly some future challenges necessary to approach the whole Si cycle in forests (see below).

v) A figure summarizing the main findings of our study with comparisons with other studies was added in the conclusion. This clearly reveals the contribution of our study.

Please refer to line 951: **Fig. 7:** Summary scheme of the main findings of this study (TS) and comparison with other studies (L).

[Figure]

The approaches used in our study allow us to:

- To determine that a mean of 71% of the Si accumulated by trees returns to the soil via fine root decomposition, widely overpassing the contribution of litterfall (28%). That reveals that almost all the Si accumulated in trees is recycled.
- To assess the Si drainage in the soil: between 20 and 27 kg ha$^{-1}$ y$^{-1}$ for the three soil types with a great part in the organic horizons (biological origin), between 10 and 13 kg ha$^{-1}$ y$^{-1}$,
- To compare the two soil outputs, the leaching and the tree uptake (between 157 and 95 kg ha$^{-1}$ y$^{-1}$ for the three soil types. On average 78% to 88% of the Si produced in the soil were taken up by trees
- To discriminate the net Si production and consumption in each soil horizon and the seasonal dynamics of these fluxes in relation with biological activities,

These results coupled with other studies provide evidence to develop a strategy aiming to assess the whole Si budget in terrestrial ecosystems. In another paper...

Line 45: I am becoming a bit annoyed by all Si manuscripts starting with the same statement. Can we just accept that it is now common knowledge that there is a lot of Si in the Earth's crust, and that minerals dissolve. This manuscript is about forest Si cycling, and the role of biological processes in the Si cycle. This has been well described in several review papers over the last years (e.g. Conley, GBC, 2002, Volume 16; Cornelis et al., Biogeosciences, 2011, Volume 8; Struyf Conley, 2012, Biogeochemistry, Volume 107).

We agree with this remark and drastically modified the introduction to take into account all remarks of both reviewers. Some relevant literature was added.

Please refer to line 45: 1 Introduction:

It has recently been shown that intense biogeochemical cycling of Si occurs in the different terrestrial ecosystems, i.e., wetlands (Struyf et al., 2007; Emsens et al., 2016), grasslands (Blecker et al., 2006; White et al., 2012), tropical forests (Lucas et al., 1993, Alexandre et al., 1997) and temperate forests (Bartoli, 1983; Watteau and Villemin, 2001; Gerard et al, 2008; Cornelis et al., 2010a; Cornelis et al., 2011a; Sommer et al., 2006; Sommer et al., 2013). Several review papers well described that soil DSi is taken up by vascular plants and translocated into biogenic Si (BSi) under opal form which is deposited into the cell walls, cell luminas and intercellular spaces (Jones and Handreck, 1965; Conley et al., 2002; Cornelis et al, 2011b; Struyf and Conley, 2012). These structures are called phytoliths. Other important producers of biogenic Si are animals especially diatoms, sponges and testate amoebae (Struyf and Conley, 2012; Sommers et al., 2006; Puppe et al., 2014; Puppe et al., 2015).

According to Conley (2002), the annual fixation of DSi into terrestrial ecosystems has been estimated to range from 60 to 200 Tmoles. That represents 10 to 40 times more than yearly export DSi and suspended biogenic Si from the terrestrial geobiosphere to the coastal zone (Conley, 2002). Vegetation can thus be considered as a factory of BSi which returns to the soil as organic matter through biological recycling. Because BSi in general is more soluble than silicate minerals, BSi strongly contributes to the DSi pool (Fraysse et al., 2009 ; Cornelis and Delvaux, 2016).

Based on the assumption that the storage of Si is limited in roots (Bartoli and Souchier, 1986) and because fine root sampling and cleaning before analyses are long and tedious processes, studies in forest ecosystems mainly focus on the importance of litterfall recycling on the Si biogeochemical cycle without quantifying Si in the roots (Gérard et al., 2008; Cornelis et al., 2010a; Sommer et al., 2013).

However, Krieger et al (2017) recently showed that Si in deciduous trees (European beech, *Fagus sylvatica* and sycamore maple, *Acer pseudoplatanus*) generally precipitates as a thin layer (< 0.5 µm) around the cells, especially in roots and bark. These small-scale phytogenic Si was demonstrated to influence various soil and plant processes (Meunier et al., 2017 ; Puppe et al., 2017).

Considering the large amount of Si precipitates in roots (Krieger et al., 2017) and the rapid turnover of fine roots in forest ecosystems (approximately one year in beech forests in Europe; Brunner et al., 2013), we hypothesized that fine roots could significantly contribute to the input of BSi into the soil.

To test this hypothesis, we quantified during a four-year observation period (i) the total and annual accumulations of Si in stand belowground and abovegound biomasses while distinguishing annual and perennial compartments, ii) the Si input fluxes in the forest floor (litterfall and small woods, aboveground exploitation residues) and in the soil (fine roots and belowground exploitation residues). The study was led in a lowland (low lateral transfer of material) deciduous temperate forest developed on three soils, ranging from a shallow calcic soil to a deep acidic soil, with mull to acid mull humus. These humus forms quickly degrades, contain few soil particles and no root thus allowing to determine the DSi issued from the degradation of organic layers contrary to mor or moder humus forms (Sommer et al., 2006; Cornelis et al., 2010a). In addition, we monthly quantified in these ecosystems the Dsi inputs and outputs, i.e., rainfall, foliar leaching and drainage, in order to assess the seasonal dynamics of these fluxes induced by biological activities.

We also added a new paragraph in the conclusion to present some future challenges on the basis of our findings.

Please refer to line 721: Further research is needed in the mid-term (i) to assess the mineralisation speed of fine roots in the soil and the speed of transformation of the BSi of roots into DSi, (ii) determine the annual and seasonal fate of the Dsi issued from roots, between uptake, mineral precipitation, drainage, fixation by organisms, and (iii) quantify the vertical transfer of solid particulates between organic horizons and topsoil.

Line 57 and beyond: I really don't see why this is important to this manuscript. The division between accumulators, excluders and neutrals is anyway arbitrary, if based on concentration. The Si uptake of plants is also governed by external Si factors, such as its availability.

We agree, this sentence was deleted.

Line 62: Why also? You have not referred to forests before, so 'also' seems out of place here. What about wetlands, one of the most studied system in the biological Si cycle? If you provide a list, wetlands should be there.

We agree so this sentence was modified and expanded.

Please refer to line 46: It has recently been shown that intense biogeochemical cycling of Si occurs in the different terrestrial ecosystems, i.e., wetlands (Struyf et al., 2007; Emsens et al., 2016), grasslands (Blecker et al., 2006; White et al., 2012), tropical forests (Lucas et al., 1993, Alexandre et al., 1997) and temperate forests (Bartoli, 1983; Watteau and Villemin, 2001; Gerard et al, 2008; Cornelis et al., 2010a; Cornelis et al., 2011a; Sommer et al., 2006; Sommer et al., 2013).

Line 67: Here comes the first reference to later methodological issues. This statement is untrue. In recent years, methodologies have been developed that allow to distinguish pedogenic, reactive mineral and biogenic Si phases in soils (e.g. Barao et al. 2014, European Journal of Soil Science, 65, Barao et al., LO Methods, 13, 2015; Georgiadis et al., 2015, Soil Research 52).

This sentence was deleted.

Line 76: soap? Probably sap is meant.

Sorry for the mistake, the sentence was rewritten.

Please refer to line 61: Based on the assumption that the storage of Si is limited in roots (Bartoli and Souchier, 1986) and because fine root sampling and cleaning before analyses are long and tedious processes, studies in forest ecosystems mainly focus on the importance of litterfall recycling on the Si biogeochemical cycle without quantifying Si in the roots (Gérard et al., 2008; Cornelis et al., 2010a; Sommer et al., 2013).

Line 90: the second hypothesis is not really novel, Cornelis (et al.) (see also reference list of paper) has already published multiple papers on this issue. In these papers, he shows that methodology is quintessential in addressing the complicated soil type-Si cycling coupling, and the applied method that does not distinguish any secondary soil Si fractions from minerals is inadequate to address the hypothesis. Line 104: Why? If you want to address the whole forest Si cycle, the soil is of the essence. If you do not apply best available methods (see above) here, then you start with a strong handicap. Line 209: total fusion is unable to provide sufficiently detailed results for assessing soil Si cycling, where multiple secondary Si fractions form that are actually essential in the whole ecosystem Si balance.

We agree with all of these remarks. See our general explanation above and the changes made to the manuscript.

Line 93-95: awkward wording, consider revising

This sentence was deleted.

Line 113: without any reference to these networks, their relevance is not clear.

This sentence was expanded.

Please refer to line 154: The Montiers site is part of different national and international research networks, i.e., SOERE (Long-lasting observation and experimentation for the research on environment)-OPE (Perennial Environment Observatory; http://www.andra.fr/ope/index.php?lang=en&Itemid=127) and F-ORE-T (Functioning of Forest Ecosystems; http://www.gip-ecofor.org/f-ore-t/), and AnaEE (Analysis and Experimentations on Ecosystems; https://www.anaee.com/).

General: ceramic cups? Why not plastic? Can ceramic cups potentially add Si to solution? Has this been tested?

This material is used for many decades in our different experimental forest sites and was of course experimentally tested to ensure the absence of release of elements including Si by the ceramic. Cornelis et al. used the same equipment.

General: I miss any comparison with recent studies that have also made forest Si efflux quantifications. How do your fluxes compare to e.g. Struyf et al. (2010, Nature Communications, 1 and Clymans et al. 2013, Biogeochemistry, 11). I think a section putting the observed effluxes in the context of other literature, would be far more interesting than the attempt to discuss the role of soil Si processes in the forest Si cycle, given the flawed methodology here. The suction cups do provide an idea of the leakage, and focus should be on how this compares to root turnover and forest Si uptake. In general, I have the impression that Si efflux in this paper is rather low compared to other studies. Is this maybe because these are young forests? Or due to management?

- We agree that these remarks and thus written a new paragraph dedicated to compare our drainage flux with tree uptake, and with drainage in other studies in similar conditions.

Please refer to line 630: The annual drainage flux ranged from 21 to 27 kg Si ha$^{-1}$ y$^{-1}$ in the three soils of the Montiers site which is higher than those measured in other beech forests by Bartoli (1983; 0 kg Si ha$^{-1}$ y$^{-1}$), Cornelis et al. (2010b, 6 kg Si ha$^{-1}$ y$^{-1}$), Sommer et al. (2013; 14 kg Si ha$^{-1}$ y$^{-1}$), and Clymans et al. (2011; 18 kg Si ha$^{-1}$ y$^{-1}$). The differences can result from multiple factors, i.e., topography, soil properties (texture, structure, pH), rainfall (level and intensity) and other climatic factors, and stand characteristics (tree species and age, stem density, ground vegetal cover...). In our study, the Si leached out of the soil profile was negligible compared to the Si taken up by trees, i.e., ratios of 1:4 to 1:7 in RL and DC, respectively. If we deduce the part of Si leached from the organic horizons, these ratios rise to about 1:5 to 1:22 in RL and DC. Because biogenic Si in general is more soluble than lithogenic or pedogenic Si (Fraysse et al., 2009 ; Cornelis and Delvaux, 2016), very few of the Si leached within the soil profile directly results from the dissolution of soil minerals, as demonstrated in other studies in temperate forests (Bartoli, 1983; Watteau and Villemin, 2001; Gerard et al, 2008; Cornelis et al., 2010a; Cornelis et al., 2011a; Sommer et al., 2006; Sommer et al., 2013).

- In addition, the interesting results of Struyf et al. (2010) was also discussed, in another section, dealing with the influence of forest deforestation on Si cycle, an important point that we neglected in the original version of the manuscript.

Please refer to line 648: However, Struyf et al. (2010) observed that land use is the most important controlling factor of Si mobilization in European watersheds. These authors showed that deforestation and conversion to agricultural land or other land uses leads to a twofold to threefold decrease in baseflow delivery of Si.

- Finally, other relevant literature was added in the different sections to support our assumption or compare our data with other studies.

Please refer to line 65: However, Krieger et al (2017) recently showed that Si in deciduous trees (European beech, *Fagus sylvatica* and sycamore maple, *Acer pseudoplatanus*) generally precipitates as a thin layer (< 0.5 μm) around the cells, especially in roots and bark. These small-scale phytogenic Si was demonstrated to influence various soil and plant processes (Meunier et al., 2017 ; Puppe et al., 2017).

Please refer to line 540: The Si content in beech fine roots was very higher (2 to 6 times) than that measured by Maguire et al. (2017) for another deciduous species, i.e. sugar maple (*Acer saccharum*) but in a cooler environment. Besides Maguire et al. (2017) demonstrated in this study that increased soil freezing significantly lowers the Si content of sugar maple fine roots.

Please refer to line 572: The higher rate of soil pollution in the study of Cornelis et al. (2010a) can be explained by the presence of a thick Oh layer in the moder that was in direct contact with the superficial soil layer and was characterized by an intense mixing of degraded organic matter with soil particles, induced by biological activities, mainly bioturbation by earthworms in these soils (Lavelle, 1988).

Please refer to line 592: This flux is likely the solid particulate migration toward the topsoil layer, as demonstrated by Ugolini et al. (1977). These authors observed that organic particles containing notably silicon were predominant in the migrant material in the upper soil horizons. In our study, the solid particulate migration from the organic horizons to the topsoil may consist of the colloid transport of amoebae (Harter et al., 2000) or the transport of phytoliths (Fishkis et al. 2010). These latter observed, though a field study using fluorescent labelling, that the downrard transport distance of phytoliths after one year was 3.99± 1.21 cm for a Cambisol with a preferential translocation of small-sized phytoliths.

Please refer to line 681: The concentration of dissolved Si in the soil is known to influence opal formation in plants (Cornelis et al., 2010b) but phytolith production seems to be more affected by the phylogenetic position of a plant than by environmental factors (Hodson et al., 2005). For example, these authors demonstrated through meta-analysis of the data, that in general ferns, gymnosperms and angiosperms accumulated less Si in their shoots than non-vascular plant species and horsetails.

Please refer to line 687: Silicon plays several physiological and ecological functions in leaves and roots, such as an involvement in the detoxification of aluminum, oxalic acid, and heavy metals, in the regulation of ion balance, in the reduction of hydric, salt, and temperature stresses (Currie and Perry, 2007; Meunier et al., 2017). They also contribute to the optimization of photosynthesis by gathering and scattering light in the leaves, confer mechanical support and tissue rigidity, and facilitate pollen release, germination, and tube growth (Bauer, Elbaum, & Weiss, 2011; Currie and Perry, 2007; Gal et al.,, 2012). In addition to these physiological functions, Si has also ecological significance by protecting plants against herbivores and phytopathogens (Currie and Perry, 2007; Lins et al., 2002).

Line 451: Consumption during autumn? Rather contradictory to forest growth in spring and summer? Pedogenic processes at play? Also in apparent contrast to later references to a net Si efflux in fall (Line 536)?

The term "consumption" does not necessarily implies tree uptake. As you suggested this Si consumption in fall was probably induced by pedogenic processes such as precipitation of secondary minerals as explained in line 626: In the deeper layer, the dissolved Si budget was significantly negative and likely corresponded to mineral precipitation, induced by a decrease of Si drainage with the depth, as observed by Sommer et al. (2013).
To avoid misunderstandings, the term "consumption" was replaced in the whole manuscript by the

term "immobilization". In addition, the term "accumulation" is now used for the elements immobilized in tree biomass (instead of immobilization).

Line 451 (in the first version of the manuscript) refers to the deeper soil layer "In the 60-90 cm layer of plot S1, we observed the immobilization of dissolved Si (Figure 5)" while line 536 (in the first version of the manuscript) refers to all soil layers except the deeper one. This last sentence was modified to be clearer.

Please refer to line 619: A peak of net Si production was observed during fall (except in the deeper soil layer ; Figure 3), which was probably due to an increase in Si production through the decomposition of dead roots.

Line 462: I would not use "global" in this local ecosystem context

We agree. This paragraph was completely deleted.

Line 515-522: I don't understand. First a significant accumulation is discussed, but a few lines below limited accumulation is mentioned?

We agree that this paragraph was not clear so we rewritten it and added some relevant literature to support our assumptions.

Please refer to line 589: During the study period (2012-2015), the Si input in the organic horizons via litterfall were primarily higher than the Si output via soluble transport (assessed in ZTL solutions under the forest floor) for the three soils. This net flux of Si should have induced the accumulation of Si in the organic horizons, what we did not observe in the four years of the study. This suggests the existence of another output flux which was not quantified in our study. This flux is likely the solid particulate migration toward the topsoil layer, as demonstrated by Ugolini et al. (1977). These authors observed that organic particles containing notably silicon were predominant in the migrant material in the upper soil horizons. In our study, the solid particulate migration from the organic horizons to the topsoil may consist of the colloid transport of amoebae (Harter et al., 2000) or the transport of phytoliths (Fishkis et al. 2010). These latter observed, though a field study using fluorescent labelling, that the downrard transport distance of phytoliths after one year was 3.99± 1.21 cm for a Cambisol with a preferential translocation of small-sized phytoliths.

Line 547: I don't understand how you can state the biological origin, if you apply total fusion.

This sentence was not clear so we deleted it.

[revised manuscript text omitted]

---

## Author Comment (AC3) · 13 Feb 2018

Dear colleague,

Firstly, we would like to thank you (reviewer 1) for this careful review of our paper and the relevance of your comments and suggestions, notably regarding the literature proposed. This has considerably improved the manuscript. Below, presented are your remarks (in black), and our responses and the location of the modifications brought to the text (in blue).

General comments

First of all I would like to thank the Associate Editor in charge for the opportunity to review the present manuscript entitled 'Silicon cycle in a temperate forest ecosystem: role of fine roots and litterfall recycling and influence of soil types'. The authors (Turpault et al.) analyzed silicon (Si) cycling in three temperate forest ecosystems with different soil types (Dystric Cambisol, Eutric Cambisol, Rendzic Leptosol). In this context, Turpault et al. aimed to unravel the specific role of fine roots and soil properties on Si cycling. The authors found that fine roots potentially play an important role in Si cycling as their Si concentration seems to be comparable to the Si concentration of leaves. Furthermore, Turpault et al. found the Si concentrations in fine roots and leaves to be dependent on the concentration of dissolved Si in the soils. Turpault et al. concluded from their results that biological processes play a predominant role in Si cycling of the studied sites. In my opinion the article of Turpault et al. generally is of interest for the readers of BIOGEOSCIENCES. However, I identified several shortcomings of the manuscript which should be addressed before potential publication.

In general, the authors should:

- Use units following the rules of the 'International System of Units' (e.g., g kg-1 and not g.kg-1; Please check the whole manuscript on that because in almost all units these dots were used)

The units were modified in the whole revised version of the manuscript to follow the rules of the 'International System of Units'.

- add some literature that is most relevant to their article from my point of view and will help to present a more appropriate discussion of their results (please see my specific comments to the single sections below)

Some relevant literature, including the references proposed by both reviewers, was added in the manuscript. Besides the introduction was considerably modified.

See below the detail.

- reconsider the presentation of their results (in the current form I found reading of some subsections of the results section quite exhausting as the authors only repeat the data one-on-one as given in the Tables; I also miss a 'joining' of data, e.g., by some simple correlation analyses).

The whole result section was rewritten to improve its reading.

See an example in the result section.

- rework some subsections (in the current manuscript there are some redundant passages in different subsections; Additionnally, specific paragraphs should be displaced to corresponding subsections, e.g., methods should be given only in the Materials and Methods section)

Some subsections or paragraphs were deleted or displaced in the appropriate sections. See below the detail.

Please find corresponding details on the different subsections listed below. I am really looking forward to reading the revised manuscript.

Abstract

l.16: Actually, there is at least one publication where Si pools of fine roots were determined (Maguire, T. J., Templer, P. H., Battles, J. J., & Fulweiler, R. W. (2017). Winter climate change and fine root biogenic silica in sugar maple trees (Acer saccharum): Implications for silica in the Anthropocene. Journal of Geophysical Research: Biogeosciences, 122(3), 708-715). So, please relativize your statement ('rare is known. . .', for example).

Thanks for this remark. We thus modified the sentence in the abstract. The main results and conclusions of the work of Maguire et al. (2017) was introduced in the discussion section.

Please refer to line 21: However, to date, rare is known about the specific role of fine roots.

Please refer to line 540: The Si content in beech fine roots was very higher (2 to 6 times) than that measured by Maguire et al. (2017) for another deciduous species, i.e. sugar maple (*Acer saccharum*) but in a cooler environment. Besides Maguire et al. (2017) demonstrated in this study that increased soil freezing significantly lowers the Si content of sugar maple fine roots.

l.21: I would recommend using DC, EC and RL for Dystric Cambisol, Eutric Cambisol and Rendzic Leptosol, respectively, instead of S1, S2, S3. If you follow my recommendation, please change this within the whole manuscript (and figures and tables).

DC, EC, and RL are now used in the manuscript (text, tables and figures) instead of Dystric Cambisol, Eutric Cambisol and Rendzic Leptosol and instead of S1, S2, and S3.

Introduction

As suggested by reviewer 2, the introduction was rewritten to focus on forest ecosystems and on the main objective of the study.

Please refer to line 45: 1 Introduction:

It has recently been shown that intense biogeochemical cycling of Si occurs in the different terrestrial ecosystems, i.e., wetlands (Struyf et al., 2007; Emsens et al., 2016), grasslands (Blecker et al., 2006; White et al., 2012), tropical forests (Lucas et al., 1993, Alexandre et al., 1997) and temperate forests (Bartoli, 1983; Watteau and Villemin, 2001; Gerard et al, 2008; Cornelis et al., 2010a; Cornelis et al., 2011a; Sommer et al., 2006; Sommer et al., 2013). Several review papers well described that soil DSi is taken up by vascular plants and translocated into biogenic Si (BSi) under opal form which is deposited into the cell walls, cell luminas and intercellular spaces (Jones and Handreck, 1965; Conley et al., 2002; Cornelis et al, 2011b; Struyf and Conley, 2012). These structures are called phytoliths. Other important

producers of biogenic Si are animals especially diatoms, sponges and testate amoebae. (Struyf and Conley, 2012; Sommers et al., 2006; Puppe et al., 2014; Puppe et al., 2015).

According to Conley (2002), the annual fixation of DSi into terrestrial ecosystems has been estimated to range from 60 to 200 Tmoles. That represents 10 to 40 times more than yearly export DSi and suspended biogenic Si from the terrestrial geobiosphere to the coastal zone (Conley, 2002). Vegetation can thus be considered as a factory of BSi which returns to the soil as organic matter through biological recycling. Because BSi in general is more soluble than silicate minerals, BSi strongly contributes to the DSi pool (Fraysse et al., 2009 ; Cornelis and Delvaux, 2016).

Based on the assumption that the storage of Si is limited in roots (Bartoli and Souchier, 1986) and because fine root sampling and cleaning before analyses are long and tedious processes, studies in forest ecosystems mainly focus on the importance of litterfall recycling on the Si biogeochemical cycle without quantifying Si in the roots (Gérard et al., 2008; Cornelis et al., 2010a; Sommer et al., 2013).

However, Krieger et al (2017) recently showed that Si in deciduous trees (European beech, *Fagus sylvatica* and sycamore maple, *Acer pseudoplatanus*) generally precipitates as a thin layer (< 0.5 µm) around the cells, especially in roots and bark. These small-scale phytogenic Si was demonstrated to influence various soil and plant processes (Meunier et al., 2017 ; Puppe et al., 2017).

Considering the large amount of Si precipitates in roots (Krieger et al., 2017) and the rapid turnover of fine roots in forest ecosystems (approximately one year in beech forests in Europe; Brunner et al., 2013), we hypothesized that fine roots could significantly contribute to the input of BSi into the soil.

To test this hypothesis, we quantified during a four-year observation period (i) the total and annual accumulations of Si in stand belowground and abovegound biomasses while distinguishing annual and perennial compartments, ii) the Si input fluxes in the forest floor (litterfall and small woods, abovegound exploitation residues) and in the soil (fine roots and belowground exploitation residues). The study was led in a lowland (low lateral transfer of material) deciduous temperate forest developed on three soils, ranging from a shallow calcic soil to a deep acidic soil, with mull to acid mull humus. These humus forms quickly degrades, contain few soil particles and no root thus allowing to determine the DSi issued from the degradation of organic layers contrary to mor or moder humus forms (Sommer et al., 2006; Cornelis et al., 2010a). In addition, we monthly quantified in these ecosystems the Dsi inputs and outputs, i.e., rainfall, foliar leaching and drainage, in order to assess the seasonal dynamics of these fluxes induced by biological activities.

l.48: Please change '. . .Si in soils also had a biogenic origin. . .' to '. . .Si in soils can also be of biogenic origin. . .'.

This sentence was deleted.

l.58: Please change '. . .transpiration have also influenced. . .' to '. . .transpiration also influence. . .'.

This sentence was deleted.

l.65: I would recommend using the classification of BSi pools as given in Puppe et al. (2015) (Puppe, D., Ehrmann, O., Kaczorek, D., Wanner, M., & Sommer, M. (2015). The protozoic Si pool in temperate forest ecosystems  T Quantification, abiotic controls ˇ and interactions with earthworms. Geoderma, 243, 196-204), i.e., zoogenic, phytogenic, microbial and protistic Si pools.

This sentence was deleted.

l.70: From my point of view you should add Meunier et al. (2017) (Meunier, J. D., Barboni, D., Anwar-ul-Haq, M., Levard, C., Chaurand, P., Vidal, V., Grauby, O., Huc, R., Laffont-Schwob, I., Rabier, J., and Keller, C.: Effect of phytoliths for mitigating water stress in durum wheat, New Phytol., 215, 229–239, https://doi.org/10.1111/nph.14554, 2017) and Puppe et al. (2017) (Puppe, D., Höhn, A., Kaczorek, D., Wanner, M., Wehrhan, M., & Sommer, M. (2017). How big is the influence of biogenic silicon pools on short-term changes in water-soluble silicon in soils? Implications from a study of a 10-year-old soil–

plant system. Biogeosciences, 14(22), 5239-5252) here as these articles also show the importance especially of small-scale phytogenic Si.

We agree that this literature is of importance so we added it in the revised version of the manuscript.

Please refer to line 65: However, Krieger et al (2017) recently showed that Si in deciduous trees (European beech, *Fagus sylvatica* and sycamore maple, *Acer pseudoplatanus*) generally precipitates as a thin layer (< 0.5 µm) around the cells, especially in roots and bark. These small-scale phytogenic Si was demonstrated to influence various soil and plant processes (Meunier et al., 2017 ; Puppe et al., 2017).

l.76: I guess you mean 'sap' instead of 'soap' here, right?

This sentence was deleted.

l.93: Please change '. . .soil conditions differ between. . .' to . . .soil conditions differ, whereas climate conditions, . . .'.

This sentence was deleted.

l.95-102: From my point of view this paragraph belongs to the Material and Methods section.

This sentence was deleted.

l.104: Please replace 'where' by 'because'.

This sentence was deleted.

Materials and Methods

l.107: Please change 'referred' to 'referred to'. l.108: Please add 'located' after 'is'.

These were modified.

Please refer to line 149: The experimental site, hereafter referred to as the Montiers site (http://www.nancy.inra.fr/en/Outils-et-Ressources/montiers-ecosystem-research), is located in the Montiers-sur-Saulx beech forest in northeastern France (Meuse, France, latitude 48° 31' 54'' N, longitude 5° 16' 08'' E).

l.113: I would recommend giving the meanings of these abbreviations here.

The meanings of these abbreviations were added.

Please refer to line 154: The Montiers site is part of different national and international research networks, i.e., SOERE (Long-lasting observation and experimentation for the research on environment)-OPE (Perennial Environment Observatory) and F-ORE-T (Functioning of Forest Ecosystems), and AnaEE (Analysis and Experimentations on Ecosystems).

l.114: Please change 'are' to 'were' and add 'calculated' before 'from'.

This was modified.

Please refer to line 159: The mean annual rainfall and temperature over the last twenty years were 1069 mm and 9.8°C, respectively (calculated from Météo-France data).

l.117: Please add the scientific name of sycamore maple (Acer pseudoplatanus?).

This was added.

Please refer to line 166: The stand was mainly composed of beech (89%) and 11% of other deciduous species, i.e., sycamore maple (*Acer pseudoplatanus*), ash (*Fraxinus excelsior*), pedunculate oak (*Quercus robur* L.), European hornbeam (*Carpinus betulus* L.), and wild cherry (*Prunus avium*).

l.142: Please add 'at 130 cm height' after 'circumferences'.

This was added.

Please refer to line 193: Trees were chosen to cover most of the range of stem circumferences at 130 cm height (C130) in each plot.

l.154/155: Do you mean: 'Subsequently, the branches were separated. . .'?

Yes, the sentence was modified.

Please refer to line 205: Subsequently, the branches latter were separated into different classes, i.e., < 4, 4-7 and > 7 cm diameter, according to Henry et al. (2011).

l.155: Please add 'in' before 'diameter'.

This was added.

Please refer to line 206: the branches latter were separated into different classes, i.e., < 4, 4-7 and > 7 cm in diameter, according to Henry et al. (2011).

l.160: I would recommend deleting '(at least fifty kg of soil sample)'.

This was deleted.

Please refer to line 211: A two-step procedure was applied to accurately assess the fine root biomass (Bakker et al., 2008), without having to transport soil to the laboratory.

l.169: I guess you mean '20 cm depth', right?

Right so we modified the text.

Please refer to line 219: Roots with a diameter > 2 cm (small and coarse roots) were collected in February 2017 in three soil pits (approximately 0.4 m wide) for each plot where soil material was cut and extracted at approximately 20 cm depth.

l.171/172: Do you mean 'element concentration' instead of 'mineral content'?

Yes, this was modified.

Please refer to line 224: An aliquot of each root sample (fine, small and coarse) was then collected to determine element concentration.

l.172: Please add the magnification used for microscopical analyses.

This was added.

Please refer to line 226: The absence of soil particles was carefully checked under a binocular microscope with a magnification of 10x.

l.174: Did you check these samples, e.g., by SEM-EDX, to ensure that you removed all soil particles (especially the ones on a µm-scale)?

All samples were observed with binocular microscope but only some samples of fine roots were observed by SEM-EDX (see part 2.3.4. Microscopic analysis).

l.216: Please change 'spectroscopy' to 'spectrometer'.

This was changed.

Please refer to line 267: The samples were examined at the GeoRessources laboratory (University of Lorraine) for biomineral occurrence and composition, using a Hitachi S-4800 scanning electron microscope (SEM) equipped with an energy-dispersive X-ray spectrometer (EDX), containing a lithium-drifted Si detector.

l.232: Please add 'Titanium' before 'Ti' and set 'Ti' in brackets.

This was added.

Please refer to line 285: The percentage of soil mixed with the organic horizons was determined through the use of titanium (Ti).

l.243: You already introduced 'C130' as abbreviation before (l. 142).

This was corrected.

Please refer to line 296: To transform the stemflow volumes to a water flux, C130 was assumed to explain the inter-individual stemflow volume variability within a species.

l.265: Please replace 'D(X)' by 'DSi'.

This was replaced.

Please refer to line 318: where $D_{Si}$ is the drainage flux of Si, $D_G$ is the water drainage via rapid gravitational transfer,

l.278 & l.279: Please change 'C1.30' to 'C130'.

This was changed.

Please refer to lines 332: It included four steps, (i) the circumference of all trees was measured at 1.30 m height, $C_{130}$, in 2011 and 2015; (ii) eight trees in each plot, representing the range of $C_{130}$, stem bark and wood and 0-4, 4-7 and > 7 cm diameter branches were sampled; (iii) the weighed allometric equations fitted for each ecosystem compartment were calculated according to Calvaruso et al. (2017); and (iv) tree biomass (stem bark and wood and 0-4, 4-7 and > 7 cm diameter branches) was quantified per hectare by applying fitted equations to the stand inventories.

l.310: Please replace 'are' by 'were'.

This was replaced.

Please refer to line 364: The roots were not exported

l.318 & l.320 & l.326: Please replace 'kg of Si by ha-1.y-1' by 'kg Si ha-1 y-1'.

This was replaced in the whole manuscript.

l.332: How did you analyze the amorphous Si fraction (alkaline extraction?)? I cannot find it in the M&M section.

The data of the amorphous Si fraction are not presented in this manuscript. This was deleted.

Please refer to line 383: The normality of the distribution was checked, using the Shapiro-Wilk test. As our data did not follow a normal distribution, the non-parametrical Kruskal-Wallis test was performed to compare the different soil types, biomass pools, biomass increments, Si content, Si pools, and Si fluxes for each tree compartment, and the total soil Si at the threshold level of 0.05.

l.335: Why did you use 'year' as a factor here? You generally assume Si pools, in- and outputs to be more or less equal each year (otherwise you would not calculate means for the analyzed period 2012-2015), so you should not expect any time-related effects, right?

This was a mistake. The term "year" was deleted.

Please refer to line 389: We used the R version 3.3.1 statistical software (R Development Core Team, 2016) and specifically, the R package nlme to test the effect of soil type on annual Si fluxes, by means of a mixed linear analysis of variance (ANOVA) with soil type and their interaction as fixed effects.

l.337: If your data are not normally distributed (as you said before) you should use nonparametric tests only (i.e., the Mann-Whitney U test instead of the Student's test).

We do not agree, the Kruskal-Wallis test is also a non-parametric test used to test at least three samples. I join the procedure of selection of the statistical test for this study at the end of this document.

Results

l.348: Do you mean 'Aged' instead of 'Altered'? l.348 & l. 349: I would recommend using 'testate amoebae' instead of 'amoebae'.

Yes, this was corrected.

Please refer to line 402: Aged leaves in the organic horizon were colonized by hyphae and amoebae (Figure 1c) and presented large voids. The Si deposits disappeared from the plant cells but were present in the observed testate amoebae.

l.373: The numbers did not change after calculation? Please check again your calculations.

Sorry for the mistake. This was modified.

Please refer to line 427: The Si pools in the fine roots were important and ranged from 61.2 kg ha-1 in the RL to 98.7 kg ha-1 in the DC. Based on the turnover rate of fine roots, as determined by Brunner et al. (2013) for beech trees, i.e., 1.11 ± 0.21 y$^{-1}$, we calculated that the annual Si fluxes resulting from fine root decomposition ranged from 67.9 ± 14.3 kg ha-1 in the RL to 109.5 ± 23.0 kg ha-1 in the DC.

Results: I know the results section of a paper often is not like a thriller. However, you should try to make it at least easy to read. So please do not only repeat the data as they are already given in the figures and tables because this is quite exhausting to read. Please rework your results section (from my point of view, subsection 3.1.6 is a good example how to present you results in a more appropriate way).

The results section was rewritten to be less exhausting to read.

Please refer to lines 393 to 526:

**Example : 3.1.2 Si pools and fluxes in aboveground tree biomass**

The calculated standing aboveground biomass in 2011 increased as follows: RL < DC < EC with significant differences between EC and RL (factor 1.4). (Table 2). The stem bark had the highest Si concentration in the three plots, and the Si pool in this compartment represented approximately 40% of the total Si pool in the aboveground tree biomass. The younger the structures were, the higher Si concentration. Small branches were approximately three times more concentrated than coarse branches in the three soils (Table 2). The amount of Si immobilized in the standing aboveground biomass ranged from 20.1 kg ha$^{-1}$ on the RL to 26.2 kg ha$^{-1}$ on the EC. The annual biomass production between 2011 and 2015 increased as follows: RL < EC < DC with significant differences between DC

and RL (factor 1.7). As a result, the amount of Si immobilized in the aboveground biomass each year between 2011 and 2015 ranged from 0.98 kg ha$^{-1}$ on the RL to 1.82 kg ha$^{-1}$ on the DC.

l.414: Please use powers of 10 for such big numbers.

This was modified.

Please refer to line 477: The total Si pools in the first 90 cm of soil overpassed 2.4.10$^6$ kg ha$^{-1}$ in the DC and EC as opposed to approximately 7.2.10$^5$ kg ha$^{-1}$ in the RL.

ll.451-454: Please avoid to give redundant information (see 3.2.2) and to 'jump' between your figures (try to refer to every figure only one time).

We agree that it is not the optimal mean, but in this specific case, we have to make reference a second time to the figure 3 to make the link with the observations resulting from figure 5.

ll.456-459: Where can I find this information (Fig., Table?)?

This information is presented in the synthesis figure 6.

Discussion

ll.461-464: I would recommend deleting this paragraph.

We agree, this paragraph was deleted.

l.473: You should discuss your results in the context of the results of Maguire et al. (2017) here.

This was added.

Please refer to line 540 : This Si content in beech fine roots was very higher (2 to 6 times) than that measured by Maguire et al. (2017) for another deciduous species, i.e. sugar maple (*Acer saccharum*) in a cooler environment. Besides Maguire et al. (2017) demonstrated in this study that increased soil freezing significantly lowers the Si content of sugar maple fine roots.

l.484: Do you mean 'Sommer et al. (2013)' instead of 'Sommer et al. 2003'?

Yes, sorry for the mistake.

Please refer to lines 555: As demonstrated by Sommer et al. 2013, only a small fraction (approximately 1%; from 1.0 kg ha-1 in plot S3 to 1.8 kg ha-1 in plot S1) of the Si taken up by the tree stand accumulated each year in the perennial tree compartments, i.e., the stem, branch and coarse roots (Figure 6, Table 2).

l.497: I would recommend using 'Mineral soil content' instead of 'Soil pollution'.

This was changed.

Please refer to line 569 : 4.2.1 Mineral soil content in organic horizons

l.502: Please give an example for biological activities (e.g., bioturbation by earthworms).

This was added.

Please refer to line 573: The higher rate of soil pollution in the study of Cornelis et al. (2010a) can be explained by the presence of a thick Oh layer in the moder that was in direct contact with the superficial soil layer and was characterized by an intense mixing of degraded organic matter with soil

particles, induced by biological activities, mainly bioturbation by earthworms in these soils (Lavelle, 1988).

l.503: Please replace 'Si pollution' by 'Si input'.

This was replaced.

Please refer to line 575: The Si input by dust deposits in the organic horizons was negligible, with a maximum value of 6.0 kg ha$^{-1}$ y$^{-1}$ (no stand interception) against 151 to 246 kg ha$^{-1}$.

l.505: What did they study, dust deposits or Si in the litterfall? Please be more precise.

This was clarified.

Please refer to line 577: Lequy et al. (2014), who studied the mineralogy of the dust deposits of the Montiers site, observed that the Si deposits in throughfall was mainly quartz.

l.507: A space is missing between 'et al.' and '2017'.

This was corrected.

Please refer to line 581: The main phytogenic Si input into the organic horizons was opal phytoliths (Krieger et al., 2017), which dissolve slowly (Fraysse et al., 2009) in comparison to the rate of organic matter mineralization.

l.509: Please add 'that of' after 'than' and give references for these data.

This was added.

Please refer to line 582: The residence time of Si in the organic horizons is higher than that of carbon (5.3 ± 0.8 vs 1.9 ± 0.4 y).

l.510: Diverse taxa of testate amoebae synthesize SiO2-platelets for shell construction, but they do not possess a skeleton. l.511: Actually, testate amoeba shells represent the protozoic Si pool in soils and not the zoogenic one (which is represented by sponge spicules) (see Puppe et al. 2015).

This was corrected.

Please refer to line 583: In addition, the presence of testate amoebae, organisms rich in Si (Figure 1; Sommer et al., 2013), in the organic horizons suggests that a part of the Si from the phytoliths belonged to the protozoic Si pool.

l.512: I would recommend changing 'zoogenic pool could represent half. . .' to 'testate amoebae may use half . . . for shell synthesis'. l.513: I would recommend deleting 'in Europe' as Sommer et al. (2013) only analyzed one site (in Germany).

This was changed.

Please refer to line 585: Sommer et al. (2013) estimated that testate amoebae may use half of the Si input by litterfall in beech organic horizons (17 kg ha$^{-1}$ vs 34 kg ha$^{-1}$) for shell synthesis.

l.518: Another output flux is only likely if you assume balanced in- and outputs in general. From my point of view your data clearly indicate an accumulation of BSi in the organic layers. Please give some more references here to support your findings. l.520-522: Please give some references to support your assumption.

This paragraph was modified to be clearer, and some relevant literature was added to support our assumptions.

Please refer to lines : During the study period (2012-2015), the Si input in the organic horizons via litterfall were primarily higher than the Si output via soluble transport (assessed in ZTL solutions under

the forest floor) for the three soils. This net flux of Si should have induced the accumulation of Si in the organic horizons, what we did not observe in the four years of the study. This suggests that another output flux existed but was not quantified in our study. This flux is likely the solid particulate migration toward the topsoil layer, as demonstrated by Ugolini et al. (1977). These authors observed that organic particles containing notably silicon were predominant in the migrant material in the upper soil horizons. In our study, the solid particulate migration from the organic horizons to the topsoil may consist of the colloid transport of amoebae (Harter et al., 2000) or the transport of phytoliths (Fishkis et al. 2010). These latter observed, though a field study using fluorescent labelling, that the downrard transport distance of phytoliths after one year was 3.99± 1.21 cm for a Cambisol with a preferential translocation of small-sized phytoliths.

l.524: In general, the amorphous Si fraction includes pedogenic and biogenic Si. l.525: I would recommend adding Puppe et al. (2015) here as they analyzed testate amoebae and corresponding Si pools in detail.

The sentence was modified to integrate these remarks.

Please refer to lines : The Si production (source) in the soil mainly results from pedogenic Si from soil mineral dissolution and from biogenic Si from plant tissues and testate amoebae (Cornelis et al., 2011; Sommer et al., 2013; Puppe et al., 2015).

l.527: What do you mean with 'Si-amorphous'?

The sentence was modified to be more specific.

Please refer to line 606 : The immobilization (sink) of dissolved Si in the soil is due to plant and organism immobilization and precipitation of secondary minerals, such as phyllosilicates or Si-bearing short range organization minerals or allophane, immogolite (Dahlgren and Ugolini, 1989; Ma and Yamaji, 2006; Sommer et al., 2013; Tubana et al., 2016; Kabata-Pendias and Mukherjee, 2007).

l.529: Please replace 'until' by 'down to'.

"Until" was replaced by "down to"

Please refer to line 612: A net production of dissolved Si in the soils was observed on the three studied plots down to a depth of 60 cm, showing a positive production/immobilization budget.

l.532: Do you show these relationships in the results section? If not, you should do so.

The correlation coefficient was added in the text.

Please refer to line 615: This is corroborated by the strong relationship between annual Si production in the 10-60 cm soil layers and fine root content (data not shown, $r^2$ = 0.94).

l.534/535: Do you have data on this? If yes, you should present them in your paper. If not, please give some references to support your assumption. Do you have data on this: where testate amoebae and phytoliths accumulated after being transferred from the organic horizons.

We do not have data on this so we deleted this speculative part.

l.541: What do you mean with 'biogenic origin'? Please clarify.

This part of the sentence was deleted to avoid misunderstandings.

Please refer to line 622: At our site, this period was also characterized by a maximum concentration of Si in the bound waters and a negative budget in the 10-cm and 60-cm soil layers, resulting from the precipitation of secondary minerals.

542: Please change 'plant' to 'plants'.

This was changed.

Please refer to line 625: As a result, a drastic decrease of Si production was observed in the surface layer during the vegetation period, where Si uptake by plants occurred (Figure 3).

l.550: I would recommend starting here that BSi in general is more soluble than soil minerals (Fraysse and co-workers did some nice experiments on this).

The sentence was modified as suggested.

Please refer to line 638: Because biogenic Si in general is more soluble than lithogenic or pedogenic Si (Fraysse et al., 2009 ; Cornelis and Delvaux, 2016), very few of the Si leached within the soil profile directly results from the dissolution of soil minerals, as demonstrated in other studies in temperate forests (Bartoli, 1983; Watteau and Villemin, 2001; Gerard et al, 2008; Cornelis et al., 2010a; Cornelis et al., 2011a; Sommer et al., 2006; Sommer et al., 2013).

l.552: What about deforestation as an important Si output (anthropogenic desilicification)? Please also discuss this important factor and give corresponding literature.

This information was introduced in the manuscript.

Please refer to line 647 : Silicon inputs and outputs have minor contributions to the global Si budget in our forest ecosystems, and the Si cycle is mainly driven by internal fluxes, especially recycling of biogenic Si. However, Struyf et al. (2010) observed that land use is the most important controlling factor of Si mobilization in European watersheds. These authors showed that deforestation and conversion to agricultural land or other land uses leads to a twofold to threefold decrease in baseflow delivery of Si.

l.563: Please replace 'amoebae' by 'testate amoebae'. Do you have an idea about the population size of testate amoebae at your site (individual numbers)?

"Amoebae" was replaced by "testate amoebae". We do not assess the population size of testate amoebae in our study sit.

Please refer to line 661: In the organic horizons and in the soil, mainly in the 0-10 cm layer, we observed a high net Si production, likely resulting from the decomposition of litter leaves and testate amoebae in the organic horizons and of fine roots in the soil (Figure 6).

l.564-566: Please avoid to give redundant information.

We think that this information is partially redundant but important here.

ll.571/572: Please change '. . .strong influence of biological partners, mainly fine roots, and processes in the Si cycle' to '. . .strong biological influence mainly of fine roots.'

This was changed.

Please refer to line 668: The assessment of Si fluxes and pools in the different compartments of our forested site coupled with a seasonal dynamic follow-up reveal a rapid and almost total recycling of Si in our site and show the strong biological influence, mainly fine roots, and processes in the Si cycle.

l.576-579: Please avoid redundant information.

This sentence was deleted.

l.577 & l.578: Do you mean '3 x 103' and '0.7 x 103' here?

This sentence was deleted.

ll.580-583: It is known that the concentration of dissolved Si is a key factor for Si concentrations of plant components (as you also write in your introduction). So please give corresponding literature here and do not highlight this result as a new one. Furthermore, there is also a phylogenetic factor, i.e., phytolith production is probably more influenced by the phylogenetic position of a plant than by environmental factors like temperature or Si availability. In this context, you should also discuss and cite, for example, Hodson et al. (2005) (Hodson, M. J., P. J. White, A. Mead & M. R. Broadley (2005). Phylogenetic variation in the silicon composition of plants. Annals of Botany 96, 1027-1046).

Thank you for this interesting remark and paper. This information was added in the manuscript.

Please refer to line 681: The concentration of dissolved Si in the soil is known to influence opal formation in plants (Cornelis et al., 2010b) but phytolith production seems to be more affected by the phylogenetic position of a plant than by environmental factors (Hodson et al., 2005). For example, these authors demonstrated through meta-analysis of the data, that in general ferns, gymnosperms and angiosperms accumulated less Si in their shoots than non-vascular plant species and horsetails.

l.583: Please add 'in soils' after 'concentrations'.

This was added.

Please refer to line 679: This is in agreement with the observations of Heineman et al. (2016) in tropical forests, which demonstrated that nutrient concentrations in wood and leaves correlated positively with soil Ca, K, Mg and P concentrations in soils.

l.584. Why Si concentrations are higher especially in these plant components (leaves: transpiration termini; Roots: special protection of relatively fast growing fine roots)? Please give a more detailed discussion here and add corresponding literature.

A paragraph dealing with the importance of Si in leaves and roots was added.

Please refer to line 687: Silicon plays several physiological and ecological functions in leaves and roots, such as an involvement in the detoxification of aluminum, oxalic acid, and heavy metals, in the regulation of ion balance, in the reduction of hydric, salt, and temperature stresses (Currie and Perry, 2007; Meunier et al., 2017). They also contribute to the optimization of photosynthesis by gathering and scattering light in the leaves, confer mechanical support and tissue rigidity, and facilitate pollen release, germination, and tube growth (Bauer, Elbaum, & Weiss, 2011; Currie and Perry, 2007; Gal et al.,, 2012). In addition to these physiological functions, Si has also ecological significance by protecting plants against herbivores and phytopathogens (Currie and Perry, 2007; Lins et al., 2002).

**Conclusions**

A synthesis figure was added in the manuscript to summarize the main findings and compare with the data of other studies in similar stand conditions.

Please refer to line 951: **Fig. 7:** Summary scheme of the main findings of this study (TS) and comparison with other studies (L).

[Figure]

l.596: Please be careful with statements like 'the complete Si cycle'. I would recommend using 'the Si cycle' instead.

We agree and thus modified.

Please refer to line 703: By coupling different approaches (annual budget in solid vegetal and solution phases and monthly dynamics of solutions) and methods (direct *in situ* measurements and standard and site specific modelling) to quantify Si pools and fluxes in the different ecosystem compartments, our study allowed us to assess the Si cycle at the forest stand scale.

l.601: Please replace 'to give dissolved Si' by 'in the form of dissolved Si'.

This was replaced.

Please refer to line 709: This suggests that Si cycle is almost closed during the vegetation period; dissolved Si is taken up by vegetation then Si returned to the soil mainly through root and leave decomposition in the form of dissolved Si, which is again taken up by vegetation.

l.603: Please add 'on a decadal time scale' after 'biogeosystem'.

This was added.

Please refer to line 711: This observation is consistent with the observation of Sommer et al. (2013), who demonstrated a low contribution of geochemical weathering processes to the Si cycle in a forest biogeosystem on a decadal time scale.

l.606: Please add 'concentrations' after 'Si'.

This was added.

Please refer to line 715: The plant compartments were Si-enriched in the soil with higher Si concentration, i.e., DC (plot S1) compared to plant compartments in the RL (plot S3), resulting in 1.6-times higher recycling in plot S1 compared to plot S3.

l.608/609: I would recommend using 'release' or 'instead of 'production'.

This was changed.

Please refer to line 718: While Si release was relatively similar in the organic horizons for the three plots, its production in the soil, mainly in the 0-10 cm layer, was twice higher in plot S3 and richer in clays than plot S1

Figure captions

Fig. 1: Did you use EDX for elemental analyses?

Yes, the presence of Si was confirmed by EDX for each point with arrows. I join below one of the spectrum carried out on fine roots.

[Figure]

l.760: Please replace 'amoebae' by 'testate amoebae'. l.761: Please change 'altered' to 'aged' and replace 'testate amoebae' by 'testate amoeba shells'.

This was replaced.

Please refer to line 916 : **Fig. 1:** Si in biological tissues of beech trees observed through Scanning Electron Microscopy. (a) Si precipitates in the intercellular space of fresh leaves, forming phytoliths (white arrow). Deposits of Si (vertical white arrows) in the inner cell walls of fruit capsules (b), stem bark (d and e), bud scales (f), and roots (g, h, and i). (c) Hyphae, testate amoebae and large voids in aged litter leaves. Si deposits only present in the testate amoeba shells (horizontal white empty arrows). The presence of Si was confirmed with EDX (analyzed zones indicated by white vertical arrows).

l.774: Please replace 'Histograms' by 'Bars'.

This was replaced.

Please refer to line 935: Bars with an asterisk are significantly different from 0, according to a Kruskal-Wallis test at the threshold P value level of 0.05.

Tables

l.792: Please replace 'Are presented the mean values. . .' by 'Presented are the mean values. . .'

This was modified.

Please refer to line 953: **Table 1:** Physicochemical properties of the three studied soils in the Montiers site (plot S1 – DC; plot S2 – EC; plot S3 – RL). Presented are the mean values for bulk density (g cm$^{-3}$), textural distribution (g kg$^{-1}$), total rock volume (RV), soil water holding capacity (SWHC), soil water pH, organic matter content (OM), cation exchange capacity (CEC; cmol+ kg$^{-1}$) and base-cation saturation ratio (S/CEC, with S = sum of base cations). Standard deviation values are given in italic. Table adapted from Kirchen et al. (2017).
l.798: Please specify which differences were evaluated (DC vs. EC vs. RL?).

This was specified.

Please refer to line 960: **Table 2:** Mean Si contents, pools and fluxes in the biomass of the three soils of the Montiers site. Standard deviation values are given in brackets. Values with different letters are significantly different according to a Kruskal-Wallis test at the threshold P value level of 0.05 (soil effect, DC vs. EC vs. RL).
Table 3: Why did you test only the total soil depth on statistical significance? Interestingly, the upper compartments are quite comparable at the three sites, only in the deeper soils (30-90 cm) there seem to be significant differences.

Statistical significance was added for each depth.

Please refer to line 965.

Table 4: I cannot find any letters marking statistical significances. Better use 'Si concentration' for Si in g kg-1 or mg l-1 instead of 'Si content'. What about the mineral composition of the different soils? It would be nice to have also data on this.

Letters marking statistical significances were added and Si concentration was used.

Please refer to line 970.

Information regarding the geology and the mineralogy of the site was also added but details were already presented in Calvaruso et al. (2017)

Please refer to line 160: The geology of the Montiers site consists of two overlapping soil parent materials: an underlying Tithonian limestone surmounted by detrital acidic Valanginian sediments. The calcareous bedrock contains mainly calcium carbonate and ~3.4% clay minerals. The overlying detrital sediments are complex, as they result from various depositions and are composed of silt, clay, coarse sand and iron oxide nodules (for more details, see Calvaruso et al., 2017).

Figures

Fig. 1: What are the black arrows pointing at (micrograph c)? Please specify or give uniform arrows.

The empty horizontal white arrows indicate the location of testate amoebae. This was added in the legend of Figure 1.

Please refer to line 916: **Fig. 1:** Si in biological tissues of beech trees observed through Scanning Electron Microscopy. (a) Si precipitates in the intercellular space of fresh leaves, forming phytoliths

(vertical white arrow). Deposits of Si (white arrows) in the inner cell walls of fruit capsules (b), stem bark (d and e), bud scales (f), and roots (g, h, and i). (c) Hyphae, testate amoebae and large voids in aged litter leaves. Si deposits only present in the testate amoeba shells (horizontal empty white arrows). The presence of Si was confirmed with EDX (analyzed zones indicated by white vertical arrows).

Fig. 2 & 3: Why do you use single data of four years in one diagram (Fig. 2) and means with standard deviations in another one (Fig. 3)? I would recommend unifying the presentation of your results.

The objectives of the two figures are different. In the Figure 2, we want to show the seasonal and inter-annual variations (on four years) of the dissolved Si in the throughfall solution. In the figure 3, we want to show the seasonal variations over four years of the dissolved Si in the different soil compartments.

Fig. 5: Please correct the unit of dissolved Si.

This was corrected.

Fig. 6: As you can go full color in this journal I would recommend using different colors for data of the different sites.

Good idea so we did it.

Please refer to line 944: **Fig. 6:** …For each pool and flux, values presented are those of the plots S1 (in green), S2 (in orange), and S3 (in blue), respectively…

Fig. 6: Please correct the values of soil Si pools (x103).

This was corrected.

Fig. 6: What about soil pH effects? I especially wonder at Si drainage values of S3 (0-10 cm). You should also give a more detailed discussion on this aspect.

Interesting suggestion, however too much soil parameters, mainly soil texture/structure, affect the drainage and it is complicated to discriminate the influence of each one. So we prefer to do not deal with the effect of pH on drainage in this paper. Maybe in another one. However we added a part in the discussion where we compare the drainage flux in our study site with other data in the literature, and we succinctly present the possible origin of the differences observed.

[revised manuscript text omitted]

---

## Referee Report (RR1)

I welcome the changes made to the document. The authors have done a good job at addressing all review suggestions, and I think this manuscript should be published in Biogeosciences.

---

## Author Response (AR2)

Reviewer 1:

The authors would like to thank reviewer #1 for this second careful reviewing of the manuscript. We apologize for the numerous mistakes forgotten in the manuscript. The revised version of the manuscript was carefully reviewed by the authors as well as by an independent internal reviewer. Each of your remark (and several drafting corrections) was introduced in the revised version of the manuscript as you can see in the responses below and in the manuscript with tracks (first part of the provided pdf file, called revised manuscript 2). Please find below your remarks (in black), and our responses and the location of the modifications brought to the text (in blue).

First of all I would like to thank the authors (Turpault et al.) for their response. However, I am quite unsatisfied with the revised manuscript, although Turpault et al. answered all comments of reviewers # 1 & 2 in an extensive way.

I know, errare humanum est (to err is human) and I do not want to be too fussy, but the amount of errors in the current manuscript is just too big to be ignored, which makes it really frustrating to read and to review the article at all. This is especially annoying as this manuscript was approved by five (!) authors in its current form.

From my point of view the manuscript in its current form is not publishable, although I think the results generally are of interest for the readers of BIOGEOSCIENCES.

I recommend Turpault et al. a second and careful major revision of their manuscript. In this context, the authors should address the following points:

• Please double-check the citations and the reference list according to the following points:

- A lot of citations were deleted (especially in the introduction), but still appear in the reference list (e.g., Iler, 1979; McKeague and Cline, 1963; Dixon and Weed, 1989) on the one hand.

- On the other hand, a lot of citations were added, but are not listed in the reference list (e.g., Struyf and Conley, 2012; Puppe et al., 2014; Conley, 2002).

- Some citations are not correct at all (e.g., 'Sommers et al.' should be 'Sommer et al.', what is 'Conley et al., 2002'? Do you mean 'Conley, 2002'?).

- Please avoid redundant credits (e.g., change 'Cornelis et al., 2010a; Cornelis et al., 2011a' to 'Cornelis et al. 2010a, 2011a').

- Please correct 'et al' to 'et al.'.

We carefully checked and corrected the references. All references cited in the text are in the list of references and vice-versa. All the references in the text have the same form.

• If you introduce an abbreviation you should use it (so please change 'biogenic Si' to 'BSi').

Biogenic Si was replaced by BSi in all the manuscript.

• Diatoms and testate amoebae are no animals by definition (animals are multicellular, eukaryotic organisms). Please correct to 'Other important producers of BSi are sponges and protists (diatoms, testate amoebae)…'.

Thank you for this remark, this was corrected as suggested.

Please refer to lines 51 to 52: Other important producers of biogenic Si are sponges and protists (diatoms, testate amoebae) (Struyf and Conley, 2012; Sommer et al., 2006; Puppe et al., 2014; Puppe et al., 2015).

• Please change 'Dsi' to 'DSi'.

This was corrected and the abbreviation DSi is used instead of dissolved Si in all the manuscript.

• I would recommend adding 'Maguire et al.' in the introduction as this article is quite important regarding the current knowledge of the scientific background of your study.

We agree with this interesting suggestion. The study of Maguire et al. (2017) was added in the introduction.

Please refer to lines 67 to 72: Maguire et al. (2017), who examined the impact of climate change on Si uptake by trees, observed that fine roots of sugar maple (*Acer saccharum*) which represented only 4% of the tree's biomass, accumulated 29% of the Si. Considering the high Si content of fine roots (Krieger et al., 2017; Maguire et al., 2017) and their rapid turnover in forest ecosystems (approximately one year in beech forests in Europe; Brunner et al., 2013), we hypothesized that fine roots could significantly contribute to the input of BSi into the soil.

• You state in your answers that 'all samples were observed with binocular microscope but only some samples of fine roots were observed by SEM-EDX (see part 2.3.4. Microscopic analysis)'. However, in the corresponding section of your manuscript I can read 'The samples were examined at the GeoResources laboratory … using a scanning electron microscope…'. This suggests that all samples were analyzed with SEM-EDX. Please specify in your manuscript which samples were analyzed and how these samples were chosen. In addition, you should, of course, mention (if only shortly) the results of these analyses in your manuscript, because checking your fine root samples for adhering soil particles (as a Si source) is crucial for your work.

This has been clarified in the revised version of the manuscript.

Please refer to lines 158 to 165: An aliquot of each root sample (fine, small and coarse) was then collected to determine element concentration. Each aliquot was carefully washed under a binocular microscope with distilled water, using tweezers and an ultrasound gun. The absence of soil particles was carefully checked on each root sample under a binocular microscope with a magnification of 10x. The operation was repeated until all soil particles were removed to prevent soil pollution in the root analyses. A second check using a scanning electron microscope (SEM) equipped with an energy-dispersive X-ray spectrometer (EDX) was carried out on 12 randomly selected sub-samples of fine roots by plot (for more details, see part 2.3.4). All observed sub-samples were free from soil particles.

Please refer to lines 202 to 208:

**2.3.4 Microscopic analysis**

Between 9 and 12 randomly selected samples of fine roots, stem and branch bark, fruit capsules, bud scales and fresh and altered leaves (from organic horizons) collected on beech trees for each plot were mounted on glass plates, using double-coated carbon conductive tabs and covered with carbon. These samples were examined at the GeoRessources laboratory (University of Lorraine) for biomineral occurrence and composition, using a Hitachi S-4800 SEM equipped with an EDX, containing a lithium-drifted Si detector. The SEM analyses were carried out using an acceleration voltage of 10 or 15 kV.

• You state in your answers that 'the Kruskal-Wallis test is also a non-parametric test used to test at least three samples'. This is correct. However, in the corresponding point of criticism in my first review I referred to your statement 'The significance of differences in element content between the gravitational and bound solutions and between plots was tested by the Student's t-test' in your manuscript. So again my point: If your data are not normally distributed (as you said before) you should use nonparametric tests only (i.e., the Mann-Whitney U test instead of the Student's t-test).

Sorry for the mistake in the statistical analysis section. The significance of differences in Si content and fluxes in solution phases (as well as for solid phases) between the three plots was indeed tested by the non-parametrical Kruskal-Wallis test, as explained in the caption of Tables 2, 3 and 4.

The significance of differences in Si content between the gravitational and bound solutions was now tested by the Mann-Whitney U test. Asterisks were also added in Figure 4 to show the significant differences between gravitational and bound solutions.

Please refer to lines 320 to 326: As our data did not follow a normal distribution, the non-parametrical Kruskal-Wallis test was performed to determine the significance of differences in biomass pools and increments, Si content, pools, and fluxes for each tree compartment, total Si content and pool in soil, and Si content and fluxes in soil solutions between the three soils, at the threshold level of 0.05. The post hoc Bonferroni correction was used for the pairwise comparison. The non-parametrical Mann-Whitney U test was also performed to determine the significance of differences in Si content and Si fluxes between gravitational and bound solutions by soil layer for each soil type, at the threshold level of 0.05.

Please refer to lines 780 to 785: **Fig. 4:** a. Mean DSi concentration over four years (January 2012 to December 2015) in zero-tension lysimeters (ZTL) and tension lysimeters (TL) at different soil depths (0-10 cm, 10-30, 30-60, and 60-90 cm) in plots DC and RL. For each soil type and depth, values with an asterisk are significantly different according to a Mann-Whitney U test at the threshold P value level of 0.05 (solution type effect, ZTL vs. TL). b. Seasonal dynamics over four years (January 2012 to December 2015) of DSi concentrations in ZTL and TL in the 0-10 cm and 10-30 cm soil layers of plot RL.

Please refer to lines 824 to 827: Table 4: Si content and fluxes in the ZTL (Zero Tension Lysimeters) and TL (Tension Lysimeters) solutions of the three soils of the Montiers site. Standard deviation values are given in brackets. Values with different letters are significantly different according to a Kruskal-Wallis test at the threshold P value level of 0.05 (soil effect, DC vs. EC vs. RL).

• L. 382/383: Please correct '2.4.106' and 7.2.105' to '2.4 x 106' and 7.2 x 105'.

This was corrected

Please refer to lines 388 to 389: The total Si pools in the first 90 cm of soil overpassed $2.4 \times 10^6$ kg ha$^{-1}$ in the DC and EC as opposed to approximately $7.2 \times 10^5$ kg ha$^{-1}$ in the RL.

• I miss a reference to figures 6 and 7 in your results section.

A reference to the synthesis Figure 6 was added in the results section (Tree uptake data not presented in other tables and figures).

Please refer to lines 428 to 431: By adding the amounts of the Si accumulated each year in the different tree compartments, i.e., perennial aboveground biomass, leaves, bud scales, beechnuts and fruit capsules, small and coarse roots, and fine roots and the foliar leachate, we determined that the annual uptake of Si by the stand was approximately 157, 141, and 95 kg ha$^{-1}$ in plots DC, EC, and RL, respectively (Figure 6).
The Figure 7 which only summarizes data presented in other tables and figures, is only cited in the conclusion section.

• L. 436: Please correct '…fine roots was very higher…' to '…fine roots was higher…'

This was corrected.

Please refer to lines 442 to 444: The Si content in beech fine roots was higher (2 to 6 times) than that measured by Maguire et al. (2017) for another deciduous species, i.e. sugar maple (*Acer saccharum*) but in a cooler environment.

• L. 439: Too many dots after 'sugar maple fine roots'.

This was corrected.

Please refer to lines 444 to 445: Besides Maguire et al. (2017) demonstrated in this study that increased soil freezing significantly lowers the Si content of sugar maple fine roots.

• L. 441 & 443: You still use incorrect units ('t.ha-1'). Please follow the rules of the 'International System of Units'.

This was corrected in the text and in the figures.

• L. 477: Please correct '…a part of the Si from the phytoliths belonged to the protozoic Si pool' to '…a part of the BSi belonged to the protozoic Si pool'.

This was modified as suggested.

Please refer to lines 481 to 483: In addition, the presence of testate amoebae, organisms rich in Si (Figure 1; Sommer et al., 2013), in the organic horizons suggests that a part of the Si belonged to the protozoic Si pool.

• By the way, you did not explain 'protozoic Si pools' in your introduction. This makes it difficult for the reader to follow, as not every reader is a specialist in Si cycling and BSi pools. You should give all relevant knowledge for the understanding of your work in the introduction or at least give a short explanation in the corresponding passage.

We agree with this remark, so we introduce the "protozoic Si pool" in the introduction.

Please refer to lines 52 to 54: In terrestrial ecosystems BSi pools can be separated in phytogenic (phytoliths), microbial and protozoic pools, the latter represented in soils by idiosomic testate amoebae (Puppe et al., 2014).

• L. 493: What is meant by '(51 6)'?

Sorry this was a mistake. This was corrected.

Please refer to lines 498 to 500: The Si production in the soil mainly results from pedogenic Si and BSi resulting from soil mineral dissolution and plant tissues and testate amoebae degradation, respectively (Cornelis et al., 2011; Sommer et al., 2013; Puppe et al., 2015).

•       L. 567: Please cite 'Bauer, Elbaum & Weiss' as 'Bauer et al.'.

This was corrected.

Please refer to lines 567 to 569: Silicon also contributes to the optimization of photosynthesis by gathering and scattering light in the leaves, confer mechanical support and tissue rigidity, and facilitate pollen release, germination, and tube growth (Bauer et al., 2011; Currie and Perry, 2007; Gal et al., 2012)

•       Fig. 2: Please correct the unit (y-axis).

This was corrected.

•       Fig. 3: I miss a caption of the y-axis.

This was corrected.

•       Fig. 4: Please correct the units (y-axis).

This was corrected.

•       Fig. 6: Not all data are given in the corresponding colors (see data for 'organic horizons' and 'small dead woods').

This was corrected.

In addition, as suggested by the editor, the readability of the soil compartment was improved.

•       Fig. 7: Please state references for 'L' in the caption of Fig. 7 and add units for the presented data.

The references and units were added in the caption.

Please refer to lines 806 to 807: **Fig. 7:** Summary scheme of the main findings of this study (TS) and comparison with other studies carried out in beech temperate forests (L, Bartoli, 1983; Cornelis et al., 2010a; Sommer et al., 2013). The Si stocks and fluxes are in kg ha$^{-1}$.

[revised manuscript text omitted]

50 and translocated into biogenic Si (BSi) under opal form which is deposited into the cell walls, cell luminas and intercellular spaces (Jones and Handreck, 1965; Conley , 2002; Cornelis et al., 2010b1b; Struyf and Conley, 2012). These structures are called phytoliths. Other important producers of  are  sponges and protists (diatoms, testate amoebae) (Struyf and Conley, 2012; Sommer et al., 2006; Puppe et al., 2014; Puppe et al., 2015). In terrestrial ecosystems BSi pools can be separated in

55 phytogenic  (phytoliths), microbial and protozoic poolis latter represented in soils by idiosomic testate amoebae (Puppe et al., 2014).

According to Conley (2002), the annual fixation of DSi into terrestrial ecosystems has been estimated to range from 60 to 200 Tmoles. That represents 10 to 40 times more than yearly export DSi and suspended  from the terrestrial geobiosphere to the coastal zone (Conley, 2002). Vegetation can thus be considered as a factory

60 of BSi which returns to the soil as organic matter through biological recycling. Because BSi in general is more soluble than silicate minerals, BSi strongly contributes to the DSi pool (Fraysse et al., 2009; Cornelis and Delvaux, 2016).

Based on the assumption that the storage of Si is limited in roots (Bartoli and Souchier, 1986) and because fine root sampling and cleaning before analyses are long and tedious processes, studies in forest ecosystems mainly

65 focus on the importance of litterfall recycling on the Si biogeochemical cycle without quantifying Si in the roots (Gérard et al., 2008; Cornelis et al., 2010a; Sommer et al., 2013).

However, Krieger et al. (2017) recently showed that Si in deciduous trees (European beech, *Fagus sylvatica* and sycamore maple, *Acer pseudoplatanus*) generally precipitates as a thin layer (< 0.5 µm) around the cells, especially in roots and bark. These small-scale phytogenic Si  were demonstrated to influence various soil and plant

70 processes (Meunier et al., 2017; Puppe et al., 2017). Maguire et al. (2017), who examined the impact of climate change on Si uptake by trees, observed that fine roots of sugar maple (*Acer saccharum*) which represented only 4% of the tree's biomass, accumulated 29% of the Si.

Considering the high Si content of fine roots  (Krieger et al., 2017; Maguire et al., 2017) and the rapid turnover  
[revised manuscript text omitted]

330 significance of differences in biomass pools and increments, Si content, pools, and fluxes for each tree compartment, total Si content and pool in soil, compare the different soil types, biomass pools, biomass increments,

Si content, Si pools, and Si fluxes for each tree compartment and, Si content and Si fluxes in soil solutions, and the total soil Si between the three soils, at the threshold level of 0.05. The post hoc Bonferroni correction was used for the pairwise comparison. The non-parametrical Mann-Whitney U test was also performed to compare determine the significance of differences in Si content and Si fluxes between gravitational and bound solutions by soil layer for each soil type, at the threshold level of 0.05. 
[revised manuscript text omitted]
 the Si at the stand scale  and  underline the  key-role of biological processes, mainly fine roots,  in the Si cycle.

**4.5 Soil influence in the soil Si inputs/outputs**

We showed that the Si content of plant compartments (leaves, organic horizons, aboveground and belowground biomasses) was higher in the Si rich soils (plots DC and EC) compared to that of plot RL. This is in agreement with the observations of Heineman et al. (2016) in tropical forests, which demonstrated that nutrient concentrations in wood and leaves correlated positively with  Ca, K, Mg and P concentrations in soils. The concentration of

585 DSi in the soil is known to influence opal formation in plants (Cornelis et al., 2010b) but phytolith production seems to be more affected by the phylogenetic position of a plant than by environmental factors (Hodson et al., 2005). For example, these authors demonstrated through meta-analysis of the data, that in general ferns, gymnosperms and angiosperms accumulated less Si in their shoots than non-vascular plant species and horsetails. Moreover, the annual tree compartments (leaves and fine roots) were more concentrated in Si than the

590 perennial compartments (branches, stem and coarse roots). Silicon plays several physiological and ecological functions in leaves and roots, such as an involvement in the detoxification of aluminum, oxalic acid, and heavy metals, in the regulation of ion balance, in the reduction of hydric, salt, and temperature stresses (Currie and Perry, 2007; Meunier et al., 2017).  Silicon also contribute to the optimization of photosynthesis by gathering and scattering light in the leaves, confer mechanical support and tissue rigidity, and facilitate pollen release,

595 germination, and tube growth (Bauer et al., 2011; Currie and Perry, 2007; Gal et al., 2012). In addition to these physiological functions, Si has also ecological significance by protecting plants against herbivores and phytopathogens (Currie and Perry, 2007; Lins et al., 2002). The variations of Si content in the annual tree compartments induced by the soil type significantly affected the Si fluxes in the ecosystem. The annual uptake and Si recycling (leaves  and buds, beechnuts, fruit capsules  and fine roots) were 127.2 and 154.0 kg ha$^{-1}$,

600 respectively, in plot DC, as opposed to 94.8 and 92.7 kg ha$^{-1}$, respectively, in plot RL.

In return, the bound solutions were more concentrated in Si in plot RL compared to in plot DC. This is partly due to the higher clay content in plot RL compared to in plot DC (clay was two times higher in plot RL). This considerably increases the specific surface area of minerals and improves their weatherability and water retention capacity (Carroll and Starkey, 1971; De Jonge et al., 1996).

605 **5 Conclusion**

By coupling different approaches (annual budget in solid vegetal and solution phases and monthly dynamics of solutions) and methods (direct *in situ* measurements and standard and site specific modelling) to quantify Si pools and fluxes in the different ecosystem compartments, our study allowed us to assess the Si cycle at the forest stand scale. Interestingly, our study highlights the main contribution of fine roots and, to a lesser extent, of leaves in the

610 Si cycle (Figure 7). Almost all the DSi was taken up by trees at any given time (very weak leaching out of the soil profile) and was recycled each year (approximately 99%, only 1% accumulated in perennial tissues). This suggests that the Si cycle is almost closed during the vegetation period; DSi is taken up by vegetation then Si returned to the soil mainly through root and leave decomposition in the form of DSi, which is again taken up by vegetation. This observation is consistent with th  of

615 Sommer et al. (2013), who demonstrated a low contribution of geochemical weathering processes to the Si cycle in a forest biogeosystem on a decadal time scale. The seasonal dynamics of DSi confirmed the key role of biological processes in the Si cycle, notably through the massive production of DSi during the decomposition of fine roots. Our study also revealed that soil type influences the Si accumulation in tree and the Si production in the soil.  Trees  accumulated more Si when developed

620 on a Si-richer soil such as DC, resulting in a higher Si recycling (factor 1.6) . While Si release was relatively similar in the organic horizons for the three plots, its

production in the soil, mainly in the 0-10 cm layer, was twice higher in plot RL,  richer in clays than in plot DC.

625 Further research is needed in the mid-term (i) to assess the mineralisation speed of fine roots in the soil and the speed of transformation of the root BSi  into DSi, (ii) to determine the annual and seasonal fate of the DSi  issued from roots;  between uptake, mineral precipitation, drainage and  fixation by organisms, and (iii) to quantify the vertical transfer of solid particulates between organic horizons and topsoil.

**Acknowledgement**

We acknowledge S. Didier for site implementation and management, L Saint André, L. Franoux and A. Genêt for the development of allometric equations, C. Pantigny, L. Gelhaye, B. Simon, C. Nys, J. Mangin, C. Goldstein, F. César, M. D'Arbaumont and M. Simon for technical help, L. Salsi,  for preparing the samples and performing the SEM and EDX analyses,  the National Forest Office (ONF)  for welcoming our experimental site in the domanial forest of Montiers and for the stand management, and American Journal Experts and Krista Bateman for the English reviewing of the paper, as well as the reviewers which have, through their suggestions, significantly improved the quality of the manuscript.  The authors acknowledge the facilities of the French National Institute for Agricultural Research and the Service d'Analyse des Roches et des Minéraux of the French National Center for Scientific Research. This work was supported by the Andra and INRA (accord spécifique N°9) and GIP-Ecofor (contract N°1138451B).

The authors declare that they have no conflict of interest.

**Fig. 5:** Mean annual DSi budget in the different layers (-forest floor, FF; soil 0-10 cm; soil 10-30 cm; soil 30-60 cm; and soil 60-90 cm) for the three plots DC, EC, and RL. Bars represent the standard deviations. Positive and negative values represent the production and immobilization of DSi in the given layer, respectively. Bars with an asterisk are significantly different from 0, according to a Mann-Whitney U test at the threshold P value level of 0.05.

**Fig. 6:** Summary scheme of Si cycling in plots DC, EC and RL of our study forest site, including (i) pools of Si in the biomass, (ii) internal Si fluxes, i.e., in the soil-plant system, (iii) external Si fluxes entering or leaving the soil-plant system, and (iv) the DSi budget in the different layers of the ecosystem. Pools are presented by rectangular boxes (tree annual and perennial parts, organic horizons and small dead wood, and soil). Internal fluxes (solid form from the tree to the soil, i.e., fine roots, litterfall including leaves, buds and branches, and exploitation residues; and in solution from the soil to the plant, i.e., the tree uptake) are presented in boxes with rounded edges. Grey/black arrows indicate the direction and the intensity of the internal fluxes. The external fluxes (inputs: rainfall and dust deposits, and outputs: drainage and biomass harvest) are presented in flag boxes. For each pool and flux, values presented are those of the plots DC (in green), EC (in orange), and RL (in blue), respectively. The DSi budget in the different layers (forest floor and different soil layers) are represented with white arrows, which indicate the direction and the intensity of the fluxes. Arrows leaving the layer indicate the production of DSi in this layer. In contrast, arrows entering the layer indicate the immobilization of DSi in this layer. Values presented in each box and arrow are annual mean values for plots DC, EC, and RL, respectively (except for atmosphere values which are similar for the three plots). The AG and BG correspond to aboveground and belowground tree compartments, respectively.

**Fig. 7:** Summary scheme of the main findings of this study (TS) and comparison with other studies (L, Bartoli, 1983; Cornelis et al., 2010a; Sommer et al., 2013). The Si stocks and fluxes are in kg ha$^{-1}$ of Si.

[revised manuscript text omitted]